# Random features for Grassmannian kernel approximation with bounded rank-one projections

## Abstract

We propose a family of random feature maps for scalable kernel machines defined over low-dimensional subspaces in high dimensions, *i.e.*, over the Grassmannian manifold. This is typically useful in a machine learning context when data classes or clusters are well represented by the span of a few data points. Classical Grassmannian kernels such as the *projection* or *Binet–Cauchy* kernels require constructing full Gram matrices for practical applications, leading to prohibitive computational and memory costs for large subspace datasets in high dimensions. We address this limitation by computing specific random features of subspaces. These combine random rank-one projections of the subspace projection matrices with bounded non-linear transforms—periodic or binary—to tame the resulting heavy-tailed distribution. We show that, in the random feature space, inner products approximate well-defined, rotation-invariant Grassmannian kernels, *i.e.*, depending only on the principal angles of the considered subspaces. Provided the number of random features is large compared to the subspace intrinsic dimension, we show that this approximation holds uniformly over all subspaces of fixed dimensions with high probability. When the non-linear transform is periodic, the approximated kernel admits a closed-form expression with a tunable behaviour bridging inverse Binet–Cauchy and Gaussian-type regimes, while the binarised feature has no known closed-form kernel but lends itself to even more compactly represented one-bit subspace features. Moreover, we show how structured rank-one projections, leveraging randomised fast Fourier transforms, further reduce the random feature computational complexity without sacrificing accuracy in practical experiments. We demonstrate the practicality of these techniques with synthetic experiments and classification tasks on the ETH-80 dataset representing visual object images from different viewpoints. The proposed random features recover Grassmannian geometry with high accuracy while reducing computation, memory, and storage requirements. This demonstrates that rank-one embeddings offer a practical and scalable alternative to classical Grassmannian kernels.

## 1 Introduction

There exist many supervised and unsupervised learning tasks where data classes or clusters are better represented by low-dimensional *subspaces* spanned by the instances of a specific class of data. This assumption arises naturally in a range of contexts such as image and video processing (Watanabe & Pakvasa, 1973; Basri & Jacobs, 2003; Rao et al., 2010; Liu et al., 2013; Ji et al., 2015), wireless communications (Schwarz & Tsiftsis, 2021), dynamic subspace modelling and signal processing (Srivastava & Klassen, 2004; Saad-Falcon et al., 2024; Jayasumana et al., 2015). Implicitly, this means that, rather than working with separate data points in a Euclidean space, we now find ourselves dealing with data belonging to the *Grassmannian manifold* $\mathcal{G}(k, n)$, the set of all subspaces of dimension $k$ in $\mathbb{R}^n$.

In supervised learning, provided one accesses enough training data, *kernel machines* (*e.g.*, SVM, regression (Bach, 2024)) can learn arbitrary complex function or classification boundaries thanks to an appropriate positive-definite *kernel*, *i.e.*, a similarity score computed over all pairs of data and stored in a *Gram matrix*. In the context of Grassmannian data, there exist numerous kernels to quantify similarity between Grassmannian

elements (see *e.g.*, (Harandi et al., 2014) for a list of such *kernels*), hence allowing for complex learning tasks (Hamm & Lee, 2008; Wolf & Shashua, 2003).

Although effective in theory, kernel machines suffer from a scalability drawback. As noted in Rahimi & Recht (2007), large datasets (with $N$ samples) require handling an $N \times N$ Gram matrix, hence requiring $\mathcal{O}(N^2)$ (possibly costly) evaluation of the kernel and an $\mathcal{O}(N^2)$ memory cost. A first fix to this problem is to approximate the Gram matrix, *e.g.*, from a low-rank approximation as in the Nyström method (Drineas & Mahoney, 2005; Bach & Jordan, 2005). **[R-XJL8]** This reduces the storage cost to $\mathcal{O}(mN)$, where $m \leqslant N$ is the number of selected samples but the method remains data-dependent since the approximation is built from samples selected from the dataset and still requires kernel evaluations involving all $N$ samples. *Random features* provide another efficient solution directly approximating the kernel: each data point is embedded into a Euclidean space thanks to random projections, possibly coupled with non-linear functions (*e.g.*, periodic, binary). In this space, mere inner products between random features approximate, with a controlled error, a specific kernel between pairs of initial data points (Rahimi & Recht, 2007). For Grassmannian data, the compressive sensing literature provides us direct random feature constructions (Foucart, 2016; Candès & Plan, 2011) (see Sec. 3.1). We can embed Grassmannian data by vectorising and projecting them onto a few random Gaussian matrices, their number scaling with the intrinsic dimension of $\mathcal{G}(k,n)$ (Li & Gu, 2018). However, the cost of such dense Gaussian projections makes this approach prohibitive both computationally and in terms of memory.

**Main contributions.** In this work, we propose novel random features adapted to Grassmannian data. More precisely, each subspace $\mathcal{P} \in \mathcal{G}(k,n)$ is represented by its orthogonal projector $\boldsymbol{P} = \boldsymbol{U}\boldsymbol{U}^\top$, and is probed through $m$ independent rank-one quadratic measurements of the form $\boldsymbol{a}_i^\top \boldsymbol{P} \boldsymbol{b}_i$, where $\boldsymbol{a}_i, \boldsymbol{b}_i \sim_{\text{i.i.d.}} \mathcal{N}(\boldsymbol{0}, \boldsymbol{I}_n)$. Stacking these measurements defines the rank-one projection (ROP) feature map $\psi^{\text{rop}}(\boldsymbol{U}) = (\boldsymbol{a}_i^\top \boldsymbol{U}\boldsymbol{U}^\top \boldsymbol{b}_i)_{i=1}^m$, whose empirical inner product is an estimator of the projection kernel but exhibits heavy-tailed concentration. To control this behaviour, we compose $\psi^{\text{rop}}$ with bounded non-linear functions. Using the sign function yields binary random features $\psi^\pm = \text{sign} \circ \psi^{\text{rop}}$, whose empirical kernel $\widehat{\kappa}^\pm(\boldsymbol{U}, \boldsymbol{V}) = \frac{1}{m}\langle \psi^\pm(\boldsymbol{U}), \psi^\pm(\boldsymbol{V})\rangle$ converges uniformly, with high probability, to a well-defined, positive-definite and rotationally invariant Grassmannian kernel $\kappa^\pm$ that depends only on the *principal angles* between two subspaces (Conway et al., 1996; Harandi et al., 2014). Although no explicit closed form is known for $\kappa^\pm$, we show that it can be expressed as a function of the related projectors $\boldsymbol{P} = \boldsymbol{U}\boldsymbol{U}^\top$ and $\boldsymbol{Q} = \boldsymbol{V}\boldsymbol{V}^\top$ with

$$\kappa^\pm(\boldsymbol{U}, \boldsymbol{V}) = 1 - \tfrac{2}{\pi}\mathbb{E}\angle(\boldsymbol{P}\boldsymbol{g}, \boldsymbol{Q}\boldsymbol{g}),$$

where $\angle(\boldsymbol{u}, \boldsymbol{v})$ is the angle between two vectors $\boldsymbol{u}$ and $\boldsymbol{v}$ and $\boldsymbol{g} \sim \mathcal{N}(\boldsymbol{0}, \boldsymbol{I}_n)$. This semi-explicit characterisation of $\kappa^\pm$ will allow us to get an intuitive understanding of the behaviour of $\kappa^\pm$ with respect to the principal angles between subspaces. Using instead the complex exponential defines periodic random features $\psi^\circ = \exp(\mathrm{i}\omega\,\psi^{\text{rop}})$, for which the expected kernel admits the closed form

$$\kappa^\circ(\boldsymbol{U}, \boldsymbol{V}) = \mathbb{E}\langle \psi^\circ(\boldsymbol{U}), \psi^\circ(\boldsymbol{V})\rangle = \prod_{j=1}^k \left(1 + \omega^2 \sin^2 \theta_j\right)^{-1},$$

where $\{\theta_j\}_{j=1}^k$ are the principal angles between $\boldsymbol{U}$ and $\boldsymbol{V}$. In both cases, we show that inner products of the random features provide unbiased estimators of the target kernels and satisfy uniform approximation bounds over $\mathbb{V}(k,n)$ (the set of orthonormal bases of dimension $k$ in $\mathbb{R}^n$) if $m = \mathcal{O}(kn)$ (up to log factors), yielding scalable Grassmannian kernel approximations with controlled error. This process is illustrated in Fig. 1. **[R-FSmh]** Note that these different feature maps each approximate a different kernel. Dense Gaussian projections and unbounded ROPs target the projection kernel $\kappa^{\text{p}}$, while the bounded binary and periodic ROP features induce the distinct Grassmannian kernels $\kappa^\pm$ and $\kappa^\circ$, respectively.

Beyond statistical guarantees, we evaluate the computational and memory efficiency of the proposed random feature constructions. While classical Grassmannian kernels require $\mathcal{O}(N^2)$ kernel evaluations and storage for an $N$-sample Gram matrix, and dense random embeddings of projectors incur $\mathcal{O}(n^2)$ cost per feature, the proposed bounded ROP-based features can be computed in $\mathcal{O}(kmn)$ operations and stored using $\mathcal{O}(mn)$ memory. Moreover, the uniform approximation results established in Secs. 4.1 and 4.2 show that $m = \mathcal{O}(kn)$ random features (up to logarithmic factors) suffice to control the approximation error, yielding overall complexities that scale linearly with the ambient dimension. We further show in Sec. 6 that these costs can

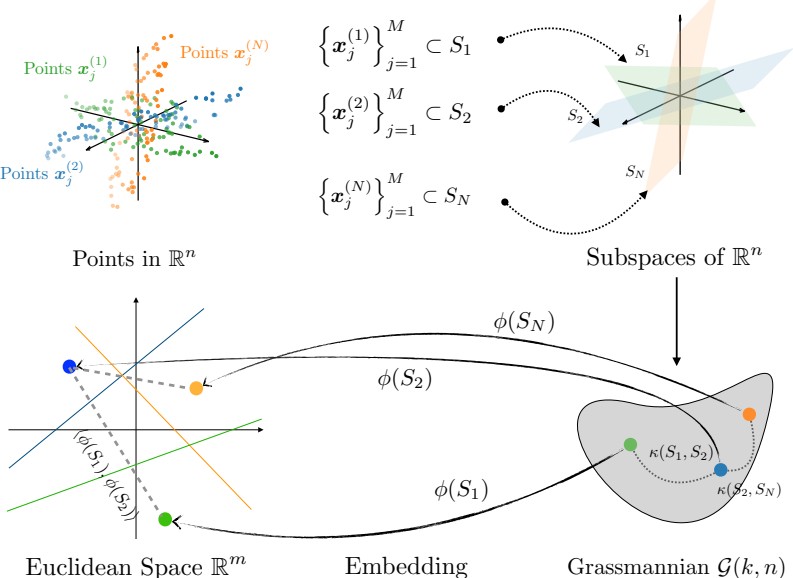

Figure 1: Schematic representation of the proposed method. Data points are assumed to belong to low-dimensional subspaces of $\mathbb{R}^n$. Subspaces are represented by their orthonormal bases, which are then embedded into a Euclidean space. Inner products in this space approximate well-defined Grassmannian kernels allowing for learning tasks to be performed efficiently.

be significantly reduced by replacing Gaussian probing vectors with structured random transforms inspired from (Choromanski et al., 2016), leading to feature maps computable in $\mathcal{O}(km \log n)$ operations with drastically reduced storage. A detailed comparison of computational and memory complexities for all considered methods is provided in Sec. 5, and the practical impact of these gains is illustrated experimentally in Sec. 8.

The rest of the paper is organised as follows. We start by providing the essential tools for navigating the geometry of $\mathcal{G}(k, n)$ as well as key reminders on random feature construction for kernel machines in Sec. 2. The limitations met by two direct designs of linear random features of Grassmannian elements are developed in Sec. 3. We provide in Sec. 4 two solutions to these limitations by composing linear rank-one projections with non-linear, bounded functions, *i.e.*, the sign and the periodic imaginary exponential functions. We provide uniform error guarantees on the possibility to approximate specific Grassmannian kernels with the corresponding binary and periodic random features. We dwell upon structured random feature schemes in Sec. 6, and then highlight how our approach connects with existing works, such as the lineage of subspace classification, in Sec. 7. We finally show in Sec. 8 how our approach allows for the classification of Grassmannian data, before concluding. We postpone to appendices all technical proofs that would otherwise slow the flow of reading.

**Notations and conventions** We find it useful to gather here some notations, concepts and conventions used throughout this paper. As we target global complexity analyses, many of the bounds developed in this work depend on constants, denoted by $C, c, c', c'', \ldots > 0$, whose values may vary from one line to another. We denote matrices and vectors with bold symbols, *e.g.*, $\boldsymbol{A} \in \mathbb{R}^{m \times n}$, $\boldsymbol{x} \in \mathbb{R}^n$, and scalar values with light symbols. The identity matrix in $\mathbb{R}^n$ is $\boldsymbol{I}_n$. The $m \times n$ zero matrix is denoted $\boldsymbol{0}_{m \times n}$, and we omit the dimensions when clear from the context. The angle between two vectors $\boldsymbol{u}$ and $\boldsymbol{v}$ reads $\angle(\boldsymbol{u}, \boldsymbol{v})$, and the $\ell_2$-norm of $\boldsymbol{u}$ is $\|\boldsymbol{u}\|$. The scalar product between two matrices in $\mathbb{R}^{n \times n}$ $\boldsymbol{A}$ and $\boldsymbol{B}$ is $\langle \boldsymbol{A}, \boldsymbol{B} \rangle = \mathrm{tr}(\boldsymbol{A}^\top \boldsymbol{B}) = \langle \mathrm{vec}(\boldsymbol{A}), \mathrm{vec}\,\boldsymbol{B} \rangle$, where $\mathrm{vec}\,\boldsymbol{A} \in \mathbb{R}^{n^2}$ denotes the vectorisation of $\boldsymbol{A}$. The related Frobenius norm reads $\|\boldsymbol{A}\|_F^2 = \langle \boldsymbol{A}, \boldsymbol{A} \rangle$. The cardinality of a finite set $\mathcal{S}$ is denoted $|\mathcal{S}|$. The standard normal and Rademacher $\pm 1$ distributions read $\mathcal{N}(0, 1)$ and $\mathcal{U}(\pm 1)$, respectively, and the multivariate normal distribution in $\mathbb{R}^n$ (with identity covariance) is $\mathcal{N}(\boldsymbol{0}, \boldsymbol{I}_n)$. We use the shorthand i.i.d. to define *identically and independently distributed* random quantities (variables, vectors or matrices). The groups of orthogonal matrices and rotations in $\mathbb{R}^n$ are denoted by

$\mathsf{O}(n)$ and $\mathsf{SO}(n)$, respectively, while $\mathcal{O}$ is reserved for the complexity symbol. Finally, as this work considers several types *embeddings*, we reserve the symbol $\phi$ for all deterministic embeddings, *e.g.*, those used in the kernel trick, and the symbol $\psi$ for all the random embeddings used to define random features.

## 2 Background and preliminaries

This section provides the tools needed to navigate the Grassmannian manifold $\mathcal{G}(k, n)$ as well as a brief introduction to random features for kernel machines in supervised learning.

### 2.1 The Grassmannian manifold

Mathematically, the Grassmannian manifold $\mathcal{G}(k, n)$ represents the set of $k$-dimensional subspaces of $\mathbb{R}^n$. For example in the case where $k = 1$ and $n = 3$, $\mathcal{G}(1, 3)$ contains all the lines through the origin in a three dimensional space.

Each subspace $\mathcal{P} \in \mathcal{G}(k, n)$ can be represented as the span of an *orthonormal basis* $\boldsymbol{U} \in \mathbb{R}^{n \times k}$, *i.e.*, $\mathcal{P} = \mathrm{span}(\boldsymbol{U})$, where $\boldsymbol{U}$ is a matrix with orthonormal columns, *i.e.*, it belongs to the Stiefel manifold $\mathbb{V}(k, n)$

$$\mathbb{V}(k, n) := \{\boldsymbol{V} \in \mathbb{R}^{n \times k} : \boldsymbol{V}^\top \boldsymbol{V} = \boldsymbol{I}_k\}.$$

If the subspace is provided through a collection of vectors $\{\boldsymbol{x}_j\}_{j=1}^N \subset \mathbb{R}^n$ such that the rank of the data matrix $\boldsymbol{X} = (\boldsymbol{x}_1, \ldots, \boldsymbol{x}_N) \in \mathbb{R}^{n \times N}$ equals $k$, then $\boldsymbol{U}$ can be obtained, for instance, from the QR or the SVD factorisation of $\boldsymbol{X}$.

A subspace $\mathcal{P}$ is not uniquely represented by a basis $\boldsymbol{U}$. In fact, any basis $\boldsymbol{U}' = \boldsymbol{U}\boldsymbol{R}$ obtained by transforming the columns of $\boldsymbol{U}$ with an orthogonal matrix $\boldsymbol{R} \in \mathsf{SO}(k)$ is also a basis of the subspace. However, each subspace of $\mathcal{G}(k, n)$ can be represented by the *unique* projector $\boldsymbol{P} = \boldsymbol{U}\boldsymbol{U}^\top$ such that $\boldsymbol{P}^2 = \boldsymbol{P} = \boldsymbol{P}^\top$ and rank $\boldsymbol{P} = k$, and we define

$$\mathbb{G}(k, n) = \{\boldsymbol{P} \in \mathbb{R}^{n \times n} : \boldsymbol{P}^2 = \boldsymbol{P} = \boldsymbol{P}^\top, \mathrm{rank}\,\boldsymbol{P} = k\},$$

which is the projector representation of $\mathcal{G}(k, n)$. This representation of $\mathcal{G}(k, n)$ is invariant to the choice of basis: rotating the columns of $\boldsymbol{U}$ leaves $\boldsymbol{P}$ unchanged, since $\boldsymbol{U}\boldsymbol{R}(\boldsymbol{U}\boldsymbol{R})^\top = \boldsymbol{U}\boldsymbol{U}^\top$ for any orthogonal matrix $\boldsymbol{R}$. In this work, we often identify a subspace $\mathcal{P} \in \mathcal{G}(k, n)$ with its projector $\boldsymbol{P} \in \mathbb{G}(k, n)$.

An alternative but equivalent characterisation views $\mathcal{G}(k, n)$ as a homogeneous space $\mathsf{O}(n)/(\mathsf{O}(k) \times \mathsf{O}(n - k))$, identifying each subspace with a set of orthogonal transformations mapping a fixed reference plane onto it. While this *Lie-group formulation* offers valuable geometric insight, it will not be needed in what follows.

Equipped with these representations, we can now define notions of distance and similarity between subspaces, which will serve as the basis for learning methods on $\mathcal{G}(k, n)$.

### 2.2 Grassmannian distances

Distances between elements of $\mathcal{G}(k, n)$ may be derived from their *principal angles*. Intuitively, in $\mathcal{G}(1, n)$, the set of lines through the origin, comparing two lines amounts to looking at the angle they form. In higher dimensions, two $k$-dimensional subspaces $\mathcal{P}, \mathcal{Q} \in \mathcal{G}(k, n)$ form $k$ principal angles $0 \leqslant \theta_1, \ldots, \theta_k \leqslant \pi/2$.

[R-XJL8] The principal angles between two subspaces $\mathcal{P}, \mathcal{Q} \in \mathcal{G}(k, n)$ measure how aligned these subspaces are. Given two orthonormal bases $\boldsymbol{U}, \boldsymbol{V} \in \mathbb{V}(k, n)$ of $\mathcal{P}$ and $\mathcal{Q}$, respectively, these angles $\theta_1, \ldots, \theta_k \in [0, \pi/2]$ are defined by

$$\cos(\theta_i) = \sigma_i(\boldsymbol{U}^\top \boldsymbol{V}), \quad i = 1, \ldots, k,$$

where $\sigma_1(\boldsymbol{U}^\top \boldsymbol{V}) \geqslant \cdots \geqslant \sigma_k(\boldsymbol{U}^\top \boldsymbol{V})$ are the singular values of $\boldsymbol{U}^\top \boldsymbol{V}$. Equivalently, the principal angles can be obtained recursively by looking for pairs of unit vectors in $\mathcal{P}$ and $\mathcal{Q}$ with maximal inner product, under orthogonality constraints with the previously selected pairs. If $\mathcal{P}$ and $\mathcal{Q}$ have dimensions $k$ and $k'$ with $k \neq k'$, then $\min(k, k')$ principal angles can be defined in the same way (Mandolesi, 2021).

Among the existing distances on $\mathcal{G}(k,n)$, two are most relevant to our work. The *projection distance*,

$$d_{\mathrm{P}}(\boldsymbol{P},\boldsymbol{Q}) = \big(\textstyle\sum_{i=1}^{k}\sin^2\theta_i\big)^{1/2} = \tfrac{1}{\sqrt{2}}\|\boldsymbol{P}-\boldsymbol{Q}\|_F,$$

which amounts to the distance of the projectors $\boldsymbol{P}$ and $\boldsymbol{Q}$, and the *Binet-Cauchy distance*,

$$d_{\mathrm{BC}}(\boldsymbol{P},\boldsymbol{Q}) = \big(1 - \textstyle\prod_{i=1}^{k}\cos^2\theta_i\big)^{1/2} = \big(1 - \mathrm{pdet}(\boldsymbol{P}\boldsymbol{Q})\big)^{1/2},$$

where the *pseudo-determinant* $\mathrm{pdet}(\boldsymbol{M})$ of any rank-$r$ matrix $\boldsymbol{M}$ is defined by (Florescu, 2014, p. 529)

$$\mathrm{pdet}(\boldsymbol{M}) = \lim_{t\to 0} t^{r-n}\det(\boldsymbol{M}+t\boldsymbol{I}_n), \tag{1}$$

with $\mathrm{pdet}(\boldsymbol{P}\boldsymbol{Q}) = \det(\boldsymbol{U}^\top\boldsymbol{V})^2$ if $\boldsymbol{P} = \boldsymbol{U}\boldsymbol{U}^\top$ and $\boldsymbol{Q} = \boldsymbol{V}\boldsymbol{V}^\top$.

The distance $d_{\mathrm{BC}}$ quantifies how the squared volume of the basis $\boldsymbol{V}$ projected onto $\boldsymbol{U}$ (as measured by $\det(\boldsymbol{U}^\top\boldsymbol{V})^2$) deviates from the (unit) value it takes when $\boldsymbol{P} = \boldsymbol{Q}$. We can also mention the *geodesic distance*, *i.e.*, the length of the shortest curve in $\mathcal{G}(k,n)$ connecting two subspaces, which is the most natural notion of distance on the manifold

$$d_{\mathrm{G}}(\boldsymbol{P},\boldsymbol{Q}) = \big(\textstyle\sum_{i=1}^{k}\theta_i^2\big)^{1/2}.$$

These distances are crucial to compare elements of $\mathcal{G}(k,n)$ and will later be used to define similarity measures and kernels between subspaces.

## 2.3 Grassmannian kernels

We are interested in solving problems on $\mathcal{G}(k,n)$ by embedding elements into a Hilbert space using kernel functions. Such kernel functions must respect specific criteria as explained in Schölkopf & Smola (2002) in the case of general kernels and in Harandi et al. (2014) for Grassmannian kernels.

First, the kernel must be *positive-semidefinite* to allow its use in kernel machines.

**Definition 1** (Positive-semidefinite kernel). *Let $\mathcal{X}$ be a non-empty set. A symmetric function $\kappa : \mathcal{X} \times \mathcal{X} \to \mathbb{R}$ is called* positive-semidefinite *if, for any $N \in \mathbb{N}$, any choice of points $x_1,\ldots,x_N \in \mathcal{X}$, and any real coefficients $c_1,\ldots,c_N$, we have $\sum_{i=1}^{N}\sum_{j=1}^{N} c_i c_j \kappa(x_i, x_j) \geqslant 0$.*

Second, as we target kernels defined over subspace bases, they must be independent of the orthonormal bases of $\mathbb{V}(k,n)$ used to represent the compared subspaces of $\mathcal{G}(k,n)$.

**Definition 2** (Well-defined Grassmannian kernel). *A well-defined Grassmannian kernel is a positive-definite function $\kappa : \mathbb{V}(k,n) \times \mathbb{V}(k,n) \mapsto \mathbb{R}$ that measures similarity between subspaces of $\mathcal{G}(k,n)$ independently of the bases chosen to represent them. Formally, given two orthonormal bases $\boldsymbol{U},\boldsymbol{V} \in \mathbb{V}(k,n)$, $\kappa$ satisfies*

$$\kappa(\boldsymbol{U},\boldsymbol{V}) = \kappa(\boldsymbol{U}\boldsymbol{R},\boldsymbol{V}\boldsymbol{R}'), \quad \forall \boldsymbol{R},\boldsymbol{R}' \in \mathsf{SO}(k),$$

*where $\mathsf{SO}(k)$ denotes the special orthogonal group in $\mathbb{R}^k$.*

Finally, the kernel must not change if both compared subspaces are similarly rotated in $\mathbb{R}^n$ (Wong, 1967).

**Definition 3** (Rotational invariance). *A Grassmannian kernel $\kappa : \mathbb{V}(k,n) \times \mathbb{V}(k,n) \mapsto \mathbb{R}$ is rotationally invariant if, given any two bases $\boldsymbol{U},\boldsymbol{V} \in \mathbb{V}(k,n)$ of the subspaces $\mathcal{P},\mathcal{Q} \in \mathcal{G}(k,n)$,*

$$\kappa(\boldsymbol{U},\boldsymbol{V}) = \kappa(\boldsymbol{R}\boldsymbol{U},\boldsymbol{R}\boldsymbol{V}), \ \forall \boldsymbol{R} \in \mathsf{SO}(n).$$

A direct consequence of the rotational invariance is the following important fact.

**Theorem 4.** *A rotationally invariant Grassmannian kernel $\kappa : \mathbb{V}(k,n) \times \mathbb{V}(k,n) \mapsto \mathbb{R}$ only depends on the principal angles of the compared subspaces, i.e., given two bases $\boldsymbol{U},\boldsymbol{V} \in \mathbb{V}(k,n)$ of the subspaces $\mathcal{P},\mathcal{Q} \in \mathcal{G}(k,n)$ and their $k$ principal angles $\{\theta_i\}_{i=1}^{k}$, there exists a function $f$ such that $\kappa(\boldsymbol{U},\boldsymbol{V}) = f(\theta_1,\ldots,\theta_k)$.*

This theorem is a direct consequence of (Wong, 1967, Thm. 3) (see also (Conway et al., 1996)).

Two examples of well-defined, positive-semidefinite, rotationally invariant kernels on $\mathcal{G}(k,n)$ are the projection kernel $\kappa^{\mathrm{p}}$ (Hamm & Lee, 2008) and the Binet-Cauchy kernel $\kappa^{\mathrm{BC}}$ (Wolf & Shashua, 2003). Given two subspaces $\mathcal{P}, \mathcal{Q} \in \mathcal{G}(k,n)$ related to projectors $\boldsymbol{P}, \boldsymbol{Q}$ and bases $\boldsymbol{U}, \boldsymbol{V} \in \mathbb{V}(k,n)$, respectively, both kernels relate to their homonymous distances:

$$\kappa^{\mathrm{p}}(\boldsymbol{U}, \boldsymbol{V}) := \|\boldsymbol{U}^\top \boldsymbol{V}\|_F^2 = k - d_{\mathrm{P}}(\boldsymbol{P}, \boldsymbol{Q})^2, \tag{2}$$

$$\kappa^{\mathrm{BC}}(\boldsymbol{U}, \boldsymbol{V}) := \det(\boldsymbol{U}^\top \boldsymbol{V})^2 = 1 - d_{\mathrm{BC}}(\boldsymbol{P}, \boldsymbol{Q})^2. \tag{3}$$

Both kernels are linked to a classical embedding of the Grassmannian manifold allowing us to prove that they are *positive-semidefinite*, *i.e.*, there exists an embedding $\phi : \mathbb{V}(k,n) \to \mathbb{R}^d$, with a possibly very large dimension $d$, such that the considered kernel $\kappa$ can be recast as $\kappa(\boldsymbol{U}, \boldsymbol{V}) = \langle \phi(\boldsymbol{U}), \phi(\boldsymbol{V}) \rangle$. In particular, the projection kernel arises as the Frobenius inner product between orthogonal projectors, corresponding to the *projection embedding*

$$\phi^{\mathrm{p}} : \boldsymbol{U} \in \mathbb{V}(k,n) \mapsto \phi^{\mathrm{p}}(\boldsymbol{U}) = \boldsymbol{U}\boldsymbol{U}^\top \in \mathbb{G}(k,n), \tag{4}$$

with $d = n^2$. Similarly, the Binet-Cauchy kernel is given by inner product of the *Plücker embedding* which represents each basis of $\mathbb{V}(k,n)$ by its $d = \binom{n}{k}$ possible $k \times k$ minors (Harandi et al., 2014). Such an embedding is never constructed explicitly in practice due to its high computational costs.

Let us also mention that beyond these two standard kernels, a larger family of positive-semidefinite, rotationally invariant Grassmannian kernels can be constructed by composing classical Euclidean kernels with the projection or Plücker embeddings. In particular, Harandi et al. (2014) show that applying radial basis function (RBF), Laplace, polynomial or binomial kernels to those embeddings yields well-defined Grassmannian kernels. A great example is the projection RBF kernel

$$\kappa_{\mathrm{RBF}}(\boldsymbol{U}, \boldsymbol{V}) = \exp\left(-\gamma \|\boldsymbol{P} - \boldsymbol{Q}\|_F^2\right), \text{ with } \gamma > 0, \tag{5}$$

which depends only on the Frobenius distance between projectors and is positive-semidefinite on $\mathcal{G}(k,n)$. As we show in Sec. 2.3, the periodic kernel $\kappa^\circ$ introduced in this work recovers this projection RBF kernel in the small-frequency regime, thereby providing a random-feature approximation that naturally interpolates between linear and RBF-type Grassmannian similarities.

In supervised learning with kernel machines, that is, when one must learn a given kernel model from $N$ instances $\{\boldsymbol{U}_i\}_{i=1}^N \subset \mathbb{V}(k,n)$ of pre-computed bases of subspaces $\{\mathcal{P}_i\}_{i=1}^N \subset \mathcal{G}(k,n)$ (*e.g.*, from an initial dataset of vectors), an $N \times N$ Gram matrix $\boldsymbol{K}$ with entries $K_{ij} := \kappa(\boldsymbol{U}_i, \boldsymbol{U}_j)$ must be constructed to allow learning the model parameters. This involves a memory complexity of $\mathcal{O}(N^2)$ as well as $\mathcal{O}(N^2)$ calls of the kernel function. However, from the closed form kernels $\kappa^{\mathrm{p}}$ and $\kappa^{\mathrm{BC}}$, each call has a respective computational complexity of $\mathcal{O}(k^2 n)$, since $\boldsymbol{U}^\top \boldsymbol{V} = (\boldsymbol{u}_i^\top \boldsymbol{v}_j)_{i,j=1}^k \in \mathbb{R}^{k \times k}$ involves $k^2$ inner products in $\mathbb{R}^n$, and $\mathcal{O}(k^2 n + k^3)$ since computing the determinant of a $k \times k$ matrix has complexity $\mathcal{O}(k^3)$. The total complexity of computing the Gram matrix is thus of $\mathcal{O}(k^2 n N^2)$ and $\mathcal{O}((k^2 n + k^3) N^2)$ for the projection and the BC kernels, respectively. For large values of $N$, computing and storing the Gram matrix can thus be challenging.

## 2.4 Random features for kernel machines

In supervised machine learning with kernel machines (such as kernel SVM or kernel regression (Bach, 2024)), a well-established strategy to circumvent the issue of storing and computing full Gram matrices is the use of *random features* (RF), introduced in the seminal work of Rahimi & Recht (2007).

Given a $d$-dimensional data space $\mathcal{X} \subset \mathbb{R}^d$, these methods involve building an explicit, $m$-dimensional random embedding

$$\psi : \boldsymbol{x} \in \mathcal{X} \mapsto \psi(\boldsymbol{x}) \in \mathbb{R}^m,$$

or *feature map*, such that, for any two samples $\boldsymbol{x}, \boldsymbol{x}' \in \mathcal{X}$, the (scaled) inner product (or empirical kernel) $\hat{\kappa}(\boldsymbol{x}, \boldsymbol{x}') := \frac{1}{m} \langle \psi(\boldsymbol{x}), \psi(\boldsymbol{x}') \rangle$ is an *unbiased estimator* of a specific kernel function $\kappa(\boldsymbol{x}, \boldsymbol{x}')$, *i.e.*,

$$\mathbb{E}\hat{\kappa}(\boldsymbol{x}, \boldsymbol{x}') = \frac{1}{m} \mathbb{E} \langle \psi(\boldsymbol{x}), \psi(\boldsymbol{x}') \rangle = \kappa(\boldsymbol{x}, \boldsymbol{x}').$$

For instance, the *random Fourier features*, defined by $\psi(\boldsymbol{x}) = (\exp(\mathrm{i}\boldsymbol{\omega}_j^\top \boldsymbol{x}))_{j=1}^m$, allows one to estimate, thanks to Bochner's theorem, the Gaussian or Laplacian kernels if the $m$ *frequencies* (or *probes*) $\{\boldsymbol{\omega}_j\}_{j=1}^m \subset \mathbb{R}^d$ are drawn i.i.d. from a Gaussian or Cauchy distribution, respectively (Rahimi & Recht, 2007; Boufounos et al., 2017; Liu et al., 2022).

Once a random feature map is available, the learning algorithm can operate directly, with bounded errors, in the labelled random feature space $\{(\psi(\boldsymbol{x}_i), \boldsymbol{y}_i)\}_{i=1}^N \subset \mathbb{R}^m \times \mathcal{Y}$ (for some label space $\mathcal{Y}$) of an $N$-sample labelled dataset $\{(\boldsymbol{x}_i, \boldsymbol{y}_i)\}_{i=1}^N \subset \mathcal{X} \times \mathcal{Y}$. This is possible if we can bound the deviation between the inner product $\langle \psi(\boldsymbol{x}), \psi(\boldsymbol{x}') \rangle$ and the kernel $\kappa(\boldsymbol{x}, \boldsymbol{x}')$, (*e.g.*, with variance analysis or measure concentration tools (Vershynin, 2018)) either on a given sample pair or *uniformly* over all of them in (a subset of) $\mathcal{X} \times \mathcal{X}$. Consequently, random features turn non-linear kernel machines into linear learning methods eliminating the need to store an $N \times N$ Gram matrix.

In practice, this shifts the cost from storing the full Gram matrix to storing the $m$-dimensional random features of each instance instead of their raw representation. One must, however, store the $m$ random "probes" (*i.e.*, vectors or matrices) used to compute $\psi$ to compute it on any new data, *i.e.*, at inference time. The challenge addressed in the next section is therefore to design fast random feature maps with reduced storage, while preserving the Grassmannian geometry.

## 3 Linear random features over the Grassmannian

In this section, we present two linear random feature maps over subspaces of $\mathcal{G}(k,n)$, that enable us to approximate the projection kernel $\kappa^{\mathrm{p}}$ between two subspaces (see Sec. 2.3). These two maps are inspired by the compressive sensing of structured matrices (Candès & Plan, 2011; Foucart, 2016) and the context of compressive covariance estimation from quadratic sampling (Chen et al., 2015). As explained below, while each of them has clear limitations, they define a benchmark from which we develop new random features in Sec. 4.

### 3.1 Dense random projections

One can first embed each $k$-dimensional subspace $\mathcal{P} \in \mathcal{G}(k,n)$ by projecting its projector $\boldsymbol{P} \in \mathbb{G}(k,n)$ onto *dense* unstructured $n \times n$ random matrices. If $\mathcal{P}$ is known through a basis $\boldsymbol{U} \in \mathbb{V}(k,n)$, then projecting the well-defined $\boldsymbol{P} = \phi^{\mathrm{p}}(\boldsymbol{U}) = \boldsymbol{U}\boldsymbol{U}^\top$ (as well as any other function of $\boldsymbol{P}$) respects the Grassmannian geometry. Formally, this random projection reads

$$\Psi : \quad \boldsymbol{X} \in \mathbb{R}^{n \times n} \quad \mapsto \quad \Psi(\boldsymbol{X}) = (\langle \boldsymbol{\Psi}_i, \boldsymbol{X} \rangle)_{i=1}^m \in \mathbb{R}^m,$$

where the $n \times n$ probing matrices $\boldsymbol{\Psi}_i$ have Gaussian entries i.i.d. as $\mathcal{N}(0,1)$. Embedding elements of $\mathbb{V}(k,n)$ can then be done by composing $\Psi$ with the projection embedding $\phi^{\mathrm{p}}$, *i.e.*, defining the *dense* random feature map

$$\psi^{\mathrm{d}} = \Psi \circ \phi^{\mathrm{p}} : \boldsymbol{U} \in \mathbb{V}(k,n) \mapsto \Psi(\phi^{\mathrm{p}}(\boldsymbol{U})) \in \mathbb{R}^m. \tag{6}$$

Interestingly, when we consider two bases of two $k$-dimensional subspaces $\mathcal{P}$ and $\mathcal{Q}$, inner products in the space mapped by $\psi^{\mathrm{d}}$ are unbiased estimators of the projection kernel $\kappa^{\mathrm{p}}$. Indeed, if $\boldsymbol{U}$ and $\boldsymbol{V}$ are their respective $n \times k$ orthonormal bases, then, defining the empirical kernel $\widehat{\kappa}^{\mathrm{d}}(\boldsymbol{U}, \boldsymbol{V}) := \frac{1}{m}\langle \psi^{\mathrm{d}}(\boldsymbol{U}), \psi^{\mathrm{d}}(\boldsymbol{V}) \rangle$,

$$\mathbb{E}\,\widehat{\kappa}^{\mathrm{d}}(\boldsymbol{U}, \boldsymbol{V}) = \tfrac{1}{m}\sum_{i=1}^m \mathbb{E}\langle \phi^{\mathrm{p}}(\boldsymbol{U}), \boldsymbol{\Psi}_i \rangle\langle \boldsymbol{\Psi}_i, \phi^{\mathrm{p}}(\boldsymbol{V}) \rangle = \langle \phi^{\mathrm{p}}(\boldsymbol{U}), \phi^{\mathrm{p}}(\boldsymbol{V}) \rangle = \kappa^{\mathrm{p}}(\boldsymbol{U}, \boldsymbol{V}),$$

since $\mathbb{E}\,\mathrm{vec}(\boldsymbol{\Psi}_i)\,\mathrm{vec}(\boldsymbol{\Psi}_i)^\top = \boldsymbol{I}_{n^2}$ by design.

One can also study how these inner products deviate from the projection kernel by invoking the compressive sensing literature (Foucart, 2016). The mapping $\Psi$ is indeed known to embed low-rank matrices into $\mathbb{R}^m$ with a controlled distortion on their Frobenius norms (Candès & Plan, 2011). Given some distortion level $0 < \delta < 1$ and provided $m = \mathcal{O}(\delta^{-2}kn)$ (up to log factors), the mapping $\Psi$ respects, with high probability, the *restricted isometry property*, or $\mathrm{RIP}(\delta, \mathcal{R}(k,n))$, over the set $\mathcal{R}(k,n)$ of $n \times n$ rank-$k$ matrices:

$$(1-\delta)\|\boldsymbol{X}\|_F^2 \leqslant \tfrac{1}{m}\|\Psi(\boldsymbol{X})\|_2^2 \leqslant (1+\delta)\|\boldsymbol{X}\|_F^2, \quad \forall \boldsymbol{X} \in \mathcal{R}(k,n). \tag{RIP}$$

The RIP allows us to reach a uniform approximation of the projection kernel $\kappa^{\mathrm{p}}$ through $\widehat{\kappa}^{\mathrm{d}}$ with distortion $\delta k$, as expressed in the following proposition.

**Proposition 5** (RIP $\Rightarrow$ uniform kernel approximation on $\mathcal{G}(k,n)$). *If the mapping $\Psi : \mathbb{R}^{n \times n} \to \mathbb{R}^m$ respects the RIP$(\delta, \mathcal{R}(2k,n))$ for some distortion $0 < \delta < 1$, then,*

$$|\widehat{\kappa}^{\mathrm{d}}(\boldsymbol{U},\boldsymbol{V}) - \kappa^{\mathrm{p}}(\boldsymbol{U},\boldsymbol{V})| \leqslant \delta k, \quad \forall \boldsymbol{U}, \boldsymbol{V} \in \mathbb{V}(k,n).$$

*Proof.* Writing $\boldsymbol{P} = \boldsymbol{U}\boldsymbol{U}^\top$ and $\boldsymbol{Q} = \boldsymbol{V}\boldsymbol{V}^\top$, by the polarisation identity, $\kappa^{\mathrm{p}}(\boldsymbol{U},\boldsymbol{V}) = \langle \boldsymbol{P}, \boldsymbol{Q} \rangle = \frac{1}{4}(\|\boldsymbol{P} + \boldsymbol{Q}\|_F^2 - \|\boldsymbol{P} - \boldsymbol{Q}\|_F^2)$ and $\widehat{\kappa}^{\mathrm{d}}(\boldsymbol{U},\boldsymbol{V}) = \frac{1}{4m}(\|\Psi(\boldsymbol{P}+\boldsymbol{Q})\|_2^2 - \|\Psi(\boldsymbol{P}-\boldsymbol{Q})\|_2^2)$. Since $\Psi$ respects the RIP over $\mathcal{R}(2k,n)$, a set which includes the matrices $\boldsymbol{P} \pm \boldsymbol{Q}$, we get $\widehat{\kappa}^{\mathrm{d}}(\boldsymbol{U},\boldsymbol{V}) \leqslant \langle \boldsymbol{P}, \boldsymbol{Q} \rangle + \frac{1}{4}(\|\boldsymbol{P}+\boldsymbol{Q}\|^2 + \|\boldsymbol{P}-\boldsymbol{Q}\|^2) \leqslant \langle \boldsymbol{P}, \boldsymbol{Q} \rangle + k\delta$, since $\|\boldsymbol{P}\|_F^2 = \|\boldsymbol{Q}\|_F^2 = k$ and $\|\boldsymbol{P} + \boldsymbol{Q}\|_F^2 + \|\boldsymbol{P} - \boldsymbol{Q}\|_F^2 = 4k$. We obtain the lower bound similarly. $\square$

Having an approximation error bounded by $\delta k$ instead of $\delta$ in Prop. 5 is a severe limitation. From a rescaling argument, if we rather target, with high probability, the uniform error

$$|\widehat{\kappa}^{\mathrm{d}}(\boldsymbol{U},\boldsymbol{V}) - \kappa^{\mathrm{p}}(\boldsymbol{U},\boldsymbol{V})| < \delta, \quad \forall \boldsymbol{U}, \boldsymbol{V} \in \mathbb{V}(k,n),$$

then $\Psi$ must respect the RIP$(\delta/k, \mathcal{R}(2k,n))$, which thus happens with high probability provided $m = \mathcal{O}(\delta^{-2}k^3 n)$. Consequently, the resulting feature map $\psi^{\mathrm{d}}$ is not scalable in practice. First, each matrix $\boldsymbol{\Psi}_i \in \mathbb{R}^{n \times n}$ contains $n^2$ entries, globally requiring the storage of $\mathcal{O}(mn^2)$ real numbers. Second, evaluating each feature $\langle \boldsymbol{\Psi}_i, \phi^{\mathrm{p}}(\boldsymbol{U}) \rangle$ in (6) costs $\mathcal{O}(n^2)$ operations, so also a total of $\mathcal{O}(mn^2)$ operations for the $m$ projections.

Consequently, evaluating $\psi^{\mathrm{d}}$ with $m = \mathcal{O}(\delta^{-2}k^3 n)$ components to estimate $\kappa^{\mathrm{p}}$ through $\hat{\kappa}$ has memory and computational complexities of $\mathcal{O}(\delta^{-2}k^3 n^3)$, which can be worse than a single evaluation of both $\kappa^{\mathrm{p}}$ and $\kappa^{\mathrm{BC}}$ for small values of $k < n$.

### 3.2 Lighter random features with rank-one projections

There exists another random feature map with reduced computational cost and storage compared to dense random projections and that respects the Grassmannian geometry. Given a subspace $\mathcal{P}$ of $\mathcal{G}(k,n)$ with basis $\boldsymbol{U} \in \mathbb{V}(k,n)$, another random feature map $\psi^{\mathrm{rop}}$, named *rank-one projections* (ROP) (Chen et al., 2015; Cai & Zhang, 2015; Delogne et al., 2023), can be built from a collection of rank-one random matrices $\boldsymbol{A}_i = \boldsymbol{a}_i \boldsymbol{b}_i^\top$ defined with $m$ Gaussian random vectors $\boldsymbol{a}_i, \boldsymbol{b}_i \sim \mathcal{N}(\boldsymbol{0}, \boldsymbol{I}_n)$, *i.e.*,

$$\psi^{\mathrm{rop}} : \quad \boldsymbol{U} \in \mathbb{R}^{n \times k} \quad \mapsto \quad \psi^{\mathrm{rop}}(\boldsymbol{U}) = (\langle \boldsymbol{A}_i, \phi^{\mathrm{p}}(\boldsymbol{U}) \rangle = \boldsymbol{a}_i^\top \boldsymbol{U}\boldsymbol{U}^\top \boldsymbol{b}_i)_{i=1}^m. \tag{7}$$

Crucially, as for $\Psi$, the feature map $\psi^{\mathrm{rop}}$, which is defined over $\phi^{\mathrm{p}}$, is invariant to the choice of basis $\boldsymbol{U}$; it depends only on the subspace $\mathcal{P}$. Moreover, one can compute the components of $\psi^{\mathrm{rop}}(\boldsymbol{U})$ with complexity $\mathcal{O}(kmn)$ by computing all the $k$-dimensional vectors $\boldsymbol{\alpha}_i := \boldsymbol{U}^\top \boldsymbol{a}_i$ and $\boldsymbol{\beta}_i := \boldsymbol{U}^\top \boldsymbol{b}_i$, $1 \leqslant i \leqslant m$, with storage and computational complexities $\mathcal{O}(kmn)$, and then evaluating all the inner products $\psi_i^{\mathrm{rop}}(\boldsymbol{U}) = \boldsymbol{\alpha}_i^\top \boldsymbol{\beta}_i$ in $\mathcal{O}(km)$ computations. This makes ROP both geometrically sound and computationally practical.

Interestingly, like for $\Psi$ again, given two $k$-dimensional subspaces $\mathcal{P}, \mathcal{Q} \in \mathcal{G}(k,n)$ with bases $\boldsymbol{U}, \boldsymbol{V} \in \mathbb{V}(k,n)$, the empirical kernel

$$\widehat{\kappa}^{\mathrm{rop}}(\boldsymbol{U},\boldsymbol{V}) := \tfrac{1}{m}\langle \psi^{\mathrm{rop}}(\boldsymbol{U}), \psi^{\mathrm{rop}}(\boldsymbol{V}) \rangle, \tag{8}$$

is an unbiased estimator of the projection kernel $\kappa^{\mathrm{p}}(\boldsymbol{U},\boldsymbol{V})$. Indeed, we observe that

$$\begin{aligned} \tfrac{1}{m}\mathbb{E}\langle \psi^{\mathrm{rop}}(\boldsymbol{U}), \psi^{\mathrm{rop}}(\boldsymbol{V}) \rangle &= \tfrac{1}{m}\sum_{i=1}^m \mathbb{E}\boldsymbol{a}_i^\top \phi^{\mathrm{p}}(\boldsymbol{U})\boldsymbol{b}_i \boldsymbol{b}_i^\top \phi^{\mathrm{p}}(\boldsymbol{V})\boldsymbol{a}_i = \tfrac{1}{m}\sum_{i=1}^m \mathbb{E}(\boldsymbol{a}_i^\top \phi^{\mathrm{p}}(\boldsymbol{U})\phi^{\mathrm{p}}(\boldsymbol{V})\boldsymbol{a}_i) \\ &= \tfrac{1}{m}\sum_{i=1}^m \operatorname{tr}(\boldsymbol{U}\boldsymbol{U}^\top \boldsymbol{V}\boldsymbol{V}^\top) = \|\boldsymbol{U}^\top \boldsymbol{V}\|_F^2 = \kappa^{\mathrm{p}}(\boldsymbol{U},\boldsymbol{V}), \end{aligned}$$

where we used the cyclic property of the trace.

**[R-FSmh]** Although $\widehat{\kappa}^{\mathrm{rop}}$ estimates $\kappa^{\mathrm{p}}$ on average, its concentration behaviour is less favourable than the one obtained from bounded random features. Indeed, each term of the sum composing $m\widehat{\kappa}^{\mathrm{rop}}$ has the same distribution as the random variable

$$Z := (\boldsymbol{a}^\top \boldsymbol{U}\boldsymbol{U}^\top \boldsymbol{b})(\boldsymbol{a}^\top \boldsymbol{V}\boldsymbol{V}^\top \boldsymbol{b}) = \boldsymbol{\alpha}^\top \boldsymbol{\beta}\boldsymbol{\alpha}'^\top \boldsymbol{\beta}',$$

where $\boldsymbol{a}, \boldsymbol{b} \sim_{\text{i.i.d.}} \mathcal{N}(\boldsymbol{0}, \boldsymbol{I}_n)$ gives the projections $\boldsymbol{\alpha} := \boldsymbol{U}^\top \boldsymbol{a}$ and $\boldsymbol{\beta} := \boldsymbol{U}^\top \boldsymbol{b}$ onto $\boldsymbol{U}$, and similarly for the projections $\boldsymbol{\alpha}'$ and $\boldsymbol{\beta}'$ onto $\boldsymbol{V}$. This is mainly the product of four correlated Gaussian random variables. The random variable $Z$ has thus distribution tails that are heavier than the sub-exponential random variables (Vershynin, 2018). As highlighted in Bong & Kuchibhotla (2023), this places our estimator in a sub-Weibull (with parameter $1/2$) regime, where deviations are still controlled, but they decay much more slowly than the exponential-type rates enjoyed by sub-exponential random variables.

[**R-FSmh**] This distinction is especially important when going from a fixed pair of subspaces to a uniform guarantee over the full Grassmannian. For any fixed pair $\mathcal{P}, \mathcal{Q}$ with bases $\boldsymbol{U}, \boldsymbol{V} \in \mathbb{V}(k, n)$, the empirical kernel $\widehat{\kappa}^{\text{rop}}(\boldsymbol{U}, \boldsymbol{V})$ still concentrates around $\kappa^{\text{p}}(\boldsymbol{U}, \boldsymbol{V})$, and this can be combined with a union bound on a finite dataset implying that unbounded ROP features can still perform well in experiments with finite datasets. However, obtaining a data-independent approximation guarantee over *all pairs* of subspaces in the continuous space $\mathcal{G}(k, n)$ is much more demanding with such heavy-tailed distributions. In fact, following the proof techniques used in this work suggests that to approximate $\kappa^{\text{p}}$ with $\widehat{\kappa}^{\text{rop}}$ for all subspaces in $\mathbb{V}(k, n)$ requires $m = O((kn)^2)$ measurements, *i.e.*, an embedding dimension much larger than the dimension of $\mathcal{G}(k, n)$. This also follows (Chen et al., 2015; Cai & Zhang, 2015) who observed that $\psi^{\text{rop}}$ cannot satisfy the RIP with a reasonable number of measurements. These limitations motivate the introduction of bounded non-linear functions in Sec. 4.

Consequently, on any fixed subspace pair $\mathcal{P}, \mathcal{Q}$ of $\mathcal{G}(k, n)$ with respective bases $\boldsymbol{U}, \boldsymbol{V} \in \mathbb{V}(k, n)$, one can expect that the empirical kernel $\widehat{\kappa}^{\text{rop}}(\boldsymbol{U}, \boldsymbol{V})$, which amounts to averaging $m$ i.i.d. copies of $Z$, will deviate from a fixed bias from $\kappa^{\text{p}}(\boldsymbol{U}, \boldsymbol{V})$ with a failure probability decaying as $\exp(-c\sqrt{m})$. This is significantly slower than the rate $\exp(-cm)$ reached in sub-exponential regimes, *e.g.*, through RIP random matrices (Foucart, 2016); this also prevents us of approximating $\kappa^{\text{p}}$ uniformly over all subspaces of $\mathcal{G}(k, n)$ with a number of projections $m$ set to $\mathcal{O}(kn)$, the number of free parameters in any projector of $\mathcal{G}(k, n)$.

## 4 Bounded ROPs for Grassmannian kernel approximations

One can both remove the computational complexity and storage limitations of dense random projections and the heavy-tailed distribution of rank-one projections by composing ROPs with a bounded, non-linear function. Practically, given a bounded function $g : \mathbb{R} \to \mathbb{K}$ (with $\mathbb{K}$ equal to $\mathbb{R}$ or $\mathbb{C}$), with $|g(\lambda)| \leqslant 1$ for all $\lambda \in \mathbb{R}$ without loss of generality, we apply it componentwise to the rank-one projection operator $\psi^{\text{rop}}$, *i.e.*, we define

$$\psi^g : \boldsymbol{U} \in \mathbb{V}(k, n) \mapsto g(\psi^{\text{rop}}(\boldsymbol{U})) \in \mathbb{K}^m. \tag{9}$$

This raises several central questions. First, which kernel $\kappa$ is reached in expectation by the empirical kernel

$$\widehat{\kappa}^g(\boldsymbol{U}, \boldsymbol{V}) = \tfrac{1}{m} \langle \psi^g(\boldsymbol{U}), \psi^g(\boldsymbol{V}) \rangle, \quad \boldsymbol{U}, \boldsymbol{V} \in \mathbb{V}(k, n).$$

Is it a valid kernel with respect to the Grassmannian geometry? And can we uniformly bound the related approximation error between $\widehat{\kappa}^g$ and $\kappa^g$?

Among the many possible choices of $g$, we focus on two specific examples. The first is the *sign function*, $g(\lambda) = \text{sign}(\lambda)$ equals to $1$ if $\lambda > 0$ and to $-1$ otherwise. It is inspired by the one-bit compressive sensing literature (Boufounos & Baraniuk, 2008; Jacques et al., 2013; Foucart, 2016) and yields very light *binary* random features $\psi^\pm := \text{sign} \circ \psi^{\text{rop}}$ with codomain in $\{\pm 1\}^m$, *i.e.*, each subspace is encoded over no more than $m$ bits. The second is reminiscent of the random Fourier features (Rahimi & Recht, 2007) and applies the complex exponential map, $g(\lambda) = \exp(\mathrm{i}\omega\lambda)$ for some frequency $\omega \in \mathbb{R}$, to $\psi^{\text{rop}}$, *i.e.*, we define the *periodic* random features $\psi^\circ(\boldsymbol{U}) := \exp(\mathrm{i}\omega\psi^{\text{rop}}(\boldsymbol{U}))$ for any subspace $\mathcal{P}$ with basis $\boldsymbol{U} \in \mathbb{V}(k, n)$. We now study in detail these two cases.

### 4.1 Binary random features over $\mathcal{G}(k, n)$

To define the binary random features $\psi^\pm : \mathbb{V}(k, n) \to \{\pm 1\}^m$, we thus set $g(\cdot) = \text{sign}(\cdot)$ in (9), *i.e.*, given the $m$ random vectors $\boldsymbol{a}_i, \boldsymbol{b}_i \sim_{\text{i.i.d.}} \mathcal{N}(\boldsymbol{0}, \boldsymbol{I}_n)$, with $1 \leqslant i \leqslant m$,

$$\psi^\pm(\boldsymbol{U}) := \Big( \text{sign}(\boldsymbol{a}_i^\top \phi^{\text{p}}(\boldsymbol{U}) \boldsymbol{b}_i) \Big)_{i=1}^m, \quad \boldsymbol{U} \in \mathbb{V}(k, n). \tag{10}$$

This provides the empirical kernel

$$\widehat{\kappa}^{\pm}(\boldsymbol{U}, \boldsymbol{V}) := \tfrac{1}{m} \langle \psi^{\pm}(\boldsymbol{U}), \psi^{\pm}(\boldsymbol{V}) \rangle,$$

and, given any pair of bases $\boldsymbol{U}, \boldsymbol{V} \in \mathbb{V}(k, n)$, we are interested in the kernel

$$\kappa^{\pm}(\boldsymbol{U}, \boldsymbol{V}) := \mathbb{E}[\langle \psi^{\pm}(\boldsymbol{U}), \psi^{\pm}(\boldsymbol{V}) \rangle]. \tag{11}$$

The binary codomain of $\psi^{\pm}$ offers the advantage of reducing the cost of storing and possibly transmitting subspace random features: each of the $m$ random features requires a single bit instead of a floating-point number. This is especially valuable in settings where bandwidth or memory is constrained (*e.g.*, embedded devices or distributed systems). Moreover, binary vectors enable extremely fast bitwise operations, making both storage and computation far more efficient than in the full-precision case.

Unfortunately, when $k > 1$, there is no known closed form for $\kappa^{\pm}$. However, this kernel is valid as shown in the following proposition.

**Proposition 6** (Validity of $\kappa^{\pm}$). *For bases $\boldsymbol{U}, \boldsymbol{V} \in \mathbb{V}(k, n)$ whose subspaces form the principal angles $\theta_1, \ldots, \theta_k$, the kernel $\kappa^{\pm}$ is positive-definite, symmetric and only depends on these principal angles,* i.e., $\kappa^{\pm}(\boldsymbol{U}, \boldsymbol{V}) = f(\theta_1, \ldots, \theta_k)$ *for some function $f$. Moreover, given $\boldsymbol{g} \sim \mathcal{N}(\boldsymbol{0}, \boldsymbol{I}_n)$,*

$$\kappa^{\pm}(\boldsymbol{U}, \boldsymbol{V}) = 1 - \tfrac{2}{\pi} \mathbb{E} \angle (\phi^{\mathrm{P}}(\boldsymbol{U})\boldsymbol{g}, \phi^{\mathrm{P}}(\boldsymbol{V})\boldsymbol{g}), \tag{12}$$

*with $\angle(\boldsymbol{u}, \boldsymbol{v})$ the angle between two vectors $\boldsymbol{u}$ and $\boldsymbol{v}$.*

*Proof.* By design, $\kappa^{\pm}$ is obviously symmetric. Regarding its positive-semidefiniteness, we observe that, given an arbitrary number of $N$ elements $\boldsymbol{U}_1, \ldots, \boldsymbol{U}_N$ in $\mathbb{V}(k, n)$, with $\boldsymbol{P}_i := \boldsymbol{U}_i \boldsymbol{U}_i^{\top}$, the Gram matrix $\boldsymbol{K} \in \mathbb{R}^{N \times N}$ whose entries are $K_{ij} = \kappa^{\pm}(\boldsymbol{U}_i, \boldsymbol{U}_j)$ is such that for any $\boldsymbol{c} \in \mathbb{R}^N$, $\boldsymbol{c}^{\top} \boldsymbol{K} \boldsymbol{c} = \mathbb{E}(\boldsymbol{c}^{\top} \boldsymbol{p})^2 \geqslant 0$, with $\boldsymbol{p} \in \{\pm 1\}^N$ such that $p_i = \mathrm{sign}(\boldsymbol{a}^{\top} \boldsymbol{P}_i \boldsymbol{b})$.

Regarding the angular dependency of the kernel, since for any $n \times n$ matrix $\boldsymbol{M}$ each component of $\psi^{\mathrm{rop}}(\boldsymbol{M})$ is distributed as $\boldsymbol{a}^{\top} \boldsymbol{M} \boldsymbol{b}$, with $\boldsymbol{a}, \boldsymbol{b} \sim_{\mathrm{i.i.d.}} \mathcal{N}(\boldsymbol{0}, \boldsymbol{I}_n)$, by exploiting the rotational invariance of these random vectors, we have

$$\kappa^{\pm}(\boldsymbol{U}, \boldsymbol{V}) = \mathbb{E} \, \mathrm{sign}(\boldsymbol{a}^{\top} \boldsymbol{P} \boldsymbol{b}) \, \mathrm{sign}(\boldsymbol{a}^{\top} \boldsymbol{Q} \boldsymbol{b})$$
$$= \mathbb{E} \, \mathrm{sign}(\boldsymbol{a}^{\top} \boldsymbol{R} \boldsymbol{P} \boldsymbol{R}^{\top} \boldsymbol{b}) \, \mathrm{sign}(\boldsymbol{a}^{\top} \boldsymbol{R} \boldsymbol{Q} \boldsymbol{R}^{\top} \boldsymbol{b}) = \kappa^{\pm}(\boldsymbol{R}\boldsymbol{U}, \boldsymbol{R}\boldsymbol{V}),$$

for any orthogonal matrix $\boldsymbol{R} \in \mathsf{O}(n)$. Therefore, from Thm 4, $\kappa^{\pm}(\boldsymbol{U}, \boldsymbol{V})$ only depends on the principal angles between $\boldsymbol{U}$ and $\boldsymbol{V}$. Finally, the expression (12) is a simple consequence of the arcsin law used in Van Vleck & Middleton (1966, sec. 3, eq. 17) or Grothendieck's identity in Vershynin (2018, Lem. 3.6.6), *i.e.*, we show easily that, by the law of total expectation,

$$\kappa^{\pm}(\boldsymbol{U}, \boldsymbol{V}) = \mathbb{E}[\mathrm{sign}(\boldsymbol{a}^{\top}(\boldsymbol{P}\boldsymbol{b})) \, \mathrm{sign}(\boldsymbol{a}^{\top}(\boldsymbol{Q}\boldsymbol{b}))] = \tfrac{2}{\pi} \mathbb{E} \arcsin\Big(\frac{\boldsymbol{b}^{\top} \boldsymbol{P} \boldsymbol{Q} \boldsymbol{b}}{\|\boldsymbol{P}\boldsymbol{b}\|\|\boldsymbol{Q}\boldsymbol{b}\|}\Big),$$

which shows the result. $\qquad\square$

While we do not know any closed form formula for $\kappa^{\pm}$, (12) in Prop. 6 provides a noteworthy interpretation: $1 - \kappa^{\pm}(\boldsymbol{U}, \boldsymbol{V})$ is proportional to the average angle made by the projections of a Gaussian random vector on $\boldsymbol{U}$ and $\boldsymbol{V}$. Interestingly, a similar concept was introduced in (Ji et al., 2015) where the angle $\theta(\boldsymbol{U}, \boldsymbol{V})$ between $\boldsymbol{U}$ and $\boldsymbol{V}$ is defined as $\theta(\boldsymbol{U}, \boldsymbol{V}) = \arccos\big(\tfrac{1}{k} \sum_{i=1}^{k} \cos^2 \theta_i\big)$, a form of non-linear averaging of the principal angles and the angular similarity between these two subspaces as $\kappa^{\mathrm{sim}}(\boldsymbol{U}, \boldsymbol{V}) = 1 - \tfrac{1}{\pi} \theta(\boldsymbol{U}, \boldsymbol{V})$.

Note that, in the specific case where $k = 1$, the kernel has, however, a closed form.

**Proposition 7.** *For two vectors $\boldsymbol{u}, \boldsymbol{v} \in \mathbb{R}^n$ such that $\|\boldsymbol{u}\| = \|\boldsymbol{v}\| = 1$, with $\boldsymbol{P} = \boldsymbol{u}\boldsymbol{u}^{\top}, \boldsymbol{Q} = \boldsymbol{v}\boldsymbol{v}^{\top}$, and for $\widehat{\kappa}^{\pm}(\boldsymbol{U}, \boldsymbol{V})$ defined as earlier, we have*

$$\mathbb{E}[\widehat{\kappa}^{\pm}(\boldsymbol{u}, \boldsymbol{v})] = \Big(1 - \frac{2\theta}{\pi}\Big)^2, \tag{13}$$

*where $\theta$ is the single principal angle between $\boldsymbol{P}$ and $\boldsymbol{Q}$ (or more simply, the angle between $\boldsymbol{u}$ and $\boldsymbol{v}$).*

*Proof.* For $k = 1$, the projections of $\boldsymbol{g} \sim \mathcal{N}(\boldsymbol{0}, \boldsymbol{I}_n)$ onto the two subspaces are $\boldsymbol{Pg} = a\boldsymbol{u}$ and $\boldsymbol{Qg} = b\boldsymbol{v}$, where $a = \boldsymbol{u}^\top \boldsymbol{g}$ and $b = \boldsymbol{v}^\top \boldsymbol{g}$ are correlated Gaussian random variables with $\mathbb{E}[a] = \mathbb{E}[b] = 0$ and $\mathbb{E}[ab] = \cos\theta$. The angle between $\boldsymbol{Pg}$ and $\boldsymbol{Qg}$ is therefore $\theta$ iff $ab > 0$ and $\pi - \theta$ if $ab < 0$. Let $p = \mathbb{P}(ab > 0)$, then $\mathbb{E}[\angle(\boldsymbol{Pg}, \boldsymbol{Qg})] = \theta p + (\pi - \theta)(1 - p)$. On the other hand, since $\mathrm{sign}(a)\,\mathrm{sign}(b) = 1$ if $ab > 0$ and $-1$ otherwise, $\mathbb{E}[\mathrm{sign}(a)\,\mathrm{sign}(b)] = 2p - 1$. For correlated Gaussians with correlation $\cos\theta$, $\mathbb{E}[\mathrm{sign}(a)\,\mathrm{sign}(b)] = \frac{2}{\pi}\arcsin(\cos\theta) = -\frac{2\theta}{\pi}$ (see Rose & Smith (2002), p.230). Hence $2p - 1 = 1 - \frac{2\theta}{\pi}$ and therefore $p = 1 - \frac{\theta}{\pi}$. Substituting in the expectation above gives $\mathbb{E}\angle(\boldsymbol{Pg}, \boldsymbol{Qg}) = \theta(1 - \frac{\theta}{\pi}) + (\pi - \theta)\frac{\theta}{\pi}$. Inserting this quantity into (12) of Proposition 6, we obtain $\kappa^\pm(\boldsymbol{U}, \boldsymbol{V}) = 1 - \frac{2}{\pi}\mathbb{E}\angle(\boldsymbol{Pg}, \boldsymbol{Qg}) = (1 - \frac{2\theta}{\pi})^2$. $\qquad\square$

We now address the question of bounding the approximation error of $\widehat{\kappa}^\pm$ relatively to $\kappa^\pm$. We can first observe the following concentration phenomenon, *i.e.*, a pointwise approximation error bound on a fixed pair of subspaces.

**Proposition 8** (Pointwise approximation error of $\kappa^\pm$ by $\widehat{\kappa}^\pm$)**.** *Given a pair of subspaces in $\mathcal{G}(k, n)$ with bases $\boldsymbol{U}, \boldsymbol{V} \in \mathbb{V}(k, n)$, the $m$-component binary random feature map $\psi^\pm$ in (10) related to $m$ i.i.d. Gaussian random vectors $\boldsymbol{a}_j, \boldsymbol{b}_j \sim \mathcal{N}(\boldsymbol{0}, \boldsymbol{I}_n)$, and an error $\delta > 0$, we have*

$$\mathbb{P}(|\tfrac{1}{m}\langle \psi^\pm(\boldsymbol{U}), \psi^\pm(\boldsymbol{V})\rangle - \kappa^\pm(\boldsymbol{U}, \boldsymbol{V})| < \delta) \geqslant 1 - 2\exp(-\tfrac{1}{2}m\delta^2).$$

*Proof.* The proof is a direct application of Hoeffding's inequality (see *e.g.*, (Vershynin, 2018)) since given the random variables

$$X_j := \mathrm{sign}(\boldsymbol{a}_j^\top \boldsymbol{P}\boldsymbol{b}_j)\,\mathrm{sign}(\boldsymbol{a}_j^\top \boldsymbol{Q}\boldsymbol{b}_j), \quad 1 \leqslant j \leqslant m,$$

all $X_j$'s are i.i.d., bounded, and $\mathbb{E}[\widehat{\kappa}^\pm(\boldsymbol{U}, \boldsymbol{V})] = \mathbb{E}\sum_{j=1}^m X_j/m = \kappa^\pm(\boldsymbol{U}, \boldsymbol{V})$. $\qquad\square$

Second, we can show that $\widehat{\kappa}^\pm$ provides a uniform approximation of $\kappa^\pm$ over all subspace pairs in $\mathcal{G}(k, n)$.

**Proposition 9** (Uniform approximation error for $\widehat{\kappa}^\pm \approx \kappa^\pm$)**.** *Let $\delta > 0$ and $C, C', c > 0$ be absolute constants. If*

$$m \geqslant C\delta^{-2}nk\log(\tfrac{n^2}{k\delta^2}),$$

*then, with probability exceeding $1 - C'\exp(-c\delta^2 m)$,*

$$\sup_{\boldsymbol{U}, \boldsymbol{V} \in \mathbb{V}(k,n)} |\widehat{\kappa}^\pm(\boldsymbol{U}, \boldsymbol{V}) - \kappa^\pm(\boldsymbol{U}, \boldsymbol{V})| = \sup_{\boldsymbol{U}, \boldsymbol{V} \in \mathbb{V}(k,n)} |\tfrac{1}{m}\langle \psi^\pm(\boldsymbol{U}), \psi^\pm(\boldsymbol{V})\rangle - \kappa^\pm(\boldsymbol{U}, \boldsymbol{V})| \leqslant \delta.$$

The proof of this proposition is postponed to App. A. It requires some specific tools to tackle the discontinuities of the map $\psi^\pm$.

Prop. 9 thus shows that having a binary feature map $\psi^\pm$ with $m = \mathcal{O}(kn)$ components, *i.e.*, with $m$ greater than the intrinsic dimension of each subspace of $\mathcal{G}(k, n)$, allows us to approximate, with high and controlled probability, the expected kernel $\kappa^\pm$ of $k$-dimensional subspaces by the empirical kernel $\widehat{\kappa}^\pm$ reached by mere inner product of the random features of the two subspaces.

### 4.2 Periodic random features over $\mathcal{G}(k, n)$

The second bounded function that we are going to apply to the ROP operator $\psi^{\mathrm{rop}}$ is the complex exponential, *i.e.*, $g(\cdot) = \exp(\mathrm{i}\omega\cdot)$ for some frequency $\omega \in \mathbb{R}$. The resulting periodic random feature map $\psi^\circ = g \circ \psi^{\mathrm{rop}}$ is reminiscent of the random Fourier features from Rahimi & Recht (2007) composing random projections of vectors with a periodic non-linear function; given the $m$ random vectors $\boldsymbol{a}_i, \boldsymbol{b}_i \sim_{\mathrm{i.i.d.}} \mathcal{N}(\boldsymbol{0}, \boldsymbol{I}_n)$, $1 \leqslant i \leqslant m$, it is defined as

$$\psi^\circ(\boldsymbol{U}) := \Big(\exp(\mathrm{i}\omega\,\boldsymbol{a}_i^\top \phi^{\mathrm{p}}(\boldsymbol{U})\boldsymbol{b}_i)\Big)_{i=1}^m, \quad \boldsymbol{U} \in \mathbb{V}(k, n). \tag{14}$$

One can then show that the empirical kernel $\widehat{\kappa}^\circ := \frac{1}{m}\langle \psi^\circ(\boldsymbol{U}), \psi^\circ(\boldsymbol{V})\rangle$ between two $k$-dimensional subspace bases $\boldsymbol{U}, \boldsymbol{V} \in \mathbb{V}(k, n)$ is an unbiased estimator of the following kernel.

**Proposition 10** (Validity of $\kappa^\circ$). *Given $\boldsymbol{U}, \boldsymbol{V} \in \mathbb{V}(k,n)$, with principal angles $\theta_1, \ldots, \theta_k$ between them, and $\psi^\circ$ defined as above, the kernel $\kappa^\circ$ is positive-definite, symmetric and only depends on the principal angles between the two subspaces $\boldsymbol{U}$ and $\boldsymbol{V}$. Moreover,*

$$\kappa^\circ(\boldsymbol{U}, \boldsymbol{V}) := \mathbb{E}\widehat{\kappa}^\circ(\boldsymbol{U}, \boldsymbol{V}) = \prod_{j=1}^{k}(1 + \omega^2 \sin^2 \theta_j)^{-1}. \tag{15}$$

*Proof.* The symmetry and positive-semidefiniteness of $\kappa^\circ$ are proven similarly to the one of $\kappa^\pm$. Regarding the dependence of $\kappa^\circ$ to the subspace principal angles, we first note that, given $\boldsymbol{U}, \boldsymbol{V} \in \mathbb{V}(k,n)$ and the random vectors $\boldsymbol{a}, \boldsymbol{b} \sim_{\text{i.i.d.}} \mathcal{N}(\boldsymbol{0}, \boldsymbol{I}_n)$,

$$\kappa^\circ(\boldsymbol{U}, \boldsymbol{V}) = \mathbb{E}\widehat{\kappa}^\circ(\boldsymbol{U}, \boldsymbol{V}) = \mathbb{E}\exp(\mathrm{i}\omega \boldsymbol{a}^\top (\boldsymbol{P} - \boldsymbol{Q})\boldsymbol{b}),$$

with the projectors $\boldsymbol{P} = \boldsymbol{U}\boldsymbol{U}^\top$ and $\boldsymbol{Q} = \boldsymbol{V}\boldsymbol{V}^\top$. The rotational invariance of the random vectors $\boldsymbol{a}$ and $\boldsymbol{b}$ thus shows that $\kappa^\circ$ is rotationally invariant, *i.e.*, $\kappa^\circ(\boldsymbol{U}, \boldsymbol{V}) = \kappa^\circ(\boldsymbol{R}\boldsymbol{U}, \boldsymbol{R}\boldsymbol{V})$, for any rotation $\boldsymbol{R} \in \mathsf{SO}(n)$. From Thm 4, $\kappa^\circ(\boldsymbol{U}, \boldsymbol{V})$ only depends on the principal angles between $\boldsymbol{U}$ and $\boldsymbol{V}$. Moreover, following Conway et al. (1996), we can always apply a rotation $\boldsymbol{R}$ such that $\boldsymbol{U}$ and $\boldsymbol{V}$ are "rotated" to make angles $\pm\theta_j/2$ around the identity, *i.e.*,

$$\boldsymbol{U}^\top = (\boldsymbol{C}, \boldsymbol{S}, \boldsymbol{0}_{k \times n - 2k}) \in \mathbb{R}^{k \times n}, \quad \boldsymbol{V}^\top = (\boldsymbol{C}, -\boldsymbol{S}, \boldsymbol{0}_{k \times n - 2k}) \in \mathbb{R}^{k \times n},$$

with $\boldsymbol{C} = \mathrm{diag}\left(\cos(\theta_1/2), \ldots, \cos(\theta_k/2)\right)$ and $\boldsymbol{S} = \mathrm{diag}\left(\sin(\theta_1/2), \ldots, \sin(\theta_k/2)\right)$.

Using this representation, a few computations show that, up to a permutation matrix $\boldsymbol{\Pi} \in \{0,1\}^{n \times n}$, $\boldsymbol{P} - \boldsymbol{Q}$ is block-diagonal with $k$ independent $2 \times 2$ blocks, *i.e.*, $\boldsymbol{P} - \boldsymbol{Q} = \boldsymbol{\Pi}^\top \boldsymbol{\Gamma} \boldsymbol{\Pi}$, with

$$\boldsymbol{\Gamma} := \mathrm{bdiag}\left(\sin\theta_1 \begin{pmatrix} 0 & 1 \\ 1 & 0 \end{pmatrix}, \sin\theta_2 \begin{pmatrix} 0 & 1 \\ 1 & 0 \end{pmatrix}, \ldots, \sin\theta_k \begin{pmatrix} 0 & 1 \\ 1 & 0 \end{pmatrix}, 0, \ldots, 0\right) \in \mathbb{R}^{n \times n},$$

where we append as many zeros as necessary to get an $n \times n$ matrix. Since the distribution of the vectors $\boldsymbol{a}$ and $\boldsymbol{b}$ is permutation invariant, $\boldsymbol{a}^\top (\boldsymbol{P} - \boldsymbol{Q})\boldsymbol{b}$ is distributed as $\boldsymbol{a}^\top \boldsymbol{\Gamma} \boldsymbol{b}$, so that $\kappa^\circ(\boldsymbol{U}, \boldsymbol{V}) = \mathbb{E}\exp(\mathrm{i}\boldsymbol{a}^\top \boldsymbol{\Gamma} \boldsymbol{b})$. However, since $\boldsymbol{a}^\top \boldsymbol{\Gamma} \boldsymbol{b} = \sum_{j=1}^{k} \sin\theta_{2j}(a_{2j}b_{2j+1} + a_{2j+1}b_{2j})$, we get from the independence of the components of $\boldsymbol{a}$ and $\boldsymbol{b}$,

$$\kappa^\circ(\boldsymbol{U}, \boldsymbol{V}) = \prod_{j=1}^{k} \mathbb{E}\exp(\mathrm{i}\, a_{2j}b_{2j+1} \sin\theta_j)\mathbb{E}\exp(\mathrm{i}\, b_{2j}a_{2j+1} \sin\theta_j)$$
$$= \prod_{j=1}^{k} \left(\mathbb{E}\exp(\mathrm{i}\sin\theta_j gg')\right)^2,$$

with $g, g' \sim_{\text{i.i.d.}} \mathcal{N}(0,1)$. Defining $u = (g + g')/\sqrt{2}$, $v = (g - g')/\sqrt{2}$, we observe that $gg' = \frac{1}{2}(u^2 - v^2)$ with independent $u, v \sim \mathcal{N}(0,1)$, *i.e.*, $2gg'$ is the difference of two independent $\chi^2$-distributions with one degree of freedom. Since for any $t \in \mathbb{R}$ $\mathbb{E}e^{\mathrm{i}tX} = (1 - 2\mathrm{i}t)^{-1/2}$ if $X \sim \chi_1^2$, we obtain for any $\alpha \in \mathbb{R}$,

$$\mathbb{E}\exp(\mathrm{i}\alpha gg') = (1 - \mathrm{i}\alpha)^{-1/2}(1 + \mathrm{i}\alpha)^{-1/2} = (1 + \alpha^2)^{-1/2}.$$

Meaning that, $\kappa^\circ(\boldsymbol{U}, \boldsymbol{V}) = \prod_{j=1}^{k}\left(\mathbb{E}\exp(\mathrm{i}\omega\, gg' \sin\theta_j)\right)^2 = \prod_{j=1}^{k}(1 + \omega^2 \sin^2 \theta_j)^{-1}$, which completes the proof. $\square$

**Remark 11.** *While the periodic random feature $\psi^\circ$ defined in* (14) *is complex, the resulting kernel $\kappa^\circ$ is real, i.e., the imaginary part of $\widehat{\kappa}^\circ$ vanishes in expectation. Moreover, we show easily that the $2m$-component periodic random feature map*

$$\psi^{\circ\prime}(\boldsymbol{U}) := \begin{pmatrix} \Re(\psi^\circ(\boldsymbol{U})) \\ \Im(\psi^\circ(\boldsymbol{U})) \end{pmatrix}, \tag{16}$$

*provides also an unbiased estimator of $\kappa^\circ(\boldsymbol{U}, \boldsymbol{V})$ via $\frac{1}{m}\langle \psi^{\circ\prime}(\boldsymbol{U}), \psi^{\circ\prime}(\boldsymbol{V})\rangle$ since*

$$\langle \psi^{\circ\prime}(\boldsymbol{U}), \psi^{\circ\prime}(\boldsymbol{V})\rangle = \Re[\langle \psi^\circ(\boldsymbol{U}), \psi^\circ(\boldsymbol{V})\rangle].$$

Just like we did for $\kappa^\pm$ we now consider the approximation error we make by replacing $\kappa^\circ$ by $\widehat{\kappa}^\circ$. We first study a bound on the pointwise approximation error, *i.e.*, the absolute error made on a fixed pair of subspaces.

**Proposition 12** (Pointwise approximation error of $\kappa^\circ$ by $\widehat{\kappa}^\circ$). *Given two subspace bases $\boldsymbol{U}, \boldsymbol{V} \in \mathbb{V}(k,n)$, the $m$-component periodic random feature map $\psi^\circ$ in (14) and related to $m$ i.i.d. Gaussian random vectors $\boldsymbol{a}_j, \boldsymbol{b}_j \sim \mathcal{N}(\boldsymbol{0}, \boldsymbol{I}_n)$, and an error $\delta > 0$, we have*

$$\mathbb{P}\Big(|\tfrac{1}{m}\langle\psi^\circ(\boldsymbol{U}), \psi^\circ(\boldsymbol{V})\rangle - \kappa^\circ(\boldsymbol{U},\boldsymbol{V})| < \delta\Big) \geqslant 1 - 4\exp(-\tfrac{1}{4}m\delta^2).$$

*Proof.* Defining the $m$ i.i.d. complex random variables $X_j := \psi^\circ{}_j(\boldsymbol{U})\psi^{\circ *}_j(\boldsymbol{V})$, for $1 \leqslant j \leqslant m$, we have $m\widehat{\kappa}^\circ(\boldsymbol{U},\boldsymbol{V}) = \sum_{j=1}^m X_j$, $m\kappa^\circ(\boldsymbol{U},\boldsymbol{V}) = \sum_{j=1}^m \mathbb{E}X_j$ and

$$\mathbb{P}[|\widehat{\kappa}^\circ(\boldsymbol{U},\boldsymbol{V}) - \kappa^\circ(\boldsymbol{U},\boldsymbol{V})| \geqslant \delta]$$
$$\leqslant \mathbb{P}[|\tfrac{1}{m}\textstyle\sum_{j=1}^m \Re(X_j) - \Re\kappa^\circ(\boldsymbol{U},\boldsymbol{V})| \geqslant \delta/\sqrt{2}] + \mathbb{P}[|\tfrac{1}{m}\textstyle\sum_{j=1}^m \Im(X_j) - \Im\kappa^\circ(\boldsymbol{U},\boldsymbol{V})| \geqslant \delta/\sqrt{2}].$$

Since $\max(|\Re(X_j)|, |\Im(X_j)|) \leqslant 1$, Hoeffding's inequality provides

$$\mathbb{P}[|\tfrac{1}{m}\textstyle\sum_{j=1}^m \Re(X_j) - \Re\kappa^\circ(\boldsymbol{U},\boldsymbol{V})| \geqslant \delta/\sqrt{2}] \leqslant 2\exp(-\tfrac{1}{4}m\delta^2),$$

and similarly for the imaginary part. Gathering the bounds provides the result. $\qquad\square$

Armed with these results we can now attack the uniform bound in the same way as we did for $\kappa^\pm$. The statement and the proof follow a very similar pattern.

**Proposition 13** (Uniform approximation error for $\widehat{\kappa}^\circ \approx \kappa^\circ$). *Given a distortion $\delta > 0$, and some absolute constants $C, C', c > 0$, if*

$$m \geqslant C\delta^{-2}nk\log(\omega\sqrt{k}n/\delta),$$

*then*

$$\sup_{\boldsymbol{U},\boldsymbol{V}\in\mathbb{V}(k,n)} |\widehat{\kappa}^\circ(\boldsymbol{U},\boldsymbol{V}) - \kappa^\circ(\boldsymbol{U},\boldsymbol{V})| = \sup_{\boldsymbol{U},\boldsymbol{V}\in\mathbb{V}(k,n)} \left|\tfrac{1}{m}\langle\psi^\circ(\boldsymbol{U}), \psi^\circ(\boldsymbol{V})\rangle - \kappa^\circ(\boldsymbol{U},\boldsymbol{V})\right| \leqslant \delta$$

*with probability at least $1 - C'\exp(-c\delta^2 m)$, for some absolute constant $c, c_0 > 0$.*

*Proof.* The proof for this proposition is delayed to appendix B. $\qquad\square$

**[R-XJL8]**

**Remark 14.** *The uniform guarantees of Prop. 9 and Prop. 13 are worst-case results over all pairs of $k$-dimensional subspaces in $\mathbb{R}^n$. Their dependences on $nk$ are therefore coherent with the intrinsic dimension of $\mathcal{G}(k,n)$. It may be possible to improve resquirements on $m$ when restricting the approximation to structured subsets of $\mathcal{G}(k,n)$, for instance subspaces with sparse bases, or subspaces satisfying additional constraints on their principal angles. In this case, the proof would require replacing the covering number of the full Grassmannian by a covering number adapted to the restricted class. We leave this direction for future work.*

We conclude this section by studying the shape and properties of the kernel $\kappa^\circ$ and its dependence to the frequency parameter $\omega$; in particular, we show below that this parameter determines two specific regimes where $\kappa^\circ$ is either close to the projection kernel or to a kernel that is reminiscent of the Binet-Cauchy kernel. The first regime is valid at large frequency.

**Proposition 15** (Large-frequency regime of $\kappa^\circ$). *Let us consider two subspace bases $\boldsymbol{U}, \boldsymbol{V} \in \mathbb{V}(k,n)$ with projectors $\boldsymbol{P}$ and $\boldsymbol{Q}$ and principal angles $0 < \theta_1 \leqslant \cdots \leqslant \theta_k \leqslant \frac{\pi}{2}$. We have*

$$\lim_{\omega\to+\infty} \omega^{2k}\kappa^\circ(\boldsymbol{U},\boldsymbol{V}) = \textstyle\prod_{j=1}^k (\sin^2\theta_j)^{-1} = \mathrm{pdet}(\boldsymbol{P}(\boldsymbol{I}_n - \boldsymbol{Q})\boldsymbol{P})^{-1}.$$

*Proof.* This is a direct consequence of (15). $\qquad\square$

Interestingly, one can relate the shape of $\kappa^\circ$ in this large-frequency limit to an extension of the Binet-Cauchy kernel. Generalising this kernel to subspaces $\boldsymbol{U}_1$ and $\boldsymbol{U}_2$ of different dimensions ($k_1$ and $k_2$, respectively) and principal angles $\{\theta'_j\}_{j=1}^{\min(k_1,k_2)}$, through the definition

$$\lim_{\omega\to+\infty} \omega^{2k}\kappa^\circ(\boldsymbol{U},\boldsymbol{V}) := \prod_{j=1}^{\min(k_1,k_2)} \cos^2\theta'_j,$$

we get, under the conventions of Prop. 15 and if $k < n/2$,

$$\kappa^\circ(\boldsymbol{U},\boldsymbol{V}) = \kappa^{\mathrm{BC}}(\boldsymbol{U},\boldsymbol{V}^\perp)^{-1},$$

since $\boldsymbol{U}$ and $\boldsymbol{V}^\perp$ then make $k$ principal angles $\{\frac{\pi}{2}-\theta_j\}_{j=1}^k$ (Mandolesi, 2021; Knyazev & Zhu, 2012). Following a previous interpretation, this shows also that, in this regime, the kernel $\kappa^\circ(\boldsymbol{U},\boldsymbol{V})$ is inversely proportional to the volume made by the basis $\boldsymbol{U}$ when projected onto $\boldsymbol{V}$.

The second regime is reached at small frequency. We show below that $\kappa^\circ$ then mimics a radial basis function (RBF) kernel (see Eq. (5)).

**Proposition 16** (Small-frequency regime of $\kappa^\circ$). *Let us consider two subspace bases $\boldsymbol{U},\boldsymbol{V} \in \mathbb{V}(k,n)$ with projectors $\boldsymbol{P}$ and $\boldsymbol{Q}$. We have*

$$|\kappa^\circ(\boldsymbol{U},\boldsymbol{V}) - \exp\left(-\tfrac{1}{2}\omega^2\|\boldsymbol{P}-\boldsymbol{Q}\|_F^2\right)| \leqslant \tfrac{1}{4}\omega^4\|\boldsymbol{P}-\boldsymbol{Q}\|_F^2 \leqslant \tfrac{k}{2}\omega^4.$$

*Proof.* Given the two subspace principal angles $\{\theta_j\}_{j=1}^k$, since $\log\kappa^\circ(\boldsymbol{U},\boldsymbol{V}) = -\sum_{j=1}^k \log(1+\omega^2\sin^2\theta_j)$ and $|\log(1+t)-t| \leqslant t^2/2$ for any $t > 0$, we get $|\log\kappa^\circ(\boldsymbol{U},\boldsymbol{V}) + \omega^2\sum_{j=1}^k\sin^2\theta_j| \leqslant \tfrac{1}{2}\omega^4\sum_{j=1}^k\sin^4\theta_j \leqslant \tfrac{1}{2}\omega^4\sum_{j=1}^k\sin^2\theta_j$. Therefore, since $|e^u - e^v| \leqslant \max(e^u,e^v)|u-v|$ for real $u$ and $v$, we get

$$|\kappa^\circ(\boldsymbol{U},\boldsymbol{V}) - \exp(-\omega^2\sum_{j=1}^k\sin^2\theta_j)| \leqslant \tfrac{1}{2}\omega^4\sum_{j=1}^k\sin^2\theta_j,$$

where we used the fact that, by design, $\kappa^\circ(\boldsymbol{U},\boldsymbol{V}) \leqslant 1$, and $\exp(-\omega^2\sum_{j=1}^k\sin^2\theta_j) \leqslant 1$. The result is obtained by observing that $2\sum_{j=1}^k\sin^2\theta_j = \|\boldsymbol{P}-\boldsymbol{Q}\|_F^2$. $\qquad\square$

[R-XJL8] In summary, Props. 15 and 16 show that $\kappa^\circ$ can adopt two distinct behaviours. At large frequency $\omega \gg 1$, $\kappa^\circ(\boldsymbol{U},\boldsymbol{V})$ is related to a variant of the Binet-Cauchy kernel between $\boldsymbol{U}$ and the orthogonal subspace $\boldsymbol{V}^\perp$, and it is more sensitive to small changes in the subspace principal angles. By contrast, at low frequency, $\kappa^\circ$ is well approximated by a projection RBF kernel of the form $\exp(-\tfrac{1}{2}\omega^2\|\boldsymbol{P}-\boldsymbol{Q}\|_F^2)$, with an error controlled by $\omega^4$. In the very small-frequency regime, this RBF kernel is itself close to the constant kernel equal to 1, and is therefore less discriminative. This regime can however be useful when a very smooth similarity is desired. Larger values of $\omega$ lead to more a discriminative kernels, while it remains well-defined for every $\omega \in \mathbb{R}$ by Prop. 10. These characteristics are numerically confirmed in Sec. 8.2 when comparing 2-dimensional subspaces.

## 5 Computational and memory complexity analysis

The rapidity and efficiency with which we compute the two random feature maps $\psi^\pm$ and $\psi^\circ$ (introduced in Sec. 4) on subspaces of $\mathcal{G}(k,n)$ depend on those of the ROP projection operator $\psi^{\mathrm{rop}}$.

A priori, when we compute the $m$-component ROP $\psi^{\mathrm{rop}}(\boldsymbol{U})$ of a subspace basis $\boldsymbol{U} \in \mathbb{V}(k,n)$, each component $1 \leqslant j \leqslant m$ of $\psi^{\mathrm{rop}}$ requires storing $\mathcal{O}(n)$ values from the random generation of the probing vectors $\boldsymbol{a}_j, \boldsymbol{b}_j \sim_{\mathrm{i.i.d.}} \mathcal{N}(\boldsymbol{0}, \boldsymbol{I}_n)$. It also needs $\mathcal{O}(kn)$ operations from the cost pf computing $\boldsymbol{a}_j^\top \boldsymbol{U}\boldsymbol{U}^\top \boldsymbol{b}_j$ from the inner product of $\bar{\boldsymbol{a}}_j := \boldsymbol{U}^\top \boldsymbol{a}_j$ and $\bar{\boldsymbol{b}}_j := \boldsymbol{U}^\top \boldsymbol{b}_j$. This way of factoring the computation also requires storing $\mathcal{O}(km)$ intermediate values, those from the $k$-component vectors $\bar{\boldsymbol{a}}_j$ and $\bar{\boldsymbol{b}}_j$, and an additional computational complexity $\mathcal{O}(km)$ for the intermediate inner products evaluation. Consequently, the projection of one subspace into $\mathbb{R}^m$ through $\psi^{\mathrm{rop}}$ needs a total storage of $\mathcal{O}(mn+km)$ values and a computational complexity of $\mathcal{O}(kmn+km) = \mathcal{O}(kmn)$.

Regarding the complexities of $\psi^{\pm}$ and $\psi^{\circ}$, considering that evaluating their respective non-linear functions brings negligible extra computations, we established through Props. 9 and 13 that we must take $m = \mathcal{O}(\delta^{-2}kn)$ random features, up to log factors, to ensure that the approximation error of the target kernels is uniformly bounded by $\delta > 0$. The overall memory complexity of $\psi^{\pm}$ and $\psi^{\circ}$ is thus $\mathcal{O}(\delta^{-2}(kn^2 + k^2n))$[**R-FSmh**] $= \mathcal{O}(kn^2)$ (since $k \leqslant n$), with a computational complexity of $\mathcal{O}(\delta^{-2}k^2n^2)$.

Overall, for large values of $k$ and $n$, these complexities are smaller complexities than those of the dense random projections $\Psi$ introduced in Sec. 3.1. Since dense random projections require us to pick $m = \mathcal{O}(\delta^{-2}k^3n)$ projections to approximate the projection kernel with uniform error $\delta$, they have a complexity $\mathcal{O}(\delta^{-2}k^3n^3)$ for both storing the $m$ probing matrices of size $n^2$ and computing the $m$ dense projections.

Nevertheless, the computational and memory complexities of $\psi^{\mathrm{rop}}$ may still be large for large datasets of subspaces in high dimensions.

# 6 Faster random features over $\mathcal{G}(k, n)$

A specific line of work has shown that random projections of vectors performed by dense random matrices can be replaced by compositions of *fast orthogonal transforms* and *random diagonal sign matrices*, creating vectors that behave similarly to i.i.d. Gaussian vectors at a lower computational cost. Examples include the Fast Johnson–Lindenstrauss Transform (Ailon & Chazelle, 2009), the Fastfood construction for random features (Le et al., 2013), the BCH–code based JL embeddings (Ailon & Liberty, 2009), Orthogonal Random Features (Yu et al., 2016), or Hadamard blocks for compressive covariance sketching (Vayer et al., 2023). While differing in detail, these methods share a common leitmotif: a small number of random diagonal matrices mixed with fast transforms (Hadamard or Fourier), sometimes organised in blocks to generate large numbers of structured random vectors. In particular, the use of a small number of successive Hadamard–diagonal compositions has been studied explicitly in the context of structured random projections, with theoretical and empirical evidence that as few as three such blocks already yield distributions close to Gaussian (Bojarski et al., 2017; Choromanski et al., 2016).

We decide to adapt this philosophy to rank-one projections of low-rank matrices (obtained from low-rank subspace projectors) by adopting the method proposed in (Choromanski et al., 2016; Yu et al., 2016; Vayer et al., 2023). Mathematically, given a subspace basis $\boldsymbol{U} \in \mathbb{V}(k, n)$ (with projector $\boldsymbol{P} = \boldsymbol{U}\boldsymbol{U}^{\top}$) made of $k$ orthonormal vectors $\boldsymbol{U} = (\boldsymbol{u}_1, \ldots, \boldsymbol{u}_k)$, defining $T = \lceil m/n \rceil$, we pick the $m$ probing vectors $\{\boldsymbol{a}_j\}_{j=1}^m$ and $\{\boldsymbol{b}_j\}_{j=1}^m$ involved in $\psi^{\mathrm{rop}}(\boldsymbol{U})$ as the $m$ first columns of two $n \times Tn$ matrices $\overline{\boldsymbol{G}} := [\boldsymbol{G}_1, \ldots, \boldsymbol{G}_T]$ and $\overline{\boldsymbol{G}}' := [\boldsymbol{G}_1', \ldots, \boldsymbol{G}_T']$, respectively. Each $n \times n$ matrix $\boldsymbol{G}_t, \boldsymbol{G}_t'$, $1 \leqslant t \leqslant T$, are i.i.d. as the random matrix $\boldsymbol{G}$ whose *distribution* is defined by the following $S$-factor generative model:

$$\boldsymbol{G} = \sqrt{n} \prod_{j=1}^{S} (\operatorname{diag}(\boldsymbol{\varepsilon}_j)\boldsymbol{H}), \quad \text{with } \boldsymbol{\varepsilon}_1, \ldots, \boldsymbol{\varepsilon}_S \sim_{\text{i.i.d.}} \boldsymbol{\varepsilon}, \qquad (17)$$

with the normalised Walsh-Hadamard matrix $\boldsymbol{H} \in \{\pm 1/\sqrt{n}\}^{n \times n} \cap \mathsf{O}(n)$, and $\boldsymbol{\varepsilon}$ an $n$-length Rademacher random vector with entries i.i.d. as $\mathcal{U}(\pm 1)$. The normalisation $\sqrt{n}$ in (17) ensures that the columns of $\boldsymbol{G}$ have squared norm $n$, which is also the expected squared norm of a Gaussian random vector in $\mathbb{R}^n$.

For small values of $S$ (typically $S = 3$), picking $m$ columns of $\overline{\boldsymbol{G}}$ (or $\overline{\boldsymbol{G}}'$) is known to generate an $n \times m$ matrix $\boldsymbol{W}^{\top}$ with similar properties to an $n \times m$ Gaussian random matrix, *e.g.*, $\boldsymbol{W}$ satisfies the Johnson-Lindenstrauss lemma with high probability (Ailon & Liberty, 2009).

Thanks to this specific design of the ROP probing vectors, to project the subspace $\boldsymbol{U}$, we first compute, for $1 \leqslant i \leqslant k$,

$$(\boldsymbol{a}_j^{\top}\boldsymbol{u}_i)_{j=1}^m = \boldsymbol{S}(\overline{\boldsymbol{G}}^{\top}\boldsymbol{u}_i), \quad (\boldsymbol{b}_j^{\top}\boldsymbol{u}_i)_{j=1}^m = \boldsymbol{S}(\overline{\boldsymbol{G}}'^{\top}\boldsymbol{u}_i). \qquad (18)$$

with $\boldsymbol{S} \in \{0, 1\}^{m \times Tn}$ the selection matrix extracting the first $m$ components of any vector in $\mathbb{R}^{Tn}$. From the structure of $\overline{\boldsymbol{G}}$ and $\overline{\boldsymbol{G}}'$ and the use of the fast (butterfly) Walsh-Hadamard transform[1], computing $\overline{\boldsymbol{G}}^{\top}\boldsymbol{u}_i$ and $\overline{\boldsymbol{G}}'^{\top}\boldsymbol{u}_i$ for all $1 \leqslant i \leqslant k$ in (18) requires $\mathcal{O}(SkTn \log n) = \mathcal{O}(Skm \log n)$ operations, and storing

---

[1]We used the implementation developed for (Andoni et al., 2015) and available at `https://github.com/FALCONN-LIB/FFHT`.

| Random feature | # of features $m$ | Memory | Computation | Error bound | Target kernel |
|---|---|---|---|---|---|
| Dense $\Psi$ | $\mathcal{O}(\delta^{-2}k^3 n)$ | $\mathcal{O}(\delta^{-2}k^3 n^3)$ | $\mathcal{O}(\delta^{-2}k^3 n^3)$ | Prop. 5 | Proj. kernel $\kappa^{\mathrm{p}}$ |
| Binary $\psi^{\pm}$ | $\mathcal{O}(\delta^{-2}kn)$ | $\mathcal{O}(\delta^{-2}(kn^2))$ | $\mathcal{O}(\delta^{-2}k^2 n^2)$ | Prop. 9 | New kernel: $\kappa^{\pm}$ |
| Periodic $\psi^{\circ}$ | $\mathcal{O}(\delta^{-2}kn)$ | $\mathcal{O}(\delta^{-2}(kn^2))$ | $\mathcal{O}(\delta^{-2}k^2 n^2)$ | Prop. 13 | New kernel: $\kappa^{\circ}$ |
| Structured ($\supset \psi^{\mathrm{st}}$) | $\mathcal{O}(\delta^{-2}kn)$ (assumed) | $\mathcal{O}(\delta^{-2}k^2 n)$ | $\mathcal{O}(\delta^{-2}k^2 n \log n)$ | (assumed) | Various: $\kappa^{\pm}$, $\kappa^{\circ}$ |

Table 1: Computational and memory complexities per instance for $k$-dimensional subspaces of $\mathbb{R}^n$ to get a uniform approximation error $\delta > 0$ between the empirical kernel and the related target kernel.

$\mathcal{O}(SkTn) = \mathcal{O}(Skm)$ Rademacher components (from (17)) as well as the storage of the $\mathcal{O}(km)$ intermediate values computed in (18). The final multiplication by $\boldsymbol{S}$ has negligible complexity.

Next, we form the fast, structured ROP operator

$$\psi^{\mathrm{st}}(\boldsymbol{U}) = \Big( \sum_{i=1}^{k} \boldsymbol{a}_j^\top \boldsymbol{u}_i \boldsymbol{u}_i^\top \boldsymbol{b}_j \Big)_{j=1}^m = \Big( \boldsymbol{a}_j^\top \boldsymbol{U} \boldsymbol{U}^\top \boldsymbol{b}_j \Big)_{j=1}^m,$$

which needs an extra $\mathcal{O}(km)$ operations.

The whole computation of $\psi^{\mathrm{st}}(\boldsymbol{U})$ requires a total storage of $\mathcal{O}(Skm)$ values, and a total computational complexity of $\mathcal{O}(Skm \log n)$ operations. In practice, we find that using more than $S = 3$ provides no significant gain. This empirical observation is consistent with the findings in (Yu et al., 2016, Equation 5) or Vayer et al. (2023, Section 3.1), where triple Hadamard blocks are shown to be sufficient for sketching covariance matrices.

Therefore, considering that $S = \mathcal{O}(1)$, and assuming that the results of Props. 9 and 13 still hold when $\psi^{\pm}$ and $\psi^{\circ}$ rely on $\psi^{\mathrm{rop}}$, if we take $m = \mathcal{O}(\delta^{-2}kn)$ random features (up to log factors) to ensure that the approximation error of the target kernels is uniformly bounded by $\delta > 0$, we can state that the computation of both type of random features, binary or periodic, requires a total storage of $\mathcal{O}(\delta^{-2}k^2 n)$ values, and a total computational complexity of $\mathcal{O}(\delta^{-2}k^2 n \log n)$ operations.

We summarise the computational and memory costs of the various random feature maps introduced so far in Tab. 1.

# 7 Related Works

Representing data through low-dimensional subspaces is a well-established method. It arises naturally in applications such as representing images invariant under different illuminations (Basri & Jacobs, 2003), motion segmentation (Rao et al., 2010), and dynamic subspace modelling (Liu et al., 2013; McGonigle & Peng, 2021; Jiao et al., 2018). In such settings, the object of interest is not individual samples but the subspace they span, leading naturally to the Grassmannian manifold $\mathcal{G}(k, n)$ as the underlying representation space.

Early supervised classification methods for subspace-valued data relied on nearest-subspace algorithms, notably the CLAFIC approach of Watanabe et al. (1967); Watanabe & Pakvasa (1973), which represents each class by a reference subspace and assigns labels by projecting new instances into the reference subspaces. More recent works embed subspaces into Euclidean spaces allowing efficient nearest-neighbour search. In particular, Basri et al. (2011) and Ji et al. (2015) propose deterministic and binary embeddings based on vectorised projection matrices, approximately preserving the projection distance and enabling scalable classification.

[R-XJL8] Another related line of work is about Euclidean distance geometry and the recovery of Gram matrices from partial distance measurements (Dokmanić et al., 2015; Tasissa & Lai, 2019). In this setting, a low-rank positive-semidefinite matrix such as $\boldsymbol{P} = \boldsymbol{U}\boldsymbol{U}^\top$ can be interpreted as a Gram matrix, and

measurements of the form $(\boldsymbol{e}_i - \boldsymbol{e}_j)^\top \boldsymbol{P}(\boldsymbol{e}_i - \boldsymbol{e}_j)$ correspond to squared distances between embedded points. This is related to our use of measurements of projection matrices, but with a different objective. They aim to recover the matrix $\boldsymbol{P}$ itself from limited measurements, whereas we aim to build random feature maps whose inner products approximate a Grassmannian kernel. Hence, bounded non-linear functions are not introduced for projector recovery, but to obtain kernel estimators with controlled concentration.

Beyond distance-based methods, Grassmannian kernel machines are powerful tools for learning from subspace data. Classical kernels defined through principal angles have been extensively studied in Harandi et al. (2014). While effective, these kernels suffer from poor scalability, as kernel-based learning requires forming and storing an $N \times N$ Gram matrix, which quickly becomes prohibitive for large datasets (Rahimi & Recht, 2007).

Several works have addressed this limitation through randomised representations. A first line of work applies random projections to vectorised projection matrices, relying on the Johnson-Lindenstrauss lemma to provide guarantees for control distortion (Basri et al., 2011; Ji et al., 2015). A complementary approach, analysed in Li & Gu (2018), directly projects subspaces using dense Gaussian operators to embed $\mathcal{G}(k, n)$ into a lower-dimensional Grassmannian $\mathcal{G}(k, m)$ while approximately preserving pairwise distances. Although theoretically appealing, these approaches rely on dense random matrices and incur $\mathcal{O}(n^2)$ storage or computation costs, limiting their applicability in high dimensions.

Binary and quantised measurements have also been explored for subspace estimation and search. In Wang et al. (2013), the authors propose a Grassmannian locality-sensitive hashing (LSH) scheme based on random one-dimensional subspaces, resulting in binary codes that preserve neighbourhood structure for approximate nearest-subspace search. More recently, Chi & Fu (2017) introduce a simple sensing framework for recovering a principal subspace from binary measurements, demonstrating that accurate subspace estimation is possible without transmitting full data vectors. These works highlight the effectiveness of random angular comparisons and quantisation, though their focus lies on retrieval or subspace recovery rather than kernel approximation.

Closest in spirit to the present work is Wang et al. (2018) who propose random projection operators for fast nearest-subspace search, probing subspaces using random directions to construct compact summaries. While conceptually related to our use of random rank-one projections, their goal is efficient retrieval rather than the construction of random features whose inner products approximate Grassmannian kernels with theoretical guarantees.

In contrast to existing approaches, we combine kernel methods with carefully designed rank-one random projections to construct scalable random feature maps for Grassmannian data. Building upon preliminary binary sketching ideas introduced in Delogne & Jacques (2025), we develop bounded random feature constructions that yield well-defined, positive-definite Grassmannian kernels depending only on principal angles, and establish uniform approximation guarantees. These properties allow kernel-based learning on subspaces at a computational and memory cost that scales linearly with the ambient dimension, enabling practical large-scale applications.

## 8 Experiments

In this section we demonstrate experimentally the results developed in theory in the previous section. We start with some experiments to confirm the results in expectation of our kernels. We then give a basic analysis of the efficient random embedding methods introduced in section 6. Finally, we demonstrate on a real dataset how our kernel approximations can be used in a real setting.

### 8.1 Dense projections versus rank-one projections

[**R-FSmh**] As a first analysis, we want to stress the computational advantage of using (unbounded) rank-one projections instead of dense Gaussian projections in the objective of approximating the projection kernel $\kappa^{\mathrm{p}}$. Indeed, the related empirical kernels both approximate $\kappa^{\mathrm{p}}$, but their computational and memory costs differ substantially, since dense projections require storing $m$ full $n \times n$ matrices while ROP features only

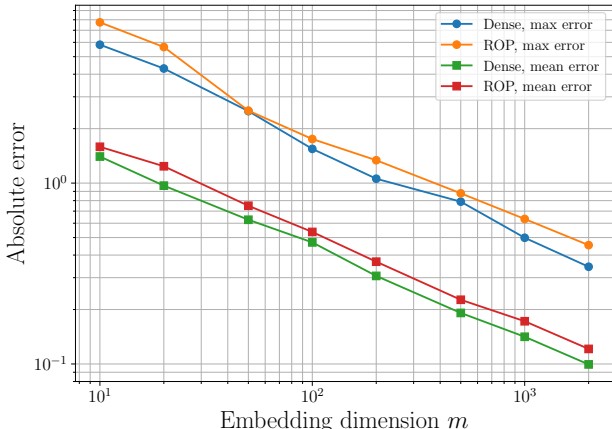

Figure 2: [**R-FSmh**] Approximation of the projection kernel with $\widehat{\kappa}^{\mathrm{d}}$ and $\widehat{\kappa}^{\mathrm{rop}}$ on $\mathcal{G}(5, 100)$. Both empirical kernels target $\kappa^{\mathrm{p}}$. Both approximations converge, although dense projections are slightly more accurate, but at the cost of a much higher computational and memory cost.

require $m$ pairs of vectors in $\mathbb{R}^n$. We remind that, as stated in Sec. 3.2, while unbounded ROPs allow for an approximation of the $\kappa^{\mathrm{p}}$ on finite datasets, they are theoretically not optimal when one targets a uniform characterization of this kernel over the whole space $\mathcal{G}(k, n)$.

We generate 100 pairs of subspaces in $\mathcal{G}(5, 100)$ with pre-selected principal angles. For each value of $m$, we compare the approximated $\widehat{\kappa}^{\mathrm{d}}$ and $\widehat{\kappa}^{\mathrm{rop}}$ with the exact projection kernel $\kappa^{\mathrm{p}}$. The experiment is repeated five times, and Fig. 2 shows the average mean and maximum absolute errors of these approximations.

Both empirical kernels converge to $\kappa^{\mathrm{p}}$ as $m$ increases. Dense projections are slightly more accurate, however, this comes at a much larger computational and memory cost. In this setting, storing dense matrices requires $mn^2$ real values, whereas storing ROP vectors requires only $2mn$. For $m = 2000$, this makes about $\widehat{\kappa}^{\mathrm{d}}$ $50\times$ less efficient than $\widehat{\kappa}^{\mathrm{rop}}$ in terms of memory and about $125\times$ slower in terms of execution time. This illustrates the trade-off between a slight accuracy gain and computational efficiency, thus motivating the use of rank-one projections.

These observations confirm the computational interest of using ROPs for kernel approximations.

## 8.2 Analysis of kernels

We now illustrate the geometry of the periodic Grassmann kernel in the simple case $k = 2$. Using the closed form expression, we evaluate $\kappa^{\circ}$ on a uniform grid of principal angles $(\theta_1, \theta_2) \in [0, \pi/2]^2$ and display the resulting $200 \times 200$ heatmaps for three values of $\omega \in \{0.5, 1, 2\}$.

We also execute the same experiment using the exact projection kernel $\kappa^{\mathrm{p}}$, the Binet-Cauchy kernel $\kappa^{\mathrm{BC}}$ and the empirical sign kernel $\kappa^{\pm}$, evaluated with the formula of Prop. 6. The corresponding heat maps are illustrated in the lower row of Figure 3. We see that all four kernels share the property of being large when the principal angles are small but their behaviour varies slightly as shown by the level curves. $\kappa^{\mathrm{BC}}$ and $\kappa^{\circ}$ with large $\omega$ show more sensitivity for small angles, $\kappa^{\mathrm{p}}$, $\kappa^{\pm}$ and $\kappa^{\circ}$ with reasonable $\omega$ are well spread, and $\kappa^{\circ}$ with small $\omega$ shows a lot less sensitivity to large angles. This confirms that $\kappa^{\circ}$ gives a trade-off between a locally and a globally sensitive similarity metric on $\mathcal{G}(2, n)$.

## 8.3 Analysis of the structured embedding

In this experiment, we investigate how the depth $S$ of the structured Hadamard–diagonal sketch influences the behaviour of individual rows of the resulting random projection matrix. For a fixed dimension $n$ and a fixed row index $j$, we consider the random vector $\boldsymbol{u} = \sqrt{n}\, \boldsymbol{e}_j^\top \boldsymbol{G}$, where $\boldsymbol{G}$ is generated according to the model in (17), and $\boldsymbol{e}_j$ denotes the $j$-th canonical basis vector of $\mathbb{R}^n$. For increasing values of the number

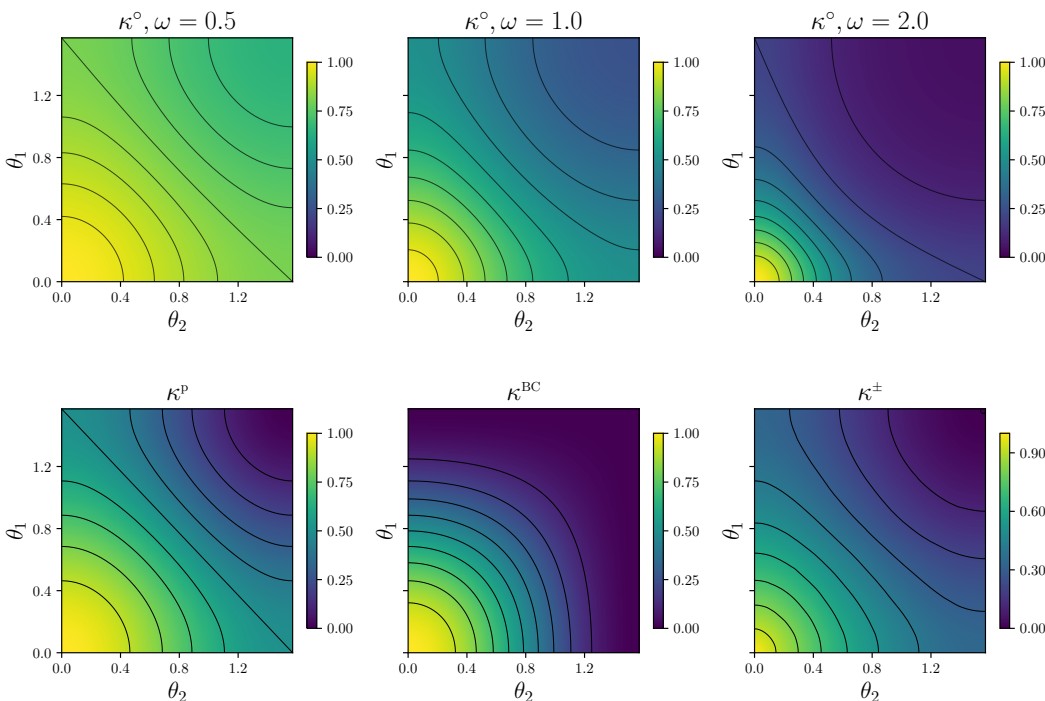

Figure 3: Top: geometry of the periodic Grassmannian kernel $\kappa^\circ$ for $k = 2$ for various $\omega$. Bottom: comparison between the projection, Binet-Cauchy and sign kernels (all normalised to $[0, 1]$) with $k = 2$.

of Hadamard blocks $S$, we examine the empirical distribution of the entries of $\boldsymbol{u}$, which corresponds to the distribution of the coefficients of a single row of the structured sketching matrix.

Figure 4 shows both the empirical histograms of the row entries (top) and the corresponding Q-Q plots against a standard Gaussian (bottom). For $S = 1$, the distribution is clearly non-Gaussian. As $S$ increases, the empirical distributions progressively smooth out and become increasingly symmetric, while the Q-Q plots approach the identity line. Already at $S = 3$, the row-wise distribution closely matches a Gaussian.

These observations empirically support the use of $S = 3$ Hadamard-diagonal blocks to generate structured random projections with an approximately Gaussian distribution. This corroborates earlier findings on structured matrices (Choromanski et al., 2016; Bojarski et al., 2017). In all subsequent experiments, we will therefore fix $S = 3$ unless otherwise stated.

## 8.4 ETH-80 Classification

We evaluate our methods on the ETH-80 dataset, which consists of 80 objects grouped into 8 *superclasses* (*e.g.*, apple, car, cow), each object being captured from 41 viewpoints under moderate illumination variations. Figure 5 illustrates the hierarchical structure of the dataset, with multiple images per object and multiple objects per superclass. All images are resized to $32 \times 32$ pixels, converted to greyscale, so that $n = 32 \times 32 = 1024$.

[**R-FSmh**] This experiment is an image-set classification problem, in line with other subspace classification experiments (Hamm & Lee, 2008; Ji et al., 2015; Wei et al., 2020). Each instance to be classified is a *group* of images of the same object acquired under different camera poses or illumination conditions. This group is then represented by the subspace spanned by the corresponding observations.

More precisely, given a collection of images associated with one object, we collect them into a matrix $\boldsymbol{X} \in \mathbb{R}^{n \times N_i}$ and extract an orthonormal basis $\boldsymbol{U} \in \mathbb{R}^{n \times k}$ by retaining the $k$ leading left singular vectors of $\boldsymbol{X}$. Unless otherwise stated, we fix $k = 9$, in line with the empirical observations reported in (Ji et al., 2015).

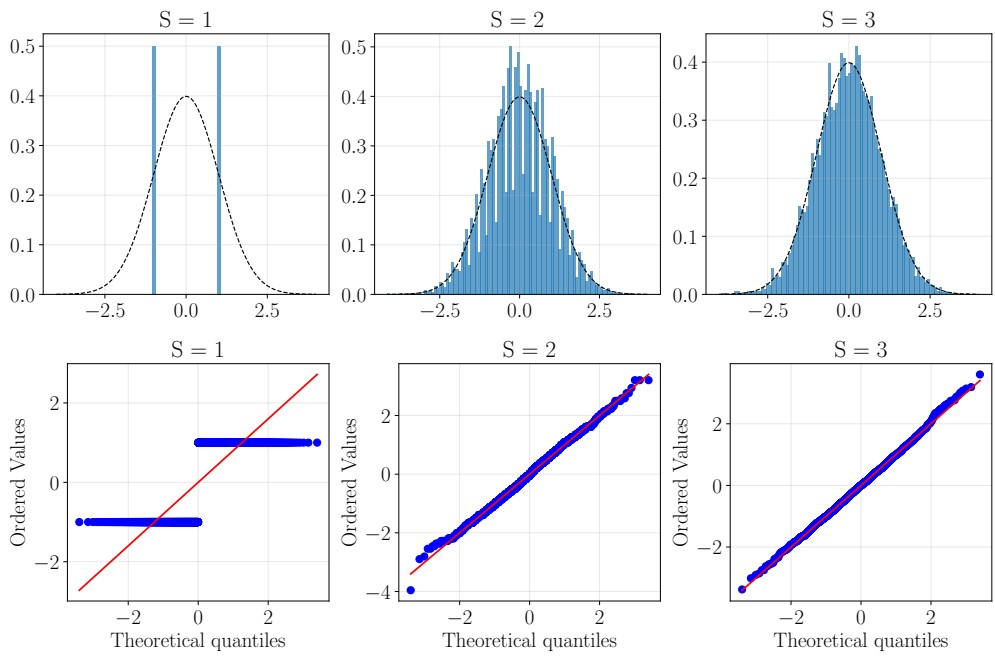

Figure 4: **Top:** Empirical distribution of a row of the structured Hadamard–diagonal sketch for increasing numbers of Hadamard blocks $S$. **Bottom:** Q–Q plots comparing the row-wise distribution to a standard Gaussian.

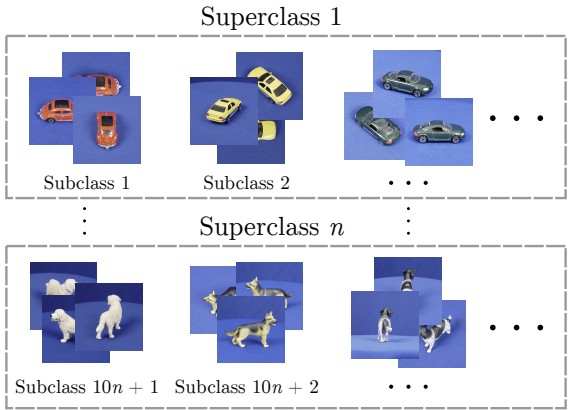

Figure 5: Structure of the ETH-80 dataset

The resulting classification problem is therefore a classification problem between subspace-valued instances. It does not address the separate problem of assigning a single isolated image to a subspace.

We consider two classification settings of increasing difficulty. In the first one, *superclass (8-way) classification*, each object is represented by a single subspace, but labels are given at the superclass level. This setting results in a small number of subspaces and is useful to assess the approximation quality of the proposed approximated kernels. In the second one, *object (80-way) classification*, each object constitutes its own class. To increase the number of available subspaces and stress the computational aspects of kernel evaluation, multiple subspaces are generated per object by subsampling images.

For both settings, we compare the exact projection kernel $\kappa^{\mathrm{p}}$ to its random approximation $\widehat{\kappa}^{\mathrm{rop}}$, evaluate the binary kernel $\widehat{\kappa}^{\pm}$ (for which we do not have a closed-form $\kappa^{\pm}$ available), and compare the periodic kernel

$\widehat{\kappa}^\circ$ to its exact version $\kappa^\circ$. All experiments are further repeated with the structured variants of the rank-one projections introduced in Sec. 6.

**[R-FSmh]** Recall that $\widehat{\kappa}^{\text{rop}}$ targets the exact projection kernel $\kappa^{\text{p}}$, whereas the binary and periodic constructions induce the kernels $\kappa^\pm$ and $\kappa^\circ$, respectively. Therefore, differences in classification accuracy may reflect both the quality of the random approximation and the suitability of the corresponding Grassmannian kernel for the considered dataset. Our goal is not to show that bounded ROP features always outperform unbounded ROP features in finite-data classification, but instead to show that they provide light random features associated with kernels for which stronger uniform approximation guarantees can be computed.

### 8.4.1 Superclass (8-way) classification

In this first setting, we perform superclass classification on ETH-80, following a similar experiment as (Wei et al., 2020). Each object is represented by a single $k$-dimensional subspace, while labels are assigned at the superclass level. For each superclass, we randomly select 7 objects for training and 3 for testing. This makes a total of 56 training subspaces and 24 test subspaces. Subspaces are constructed as described above, keeping $k = 9$ dimensions. Test subspaces are built independently from the corresponding test objects.

We begin by evaluating the exact projection kernel $\kappa^{\text{p}}$ and the exact periodic kernel $\kappa^\circ$, both of which reach perfect classification accuracy on this task. This confirms that the problem is well suited to subspace-based representations and kernel methods. We then replace these exact kernels with their approximated versions using both unstructured and structured variants.

To compare embedding sizes across methods, we show the results as a function of the *oversampling ratio* defined as $\rho = \frac{m}{nk}$, where $m$ denotes the embedding dimension. Small values of $\rho$ correspond to compressed representations, $\rho \approx 1$ to embeddings of comparable size to the original basis, and $\rho \gg 1$ to overcomplete embeddings. In the following experiments, we focus on $\rho = 0.05$ and $\rho = 0.20$, which already provide good accuracy results.

Figure 6a shows the classification accuracy as a function of $\rho$, while Tab. 2 summarises runtimes and accuracies for the two selected values of $\rho$. As expected, increasing $\rho$ improves the accuracy of all random embedding methods, with unstructured embeddings converging quickly towards the performance of the exact kernels. Structured embeddings show the same behaviour, although requiring larger embedding sizes to reach similar accuracy, but with more compact representations. **[R-FSmh]** It is important to keep in mind that different random embeddings do not approximate the same kernel. Unbounded ROP features approximate the projection kernel $\kappa^{\text{p}}$, whereas binary and periodic ROP features approximate or induce the kernels $\kappa^\pm$ and $\kappa^\circ$. Therefore, differences in accuracy reflects both the quality of the random approximation and the suitability of the corresponding kernel for the dataset and task considered.

From a computational perspective, this setting is too small for random embeddings to fully show their strength. With only 56 training subspaces, exact kernel computation is already fast, and unstructured embeddings do not yet accelerate the procedure. Structured embeddings are consistently faster than unstructured ones, but the overall gains remain modest in this setting. This experiment is therefore primarily a validation of the approximation behaviour of the proposed kernels. The computational benefits of random embeddings will become apparent when the number of subspaces increases, as shown in the next section.

### 8.4.2 Object (80-way) classification

We now turn to the object classification task, in which each individual object constitutes its own class. This setting is significantly more complex, both in terms of classification difficulty and computational cost, and is well-suited to highlight the advantages of the approximated kernels.

For each object, the 41 images are randomly split into 28 training images and 13 testing images. From the training images, we randomly select 15 images to construct a $k$-dimensional subspace, keeping $k = 9$ dimensions. This procedure is repeated 10 times per object, yielding 10 training subspaces in $\mathcal{G}(9, 1024)$ for each of the 80 objects. At test time, a single subspace per object is built from the remaining 13 images, still with $k = 9$. This ensures that no test image contributes in any way to the construction of the training subspaces.

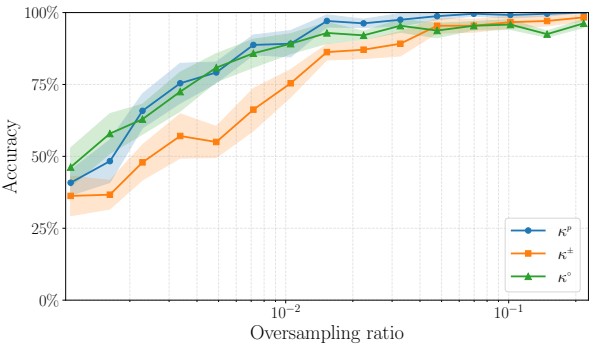
(a) Accuracy vs $m$ for superclass classification.

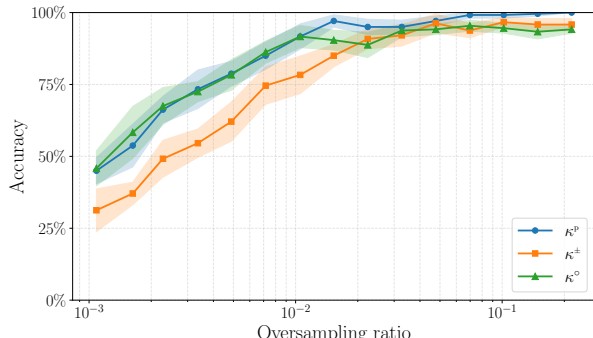
(b) Structured embedding with Fourier embedding with $S = 3$ and varying $m$

Figure 6: ETH-80 superclass classification: (a) unstructured embedding and (b) structured embedding.

| Method | $\rho$ | Total time (s) | Accuracy (%) |
|---|---|---|---|
| **Exact kernels (no embedding)** | | | |
| Projection kernel $\kappa^{\mathrm{p}}$ | — | 0.2633 | 100.00 |
| Periodic kernel $\kappa^{\circ}$ | — | 0.2381 | 100.00 |
| **Random embeddings at two oversampling ratios** | | | |
| ROP | 0.05 / 0.20 | 1.3211 / 5.3102 | 98.12 / 99.79 |
| Binary ROP | 0.05 / 0.20 | 1.3211 / 5.3111 | 93.12 / 96.04 |
| Periodic ROP | 0.05 / 0.20 | 1.3240 / 5.3145 | 93.54 / 93.75 |
| Structured | 0.05 / 0.20 | 0.1046 / 0.1963 | 94.79 / 93.54 |
| Structured Binary | 0.05 / 0.20 | 0.1048 / 0.1972 | 92.29 / 96.46 |
| Structured Periodic | 0.05 / 0.20 | 0.1055 / 0.2025 | 94.79 / 93.54 |

Table 2: Superclass classification. Total time includes embedding computation and SVM training/evaluation. Results averaged over 20 runs with different random embeddings.

As in the previous setting, we first evaluate the exact projection kernel $\kappa^{\mathrm{p}}$ and the exact periodic kernel $\kappa^{\circ}$. While both kernels achieve strong classification performance, the cost of computing the corresponding kernel matrices now becomes substantial due to the large number of subspaces involved, with computation times close to one minute under our experimental conditions.

We again evaluate the approximated versions of these kernels, using both unstructured and structured embeddings. Classification accuracy as a function of the oversamplingratio ratio $\rho$ is shown in Fig. 7, with unstructured embeddings shown in Fig. 7a and structured embeddings in Fig. 7b. Runtimes and accuracies at the same oversampling ratios $\rho = 0.05$ and $\rho = 0.20$ as in the superclass setting are summarised in Tab. 3.

The accuracy trends are consistent with those observed previously: increasing $\rho$ improves performance for all random methods, although larger embedding sizes are required in this more complex setting to approach the accuracy of the exact kernels. Unstructured embeddings eventually get to the performance of the exact projection kernel, but at the cost of increased computation time, which can exceed that of the deterministic kernels.

In contrast, the structured embeddings reach excellent classification accuracy while being substantially faster than the exact kernels. In particular, for $\rho = 0.05$ and $\rho = 0.20$, structured embeddings run much faster compared to exact kernel computation. This experiment clearly demonstrates the computational advantages of structured random embeddings in large-scale Grassmannian kernel methods.

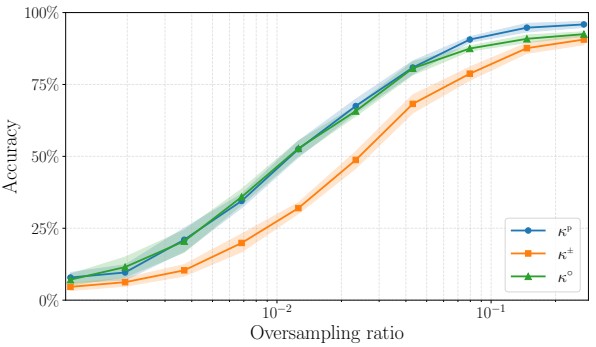 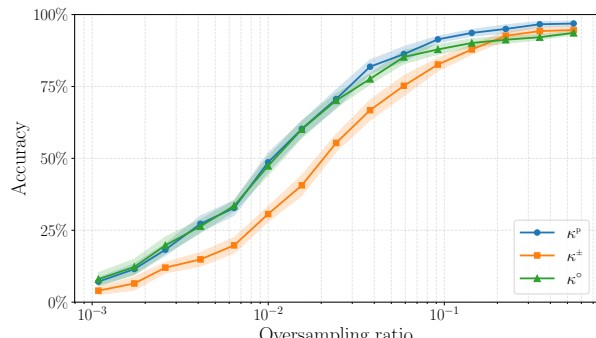

(a) Accuracy vs $m$ for sub-class classification with unstructured embedding

(b) Accuracy vs $m$ for sub-class classification with structured embedding and $S = 3$

Figure 7: ETH-80 object (80-way) classification: (a) unstructured embedding and (b) structured embedding.

| Method | $\rho$ | Total time (s) | Accuracy (%) |
|---|---|---|---|
| **Exact kernels (no embedding)** | | | |
| Projection kernel $\kappa^\mathrm{p}$ | — | 52.8856 | 98.75 |
| Periodic kernel $\kappa^\circ$ | — | 58.7874 | 90.00 |
| **Sketched methods at two oversampling ratios** | | | |
| ROP | 0.05 / 0.20 | 24.1868 / 85.2087 | 84.19 / 94.06 |
| Binary ROP | 0.05 / 0.20 | 24.2049 / 85.2449 | 72.25 / 91.75 |
| Periodic ROP | 0.05 / 0.20 | 24.3763 / 86.4142 | 81.38 / 92.31 |
| Structured | 0.05 / 0.20 | 1.8729 / 4.0293 | 83.00 / 92.38 |
| Structured Binary | 0.05 / 0.20 | 1.8925 / 4.0837 | 72.13 / 91.25 |
| Structured Periodic | 0.05 / 0.20 | 1.9967 / 5.2383 | 83.00 / 92.38 |

Table 3: Sub-class classification. Total time includes sketch computation and SVM training/evaluation. Results averaged over 20 runs with different random sketches.

## 9 Conclusion

In this work, we proposed random feature methods for approximating Grassmannian kernels, both from the theoretical and computational standpoint. We started from the observation that dense random projections are impractical in high dimensions due to their memory and computational costs and introduced asymmetric rank-one projections (ROP) as a lightweight alternative for constructing kernel approximations on the Grassmannian manifold.

Since ROP measurements have heavy tails, we combined them with bounded non-linear functions to obtain stable kernel estimators with controlled concentration. We first focused on the sign function that led to a compact representation with statistical guarantees, although it does not admit a closed-form expression for the associated kernel. We also discussed how the complex exponential function yielded a new closed-form Grassmannian kernel. For both constructions, we provided approximation guarantees with explicit control of the probability of failure uniformly over all pairs of subspaces.

We further demonstrated the practical relevance of these random embeddings through experiments on a real-world image dataset. Both the basic ROP and its sign and periodic variants achieved high classification accuracy using embeddings as small as 5% of the original subspace representation size. Using structured random mappings enabled additional computational gains while preserving accuracy, making these methods particularly well suited for kernel-based classification with random features in large-scale settings.

[**R-FSmh**] Several directions are still open for future work. From a theoretical perspective, a better understanding of the good behaviour of unbounded ROP in a finite-dataset context, despite its heavy-tailed nature and weaker uniform guarantees over the full Grassmannian, would be of interest. Another natural direction

is the study of alternative bounded non-linear functions leading to potentially new kernel approximations. The structured embedding strategy also calls for a dedicated theoretical analysis to better characterise its performance. Finally, optical computing offers another promising way to further accelerate these random projections by implementing Gaussian-like rank-one measurements directly in hardware (Saade et al., 2016).

# A    Proof of Prop. 9

Since $\psi^{\pm} = \text{sign} \circ \psi^{\text{rop}}$ is discontinuous, it is not straightforward to prove a uniform bound on the error of the approximation $\widehat{\kappa}^{\pm} \approx \kappa^{\pm}$, *i.e.*, Prop. 9. We thus need to study the stability of $\psi^{\pm}$ in different ways, *e.g.*, we need to know the probability that it could be discontinuous within a small neighbourhood. Let us first observe what is the stability of a related random mapping applied to vectors, *i.e.*, $\boldsymbol{x} \in \mathbb{R}^n \mapsto \text{sign}(\boldsymbol{a}^{\top}\boldsymbol{x})$.

**Lemma 17.** *Let $\mathcal{C}_{\alpha}(\boldsymbol{u}) = \{\boldsymbol{x} \in \mathbb{R}^n : \measuredangle(\boldsymbol{x}, \boldsymbol{u}) \leqslant \alpha\}$ be a cone of axis $\boldsymbol{u} \in \mathbb{S}^{n-1}$ and half aperture $0 \leqslant \alpha \leqslant \pi$. Given the random map $\bar{\psi} : \boldsymbol{x} \in \mathbb{R}^n \mapsto \text{sign}(\boldsymbol{a}^{\top}\boldsymbol{x})$ with $\boldsymbol{a} \sim \mathcal{N}(\boldsymbol{0}, \boldsymbol{I}_n)$, we have $\mathbb{P}[\bar{\psi} \notin \mathsf{C}^0(\mathcal{C}_{\alpha}(\boldsymbol{u}))] = 1$ if $\alpha > \pi/2$ and*

$$\mathbb{P}[\bar{\psi} \notin \mathsf{C}^0(\mathcal{C}_{\alpha}(\boldsymbol{u}))] \leqslant \sqrt{\tfrac{2}{\pi}}(\tan \alpha)\sqrt{n}, \quad if \; 0 < \alpha \leqslant \pi/2,$$

*where $\mathsf{C}^0(\mathcal{S})$ denotes continuous functions on a set $\mathcal{S}$.*

*Proof.* The proof is adapted from (Tachella & Jacques, Corollary 12). We first observe that $\bar{\psi}$ is discontinuous over $\mathcal{C}_{\alpha}(\boldsymbol{u})$ iff $|\boldsymbol{a}^{\top}\boldsymbol{u}| \leqslant \epsilon\|\boldsymbol{a}\|$ with $\epsilon := \sin \alpha$. Therefore, by the rotational invariance of the Gaussian distribution we can choose $\boldsymbol{u} = (1, 0, \ldots, 0)^{\top}$ and the probability above amounts to computing

$$p := \mathbb{P}[|a_1|^2 \leqslant \epsilon^2\|\boldsymbol{a}\|^2] = \mathbb{P}[a_1^2 \leqslant \tfrac{\epsilon^2}{(1-\epsilon^2)}(a_2^2 + \ldots + a_n^2)] = \mathbb{E}_{\xi}\mathbb{P}[a_1^2 \leqslant \tfrac{\epsilon^2}{(1-\epsilon^2)}\xi|\xi],$$

where $\xi \sim \chi^2(n-1)$ is independent of $a_1$. This gives

$$\mathbb{P}[a_1^2 \leqslant \tfrac{\epsilon^2}{(1-\epsilon^2)}\xi|\xi] \leqslant \sqrt{\tfrac{2}{\pi}}\tfrac{\epsilon}{\sqrt{1-\epsilon^2}}\sqrt{\xi} = \sqrt{\tfrac{2}{\pi}}(\tan \alpha)\sqrt{\xi}$$

Observing that $\mathbb{E}_{\xi}\sqrt{\xi} \leqslant \sqrt{\mathbb{E}_{\xi}\xi} \leqslant \sqrt{n-1} \leqslant \sqrt{n}$ by Jensen's inequality gives the result. $\square$

For this result, we can show that, when the projectors of two subspaces are close in Frobenius norm, each component of the binary random features related to $\psi^{\pm}$ coincide with high probability.

**Lemma 18** (Local stability of the binary embedding). *We consider the subspace $\boldsymbol{P}^* \in \mathbb{G}(k, n)$ and the random vectors $\boldsymbol{a}, \boldsymbol{b} \sim \mathcal{N}(\boldsymbol{0}, \boldsymbol{I}_n)$. Define the map $\bar{\psi} : \boldsymbol{M} \in \mathbb{R}^{n \times n} \mapsto \text{sign}(\boldsymbol{a}^{\top}\boldsymbol{M}\boldsymbol{b})$, the ball $\mathbb{B}_{\epsilon}^F(\boldsymbol{M}^*) = \{\boldsymbol{M} \in \mathbb{R}^{n \times n} : \|\boldsymbol{M} - \boldsymbol{P}^*\|_F \leqslant \epsilon\}$ of radius $\epsilon > 0$ and centred on $\boldsymbol{M}^* \in \mathbb{R}^{n \times n}$, and the neighbourhood $\mathcal{N}_{\epsilon}(\boldsymbol{P}^*) := \mathbb{B}_{\epsilon}^F(\boldsymbol{P}^*) \cap \mathbb{G}(k, n)$. Then,*

$$\mathbb{P}\Big[\exists \boldsymbol{P}, \boldsymbol{Q} \in \mathcal{N}_{\epsilon}(\boldsymbol{P}^*) : \bar{\psi}(\boldsymbol{P}) \neq \bar{\psi}(\boldsymbol{Q})\Big] = \mathbb{P}\Big[\bar{\psi} \notin \mathsf{C}^0(\mathcal{N}_{\epsilon}(\boldsymbol{P}^*))\Big] \leqslant 4\tfrac{n+k}{\sqrt{k}}[\boldsymbol{EiC}]\epsilon^{\frac{k}{k+1}}.$$

*Proof.* First, we can assume $\boldsymbol{P}^* = \text{bdiag}(\boldsymbol{I}_k, \boldsymbol{0}_{n-k \times n-k})$ from the rotational invariance of $\boldsymbol{a}$ and $\boldsymbol{b}$. Second, we note that since $\bar{\psi}$ is the sign of a linear functional, it fails to be continuous on $\mathcal{N}_{\epsilon}(\boldsymbol{P}^*)$ if and only if there exist $\boldsymbol{P}, \boldsymbol{Q} \in \mathcal{N}_{\epsilon}(\boldsymbol{P}^*)$ such that $\bar{\psi}(\boldsymbol{P}) \neq \bar{\psi}(\boldsymbol{Q})$. Therefore,

$$p_{\epsilon} := \mathbb{P}[\bar{\psi} \notin \mathsf{C}^0(\mathcal{N}_{\epsilon}(\boldsymbol{P}^*))] = \mathbb{P}[\exists \boldsymbol{P}, \boldsymbol{Q} \in \mathcal{N}_{\epsilon}(\boldsymbol{P}^*) : \bar{\psi}(\boldsymbol{P}) \neq \bar{\psi}(\boldsymbol{Q})]$$
$$= \mathbb{P}[\exists \boldsymbol{P}, \boldsymbol{Q} \in \mathcal{N}_{\epsilon}(\boldsymbol{P}^*) : \text{sign}(\boldsymbol{a}^{\top}(\boldsymbol{P}\boldsymbol{b})) \neq \text{sign}(\boldsymbol{a}^{\top}(\boldsymbol{Q}\boldsymbol{b}))].$$

Given a parameter $\rho > 0$ to be fixed later, let us define the event

$$\mathcal{E}_{\rho} = \{\|\boldsymbol{P}^*\boldsymbol{b}\| > \rho\|\boldsymbol{b}\|\},$$

which also defines $\boldsymbol{b}'$, the distribution of $\boldsymbol{b}|\mathcal{E}_\rho$. Therefore, by the laws of total probability and expectations,

$$
\begin{aligned}
p_\epsilon &= \mathbb{P}[\exists \boldsymbol{P}, \boldsymbol{Q} \in \mathcal{N}_\epsilon(\boldsymbol{P}^*) : \mathrm{sign}(\boldsymbol{a}^\top(\boldsymbol{P}\boldsymbol{b})) \neq \mathrm{sign}(\boldsymbol{a}^\top(\boldsymbol{Q}\boldsymbol{b}))|\mathcal{E}_\rho]\,\mathbb{P}[\mathcal{E}_\rho] \\
&\quad + \mathbb{P}[\exists \boldsymbol{P}, \boldsymbol{Q} \in \mathcal{N}_\epsilon(\boldsymbol{P}^*) : \mathrm{sign}(\boldsymbol{a}^\top(\boldsymbol{P}\boldsymbol{b})) \neq \mathrm{sign}(\boldsymbol{a}^\top(\boldsymbol{Q}\boldsymbol{b}))|\mathcal{E}_\rho^c]\,\mathbb{P}[\mathcal{E}_\rho^c] \\
&\leqslant \mathbb{P}[\exists \boldsymbol{P}, \boldsymbol{Q} \in \mathcal{N}_\epsilon(\boldsymbol{P}^*) : \mathrm{sign}(\boldsymbol{a}^\top(\boldsymbol{P}\boldsymbol{b})) \neq \mathrm{sign}(\boldsymbol{a}^\top(\boldsymbol{Q}\boldsymbol{b}))|\mathcal{E}_\rho]\ +\ \mathbb{P}[\mathcal{E}_\rho^c] \\
&= \mathbb{E}_{\boldsymbol{b}'}\mathbb{P}_{\boldsymbol{a}}[\exists \boldsymbol{P}, \boldsymbol{Q} \in \mathcal{N}_\epsilon(\boldsymbol{P}^*) : \mathrm{sign}(\boldsymbol{a}^\top(\boldsymbol{P}\boldsymbol{b}')) \neq \mathrm{sign}(\boldsymbol{a}^\top(\boldsymbol{Q}\boldsymbol{b}'))|\boldsymbol{b}']\ +\ \mathbb{P}[\mathcal{E}_\rho^c].
\end{aligned}
$$

However, if $\boldsymbol{P} \in \mathcal{N}_\epsilon(\boldsymbol{P}^*)$, then the vectors $\boldsymbol{\beta} = \boldsymbol{P}\boldsymbol{b}'$ and $\boldsymbol{\beta}^* = \boldsymbol{P}^*\boldsymbol{b}'$ are at most $\epsilon' := \epsilon\|\boldsymbol{b}'\|$ far apart since $\|\boldsymbol{\beta} - \boldsymbol{\beta}^*\| = \|(\boldsymbol{P} - \boldsymbol{P}^*)\boldsymbol{b}'\| \leqslant \|\boldsymbol{P} - \boldsymbol{P}^*\|_F\|\boldsymbol{b}'\| \leqslant \epsilon\|\boldsymbol{b}'\|$, while, similarly, $\boldsymbol{\gamma} = \boldsymbol{Q}\boldsymbol{b}'$ is at most $\epsilon'$ far apart from $\boldsymbol{\beta}^*$. Therefore,

$$
\begin{aligned}
p_\epsilon &\leqslant \mathbb{E}_{\boldsymbol{b}'}\mathbb{P}_{\boldsymbol{a}}[\exists \boldsymbol{\beta}, \boldsymbol{\gamma} \in \mathbb{B}^2_{\epsilon'}(\boldsymbol{\beta}^*) : \mathrm{sign}(\boldsymbol{a}^\top\boldsymbol{\beta}) \neq \mathrm{sign}(\boldsymbol{a}^\top\boldsymbol{\gamma})|\boldsymbol{b}']\ +\ \mathbb{P}[\mathcal{E}_\rho^c] \\
&\leqslant \mathbb{E}_{\boldsymbol{b}'}\mathbb{P}_{\boldsymbol{a}}[\bar\psi \notin \mathsf{C}^0(\mathbb{B}^2_{\epsilon'}(\boldsymbol{\beta}^*))|\boldsymbol{b}']\ +\ \mathbb{P}[\mathcal{E}_\rho^c],
\end{aligned}
$$

where $\bar\psi : \boldsymbol{x} \in \mathbb{R}^n \mapsto \mathrm{sign}(\boldsymbol{a}^\top\boldsymbol{x})$, and $\mathbb{B}^2_{\epsilon'}(\boldsymbol{u})$ is the $\ell_2$-ball of radius $\epsilon'$ around a vector $\boldsymbol{u}$.

Let us observe that $\mathbb{B}^2_{\epsilon'}(\boldsymbol{\beta}^*)$ is contained in a cone with a half aperture $\alpha$ respecting $\sin\alpha = \epsilon'/\|\boldsymbol{\beta}^*\| = \epsilon\|\boldsymbol{b}'\|/\|\boldsymbol{P}^*\boldsymbol{b}'\| \leqslant \epsilon/\rho$, since the support of $\boldsymbol{b}'$ is fixed by $\mathcal{E}_\rho$. Therefore, from Lem. 17, provided that $\epsilon < \rho$, we have

$$
\mathbb{E}_{\boldsymbol{b}'}\mathbb{P}_{\boldsymbol{a}}[\bar\psi \notin \mathsf{C}^0(\mathbb{B}^2_{\epsilon'}(\boldsymbol{\beta}^*))|\boldsymbol{b}'] \leqslant \mathbb{E}_{\boldsymbol{b}'}\sqrt{\tfrac{2}{\pi}}\frac{\epsilon\rho^{-1}}{(1-\epsilon^2\rho^{-2})^{1/2}}\sqrt{n} = \sqrt{\tfrac{2}{\pi}}\frac{\epsilon\rho^{-1}}{(1-\epsilon^2\rho^{-2})^{1/2}}\sqrt{n}.
$$

Let us now bound $\mathbb{P}[\mathcal{E}_\rho^c]$. Using the structure of $\boldsymbol{P}^*$, we have

$$
\mathbb{P}[\mathcal{E}_\rho^c] = \mathbb{P}[\,\textstyle\sum_{i=1}^k b_i^2 \leqslant \rho^2\|\boldsymbol{b}\|^2] = \mathbb{P}[\,\textstyle\sum_{i=1}^k b_i^2 \leqslant \tfrac{\rho^2}{1-\rho^2}\sum_{i=k+1}^n b_i^2] = \mathbb{P}[X^2 \leqslant \tfrac{\rho^2}{1-\rho^2}Y^2],
$$

where $X^2$ and $Y^2$ are two independent $\chi^2$-distribution with $k$ and $n-k$ degrees of freedom, respectively. However, again by the law of total expectation

$$
\mathbb{P}[\mathcal{E}_\rho^c] = \mathbb{E}_{Y^2}\mathbb{P}_{X^2}[X^2 \leqslant \tfrac{\rho^2}{1-\rho^2}Y^2]. \tag{19}
$$

However, for any $\lambda > 0$, $\mathbb{P}(X^2 \leqslant \lambda^2) = (2^{\frac{k}{2}}\Gamma(\tfrac{k}{2}))^{-1}\int_0^{\lambda^2} t^{\frac{k}{2}-1}e^{-t/2}\mathrm{d}t$. Since $\int_0^{\lambda^2} t^{\frac{k}{2}-1}e^{-t/2}\mathrm{d}t \leqslant (2/k)\lambda^k$, this shows that $\mathbb{P}(X^2 \leqslant \lambda^2) \leqslant 2^{-\frac{k}{2}}(\Gamma(\tfrac{k}{2}+1))^{-1}\lambda^k$. With the Stirling bound, i.e., $\Gamma(\tfrac{k}{2}+1) \geqslant (\tfrac{k}{2e})^{\frac{k}{2}}$, we get $\mathbb{P}(X^2 \leqslant \lambda^2) \leqslant (\tfrac{e}{k})^{\frac{k}{2}}\lambda^k$. Therefore, injecting this bound in (19) with $\lambda = \frac{\rho}{\sqrt{1-\rho^2}}\sqrt{Y^2}$ to match (19), we get

$$
\mathbb{P}[\mathcal{E}_\rho^c] \leqslant (\tfrac{e}{k})^{k/2}(\tfrac{\rho^2}{1-\rho^2})^{k/2}\mathbb{E}(Y^2)^{k/2}.
$$

Since, by Jensen and from the moments of a $\chi^2_{n-k}$-distribution, $(\mathbb{E}(Y^2)^{k/2})^2 \leqslant \mathbb{E}(Y^2)^k = \prod_{j=1}^k(n+k-2j)$, we get the crude bound $\mathbb{E}(Y^2)^{k/2} \leqslant (n+k)^{k/2}$. Therefore

$$
\mathbb{P}[\mathcal{E}_\rho^c] \leqslant \left(\tfrac{e}{k}\right)^{k/2}\left(\tfrac{\rho^2}{1-\rho^2}\right)^{k/2}(n+k)^{k/2} = \left(\tfrac{e(n+k)\rho^2}{k(1-\rho^2)}\right)^{k/2}.
$$

We decide now to set [EiC]

$$
\rho^2 = \epsilon^{\frac{2}{k+1}}\frac{k}{3e(n+k)} < \tfrac{1}{3}.
$$

As observed above, we must impose the condition $\epsilon < \rho$. Since $k \geqslant 1$, our choice provides [EiC] $3\rho^2 \geqslant \epsilon^{\frac{2}{k+1}}k/(e(n+k))$, which shows that $\epsilon < \rho$ is met if $\epsilon^2 \leqslant \epsilon^{2/k}\frac{k}{3e(n+k)}$, satisfied with the stronger condition [EiC]

$$
\epsilon < \left(\frac{k}{9e(n+k)}\right)^{\frac{k+1}{2k}}. \tag{20}
$$

Therefore, with this setting we get

[EiC]

$$
\mathbb{P}[\mathcal{E}_\rho^c] \leqslant \left(\tfrac{e(n+k)\rho^2}{k(1-\rho^2)}\right)^{k/2} \leqslant \epsilon^{\frac{k}{k+1}}\left(\tfrac{1}{2}\right)^{k/2} \leqslant \epsilon^{\frac{k}{k+1}}.
$$

$\square$

A simple rescaling allows us to simplify the last lemma (the proof is left to the reader).

**Corollary 19** (Local stability of the binary embedding (Simplified))**.** *Under the conventions and conditions of Lem. 18, for $\delta > 0$, we have* **[EiC]**

$$\mathbb{P}\Big[\bar\psi \notin \mathsf{C}^0(\mathcal{N}_{\eta(\delta)}(\boldsymbol{P}^*))\Big] \leqslant \delta, \; \text{with } \eta(\delta) := \big(\tfrac{\sqrt{k}}{4(n+k)}\delta\big)^{\frac{k+1}{k}},$$

Lem. 18 and Cor. 19 give the probability of sign flip for one element in the measurement vector when $\boldsymbol{P}$ is moved around within $\epsilon$. The next lemma will further characterise the probability of having at most a certain number of flips in the full measurements vector for a small perturbation of $\boldsymbol{P}$.

**Lemma 20.** *Under the conventions introduced in Lem. 18, given an integer $m$, we consider $m$ random maps $\bar\psi_i : \boldsymbol{P} \in \mathbb{G}(k,n) \mapsto \mathrm{sign}(\boldsymbol{a}_i^\top \boldsymbol{P} \boldsymbol{b}_i)$ with $\boldsymbol{a}_i, \boldsymbol{b}_i \sim_{\text{i.i.d.}} \mathcal{N}(\boldsymbol{0}, \boldsymbol{I}_n)$, $1 \leqslant i \leqslant m$. Given $\delta > 0$, the function $\eta(\delta)$ defined in Cor. 19, and $\boldsymbol{P}^* \in \mathbb{G}(k,n)$, we have*

$$\mathbb{P}\Big[|\{i : \bar\psi_i \notin \mathsf{C}^0(\mathcal{N}_{\eta(\delta)}(\boldsymbol{P}^*))\}| > 2\delta m\Big] \leqslant C \exp(-c\delta^2 m),$$

*for absolute constants $C, c > 0$.*

*Proof.* For each $1 \leqslant i \leqslant m$, define the binary random variable $Z_i = \mathbb{1}[\bar\psi_i \notin \mathsf{C}^0(\mathcal{N}_{\eta(\delta)}(\boldsymbol{P}^*))]$, with $\mathbb{1}[\mathcal{E}]$ being equal to 1 if the event $\mathcal{E}$ is true, and to 0 otherwise. By Cor. 19, we have $p_\delta := \mathbb{E}Z_i \leqslant \delta$. The random variables $Z_i$ are independent and bounded, hence sub-Gaussian. Therefore, from (Vershynin, 2018, Prop. 2.6.1 and Theorem 2.6.2), for any $\rho > 0$,

$$\mathbb{P}[\textstyle\sum_{i=1}^m Z_i > \delta m + \rho m] \leqslant \mathbb{P}[\textstyle\sum_{i=1}^m Z_i \geqslant m p_\delta + \rho m] \leqslant C \exp(-c\rho^2 m),$$

for some absolute constants $C, c > 0$. Choosing $\rho = \delta$ yields

$$\mathbb{P}[\textstyle\sum_{i=1}^m Z_i > 2\delta m] \leqslant C \exp(-c\delta^2 m).$$

$\square$

In addition to the previous stability properties, our proof also uses the possibility to *cover* the space of projectors.

**Proposition 21** (Covering $\mathbb{G}(k,n)$)**.** *Given $\epsilon > 0$, there exists a covering $\mathcal{N}_\epsilon \subset \mathbb{G}(k,n)$ of $\mathbb{G}(k,n)$, i.e., such that*

$$\forall \boldsymbol{P} \in \mathbb{G}(k,n), \; \exists \boldsymbol{P}^* \in \mathcal{N}_\epsilon : \; \|\boldsymbol{P} - \boldsymbol{P}^*\|_F \leqslant \epsilon,$$

*whose cardinality is bounded by $|\mathcal{N}_\epsilon| \leqslant (18\sqrt{k}/\epsilon)^{(2n+1)k}$.*

*Proof.* The set $\mathbb{G}(k,n)$ is included in the set $\mathcal{R}^F(k,n) := \mathcal{R}(k,n) \cap \sqrt{k}\mathbb{B}_F^{n \times n}$ of $n \times n$ rank-$k$ matrices with Frobenius norm bounded by $\sqrt{k}$ (since if $\boldsymbol{P} \in \mathbb{G}(k,n)$, $\mathrm{rank}(\boldsymbol{P}) = k$ and $\|\boldsymbol{P}\|_F = \sqrt{k}$), one can cover $\mathbb{G}(k,n)$ from a covering of $\mathcal{R}^F(k,n)$. Indeed, given two compact sets $\mathcal{A} \subset \mathcal{B}$ in a metric space $(\mathcal{X}, d)$ with distance $d$, if $\mathcal{B}_{\epsilon/2} \subset \mathcal{B}$ is an $\epsilon/2$-covering of $\mathcal{B}$, *i.e.*, such that for any $\boldsymbol{x} \in \mathcal{B}$ there is a $\boldsymbol{x}' \in \mathcal{B}_{\epsilon/2}$ such that $d(\boldsymbol{x}, \boldsymbol{x}') \leqslant \epsilon/2$, then one can build an $\epsilon$-covering $\mathcal{A}_\epsilon \subset \mathcal{A}$ of $\mathcal{A}$ with $|\mathcal{A}_\epsilon| \leqslant |\mathcal{B}_{\epsilon/2}|$ (see *e.g.*, (Vershynin, 2018, Chap. 4)). From (Candès & Plan, 2011, Lem. 3.1), there exists an $\epsilon/2$-covering of $\mathcal{R}(k,n) \cap \mathbb{B}_F^{n \times n}$, with $d$ set to the Frobenius distance, whose cardinality is bounded by $(18/\epsilon)^{(2n+1)k}$. This means that, by rescaling the set diameter by $\sqrt{k}$, there exists an $\epsilon/2$-covering of $\mathcal{R}^F(k,n)$ with cardinality bound $(18\sqrt{k}/\epsilon)^{(2n+1)k}$, and thus, from the considerations above, there exists an $\epsilon$-covering $\mathcal{N}_\epsilon \subset \mathbb{G}(k,n)$ with $|\mathcal{N}_\epsilon| \leqslant (18\sqrt{k}/\epsilon)^{(2n+1)k}$. $\square$

We can now use previous lemmas to provide the final proof for Prop. 9 that we restate here for simplicity.

**Proposition** (Prop. 9 (restated)). *Let $\delta > 0$ and $C, C', c > 0$ be absolute constants. If*

$$m \geqslant C\delta^{-2}nk\log(\tfrac{n^2}{k\delta^2}),$$

*then, with probability exceeding $1 - C'\exp(-c\delta^2 m)$,*

$$\sup_{\boldsymbol{U},\boldsymbol{V} \in \mathbb{V}(k,n)} |\widehat{\kappa}^{\pm}(\boldsymbol{U},\boldsymbol{V}) - \kappa^{\pm}(\boldsymbol{U},\boldsymbol{V})| = \sup_{\boldsymbol{U},\boldsymbol{V} \in \mathbb{V}(k,n)} |\tfrac{1}{m}\langle \psi^{\pm}(\boldsymbol{U}), \psi^{\pm}(\boldsymbol{V})\rangle - \kappa^{\pm}(\boldsymbol{U},\boldsymbol{V})| \leqslant \delta.$$

*Proof.* Given a radius $\epsilon > 0$ to be fixed later, we know from Prop. 21 that there exists an $\epsilon$-covering $\mathcal{N}_\epsilon \subset \mathbb{G}(k,n)$ of $\mathbb{G}(k,n)$ with cardinality $|\mathcal{N}_\epsilon| \leqslant (18\sqrt{k}/\epsilon)^{(2n+1)k}$. We also consider the set $\mathcal{V}_\epsilon \subset \mathbb{V}(k,n)$ such that for any $\boldsymbol{U}^* \in \mathcal{V}_\epsilon$, $\boldsymbol{P}^* := \boldsymbol{U}^*\boldsymbol{U}^{*\top} \in \mathcal{N}_\epsilon$, and where we ensure that each $\boldsymbol{P}^*$ is represented by only one basis in $\mathcal{V}_\epsilon$, *i.e.*, $|\mathcal{V}_\epsilon| = |\mathcal{N}_\epsilon|$. This does not mean, however, that $\mathcal{V}_\epsilon$ is a covering of $\mathbb{V}(k,n)$.

The general objective of this proof is thus to upper bound, with controlled probability, the approximation error $|\widehat{\kappa}^{\pm}(\boldsymbol{U},\boldsymbol{V}) - \kappa^{\pm}(\boldsymbol{U},\boldsymbol{V})|$ for all pairs of subspace bases $\boldsymbol{U}, \boldsymbol{V} \in \mathbb{V}(k,n)$. Let us first observe that, given $\boldsymbol{P}^* = \boldsymbol{U}^*\boldsymbol{U}^{*\top}$ and $\boldsymbol{Q}^* = \boldsymbol{V}^*\boldsymbol{V}^{*\top}$ the closest projectors in $\mathcal{N}_\epsilon$ to $\boldsymbol{P} = \boldsymbol{U}\boldsymbol{U}^\top$ and $\boldsymbol{Q} = \boldsymbol{V}\boldsymbol{V}^\top$, one can upper bound this error with 3 terms and target the following objective bound:

$$|\widehat{\kappa}^{\pm}(\boldsymbol{U},\boldsymbol{V}) - \kappa^{\pm}(\boldsymbol{U},\boldsymbol{V})| \leqslant T_1 + T_2 + T_3 \leqslant \delta,$$

where $m\widehat{\kappa}^{\pm}(\boldsymbol{U},\boldsymbol{V}) = \langle \psi^{\pm}(\boldsymbol{U}), \psi^{\pm}(\boldsymbol{V})\rangle$, $\kappa^{\pm}(\boldsymbol{U},\boldsymbol{V}) = \mathbb{E}\widehat{\kappa}^{\pm}(\boldsymbol{U},\boldsymbol{V})$, and the three terms

$$T_1 = |\widehat{\kappa}^{\pm}(\boldsymbol{U},\boldsymbol{V}) - \widehat{\kappa}^{\pm}(\boldsymbol{U}^*,\boldsymbol{V}^*)|, \quad T_2 = |\widehat{\kappa}^{\pm}(\boldsymbol{U}^*,\boldsymbol{V}^*) - \kappa^{\pm}(\boldsymbol{U}^*,\boldsymbol{V}^*)|,$$
$$T_3 = |\kappa^{\pm}(\boldsymbol{U}^*,\boldsymbol{V}^*) - \kappa^{\pm}(\boldsymbol{U},\boldsymbol{V})|.$$

We now handle these three terms separately, enforcing each of them to be at most $\delta/3$. The two first terms will be bounded conditionally to two probabilistic events (defined below), respectively, $\mathcal{E}_1$ and $\mathcal{E}_2$, while the last term is deterministic. By union bound, the final approximation error will thus have a probability failure summing the probabilities of failure of the two first events.

*(a) Bound on $T_1$:* Given the random vectors $\boldsymbol{a}_i, \boldsymbol{b}_i \sim_{\text{i.i.d.}} \mathcal{N}(\boldsymbol{0}, \boldsymbol{I}_n)$ and the mapping $\bar{\psi}_i : \boldsymbol{P} \in \mathbb{G}(k,n) \mapsto \text{sign}(\boldsymbol{a}_i^\top \boldsymbol{P}\boldsymbol{b}_i)$, $1 \leqslant i \leqslant m$, defining the feature map $\psi^{\pm}$, the first term $T_1$ can be upper bounded by

$$\begin{aligned}
T_1 &\leqslant \tfrac{1}{m}\sum_{i=1}^m |\text{sign}(\boldsymbol{a}_i^\top \boldsymbol{P}\boldsymbol{b}_i)\text{sign}(\boldsymbol{a}_i^\top \boldsymbol{Q}\boldsymbol{b}_i) - \text{sign}(\boldsymbol{a}_i^\top \boldsymbol{P}^*\boldsymbol{b}_i)\text{sign}(\boldsymbol{a}_i^\top \boldsymbol{Q}^*\boldsymbol{b}_i)| \\
&= \tfrac{2}{m}\sum_{i=1}^m \mathbb{1}[\bar{\psi}_i(\boldsymbol{P})\bar{\psi}_i(\boldsymbol{Q}) \neq \bar{\psi}_i(\boldsymbol{P}^*)\bar{\psi}_i(\boldsymbol{Q}^*)] \\
&\leqslant \tfrac{2}{m}\sum_{i=1}^m \mathbb{1}[\bar{\psi}_i(\boldsymbol{P}) \neq \bar{\psi}_i(\boldsymbol{P}^*) \text{ or } \bar{\psi}_i(\boldsymbol{Q}) \neq \bar{\psi}_i(\boldsymbol{Q}^*)] \\
&\leqslant \tfrac{2}{m}\sum_{i=1}^m \mathbb{1}[\bar{\psi}_i(\boldsymbol{P}) \neq \bar{\psi}_i(\boldsymbol{P}^*)] + \tfrac{2}{m}\sum_{i=1}^m \mathbb{1}[\bar{\psi}_i(\boldsymbol{Q}) \neq \bar{\psi}_i(\boldsymbol{Q}^*)] \\
&\leqslant \tfrac{2}{m}\sum_{i=1}^m \mathbb{1}[\bar{\psi}_i \notin \mathsf{C}^0(\mathcal{N}_\epsilon(\boldsymbol{P}^*))] + \tfrac{2}{m}\sum_{i=1}^m \mathbb{1}[\bar{\psi}_i \notin \mathsf{C}^0(\mathcal{N}_\epsilon(\boldsymbol{Q}^*))].
\end{aligned}$$

where $\mathcal{N}_\epsilon(\boldsymbol{P}^*) = \mathbb{B}_\epsilon^F(\boldsymbol{P}^*) \cap \mathbb{G}(k,n)$ is an $\epsilon$-ball centred on $\boldsymbol{P}^*$. Defining the first event

$$\mathcal{E}_1 := \{\forall \boldsymbol{P}' \in \mathcal{N}_\epsilon : |\{i : \bar{\psi}_i \notin \mathsf{C}^0(\mathcal{N}_\epsilon(\boldsymbol{P}'))\}| \leqslant \tfrac{1}{12}\delta m\},$$

it is clear that if $\mathcal{E}_1$ holds, then $T_1 \leqslant \delta/3$, for all possible $\boldsymbol{P}, \boldsymbol{Q}, \boldsymbol{P}^*$ and $\boldsymbol{Q}^*$.

From Cor. 19, setting $\epsilon = \eta(\delta/12)$, we find by union bound over all elements of $\mathcal{N}_{\eta(\delta/12)}$ that, for some $C, c > 0$,

$$\begin{aligned}
\mathbb{P}[\mathcal{E}_1^c] &\leqslant C|\mathcal{N}_{\eta(\delta/12)}|\exp(-c\delta^2 m) \leqslant C\exp((2n+1)k\log(18\tfrac{\sqrt{k}}{\eta(\delta/12)}) - c\delta^2 m) \\
&\leqslant C\exp(cnk\log(\tfrac{\sqrt{k}}{\eta(\delta/12)}) - c'\delta^2 m).
\end{aligned}$$

*(b) Bound on $T_2$:* Regarding the second term $T_2$, we can ensure that $T_2 \leqslant \delta/3$ for all possible $\boldsymbol{P}, \boldsymbol{Q}, \boldsymbol{P}^*$ and $\boldsymbol{Q}^*$ if this event holds:

$$\mathcal{E}_2 := \{\forall \boldsymbol{U}^*, \boldsymbol{V}^* \in \mathcal{V}_\epsilon, |\tfrac{1}{m}\langle \psi^{\pm}(\boldsymbol{U}^*), \psi^{\pm}(\boldsymbol{V}^*)\rangle - \kappa^{\pm}(\boldsymbol{U}^*,\boldsymbol{V}^*)| < \tfrac{\delta}{3}\},$$

with $\epsilon = \eta(\delta/12)$. From Prop. 8 and a union bound an all possible pairs of bases picked in $\mathcal{V}_\epsilon \times \mathcal{V}_\epsilon$, we known that, for some $C, c, c' > 0$,

$$\mathbb{P}[\mathcal{E}_2^c] \leqslant 2|\mathcal{N}_{\eta(\delta/12)}|^2 \exp(-1/2m\delta^2) \leqslant C \exp(cnk\log(\tfrac{\sqrt{k}}{\eta(\delta/12)}) - c'\delta^2 m).$$

*(c) Bound on $T_3$:* Finally, regarding $T_3$, we can bound with several calls to the triangular inequality, so that, for all possible $\boldsymbol{P}$, $\boldsymbol{Q}$, $\boldsymbol{P}^*$ and $\boldsymbol{Q}^*$,

$$
\begin{aligned}
T_3 &\leqslant \tfrac{1}{m}\sum_{i=1}^{m} \mathbb{E}|\operatorname{sign}(\boldsymbol{a}_i^\top \boldsymbol{P}\boldsymbol{b}_i)\operatorname{sign}(\boldsymbol{a}_i^\top \boldsymbol{Q}\boldsymbol{b}_i) - \operatorname{sign}(\boldsymbol{a}_i^\top \boldsymbol{P}^*\boldsymbol{b}_i)\operatorname{sign}(\boldsymbol{a}_i^\top \boldsymbol{Q}^*\boldsymbol{b}_i)| \\
&= \tfrac{1}{m}\sum_{i=1}^{m} \mathbb{E}[2\mathbb{1}[\bar{\psi}_i(\boldsymbol{P})\bar{\psi}_i(\boldsymbol{Q}) \neq \bar{\psi}_i(\boldsymbol{P}^*)\bar{\psi}_i(\boldsymbol{Q}^*)]] \\
&\leqslant 2\mathbb{P}[\bar{\psi}(\boldsymbol{P})\bar{\psi}(\boldsymbol{Q}) \neq \bar{\psi}(\boldsymbol{P}^*)\bar{\psi}(\boldsymbol{Q}^*)] \\
&\leqslant 2\mathbb{P}[\bar{\psi}(\boldsymbol{P}) \neq \bar{\psi}(\boldsymbol{P}^*)] + 2\mathbb{P}[\bar{\psi}(\boldsymbol{Q}) \neq \bar{\psi}(\boldsymbol{Q}^*)] \\
&= 2\mathbb{P}[\bar{\psi} \notin \mathsf{C}^0(\mathcal{N}_{\eta(\delta/12)}(\boldsymbol{P}^*))] + 2\mathbb{P}[\bar{\psi} \notin \mathsf{C}^0(\mathcal{N}_{\eta(\delta/12)}(\boldsymbol{Q}^*))],
\end{aligned}
$$

with $\bar{\psi} : \boldsymbol{P} \in \mathbb{G}(k,n) \mapsto \operatorname{sign}(\boldsymbol{a}^\top \boldsymbol{P}\boldsymbol{b})$ for $\boldsymbol{a}, \boldsymbol{b} \sim \mathcal{N}(\boldsymbol{0}, \boldsymbol{I}_n)$. Finally, Lem. 18 allows us to conclude that

$$T_3 \leqslant 2\,\mathbb{P}[\bar{\psi} \notin \mathsf{C}^0(\mathcal{N}_{\eta(\delta/12)}(\boldsymbol{P}^*))] \;+\; 2\,\mathbb{P}[\bar{\psi} \notin \mathsf{C}^0(\mathcal{N}_{\eta(\delta/12)}(\boldsymbol{Q}^*))] \leqslant 4\delta/12 = \delta/3.$$

*Final bound:* Gathering the three terms $T_1$, $T_2$, and $T_3$, we can finally state that

$$\sup_{\boldsymbol{U},\boldsymbol{V}\in\mathbb{V}(k,n)} |\widehat{\kappa}^\pm(\boldsymbol{U},\boldsymbol{V}) - \kappa^\pm(\boldsymbol{U},\boldsymbol{V})| \leqslant \delta,$$

with a failure probability given by

$$p_{\text{fail}} = \mathbb{P}(\mathcal{E}_1^c) + \mathbb{P}(\mathcal{E}_2^c) \leqslant C \exp(cnk\log(\tfrac{\sqrt{k}}{\eta(\delta/12)}) - c'\delta^2 m).$$

Therefore, imposing $m \geqslant C\delta^{-2}nk\log(\tfrac{\sqrt{k}}{\eta(\delta/12)})$, we get $p_{\text{fail}} \leqslant C \exp(-c\delta^2 m)$. Since, using the expression of $\eta$,

$$\log(\tfrac{\sqrt{k}}{\eta(\delta/12)}) \leqslant \log(\sqrt{k}) + \tfrac{k}{k-1}\log(\tfrac{48(n+k)}{\sqrt{k}\delta}) \leqslant C\log(\tfrac{n^2}{k\delta^2}),$$

the condition on $m$ simplifies into the stronger condition

$$m \geqslant C\delta^{-2}nk\log(\tfrac{n^2}{k\delta^2}).$$

$\square$

# B    Proof of Prop. 13

**Proposition** (Prop. 13 (restated)). *Let $\delta > 0$ and let $C, c, c_0 > 0$ be absolute constants. If*

$$m \geqslant C\delta^{-2}nk\log(\sqrt{k}\omega n/\delta),$$

*then*

$$\sup_{\boldsymbol{U},\boldsymbol{V}\in\mathbb{V}(k,n)} |\widehat{\kappa}^\circ(\boldsymbol{U},\boldsymbol{V}) - \kappa^\circ(\boldsymbol{U},\boldsymbol{V})| = \sup_{\boldsymbol{U},\boldsymbol{V}\in\mathbb{V}(k,n)} \left|\tfrac{1}{m}\langle\psi^\circ(\boldsymbol{U}),\psi^\circ(\boldsymbol{V})\rangle - \kappa^\circ(\boldsymbol{U},\boldsymbol{V})\right| \leqslant \delta$$

*with probability at least $1 - c\exp(-c_0\delta^2 m)$, for some absolute constant $c, c_0 > 0$.*

*Proof.* Given a radius $\epsilon > 0$ to be fixed later, we know from Prop. 21 that there exists an $\epsilon$-covering $\mathcal{N}_\epsilon \subset \mathbb{G}(k,n)$ of $\mathbb{G}(k,n)$ with cardinality $|\mathcal{N}_\epsilon| \leqslant (18\sqrt{k}/\epsilon)^{(2n+1)k}$. As in the proof of Prop. 9, we consider a set $\mathcal{V}_\epsilon \subset \mathbb{V}(k,n)$ such that for any $\boldsymbol{U}^* \in \mathcal{V}_\epsilon$, $\boldsymbol{P}^* := \boldsymbol{U}^*\boldsymbol{U}^{*\top} \in \mathcal{N}_\epsilon$, and such that each $\boldsymbol{P}^* \in \mathcal{N}_\epsilon$ is represented by a unique basis in $\mathcal{V}_\epsilon$, *i.e.*, $|\mathcal{V}_\epsilon| = |\mathcal{N}_\epsilon|$.

The proof proceeds in a similar way to that of Prop. 9, by decomposing the approximation error into three terms and then controlling each term separately. For arbitrary $\boldsymbol{U}, \boldsymbol{V} \in \mathbb{V}(k, n)$ and associated net points $\boldsymbol{U}^*, \boldsymbol{V}^* \in \mathcal{V}_\epsilon$, with projectors $\boldsymbol{P}^* = \boldsymbol{U}^* \boldsymbol{U}^{*\top}$ and $\boldsymbol{Q}^* = \boldsymbol{V}^* \boldsymbol{V}^{*\top}$, we write

$$|\widehat{\kappa}^\circ(\boldsymbol{U}, \boldsymbol{V}) - \kappa^\circ(\boldsymbol{U}, \boldsymbol{V})| \leqslant T_1 + T_2 + T_3,$$

with $T_1 := |\widehat{\kappa}^\circ(\boldsymbol{U}, \boldsymbol{V}) - \widehat{\kappa}^\circ(\boldsymbol{U}^*, \boldsymbol{V}^*)|$, $T_2 := |\widehat{\kappa}^\circ(\boldsymbol{U}^*, \boldsymbol{V}^*) - \kappa^\circ(\boldsymbol{U}^*, \boldsymbol{V}^*)|$, and $T_3 := |\kappa^\circ(\boldsymbol{U}^*, \boldsymbol{V}^*) - \kappa^\circ(\boldsymbol{U}, \boldsymbol{V})|$.

We now bound each of the three terms by $\delta/3$ in order to ensure a total approximation error bounded by $\delta$.

*(a) Bound on $T_1$:* Let us first consider the event $\mathcal{E}_1 := \{\frac{1}{mn} \sum_{j=1}^m (\|\boldsymbol{a}_j\|_2^2 + \|\boldsymbol{b}_j\|_2^2) \leqslant 3\}$. Since $\boldsymbol{a}_j, \boldsymbol{b}_j \sim_{\text{i.i.d.}} \mathcal{N}(\boldsymbol{0}, \boldsymbol{I}_n)$, the random variable $Z := \sum_{j=1}^m (\|\boldsymbol{a}_j\|_2^2 + \|\boldsymbol{b}_j\|_2^2)$ is distributed as a $\chi_{2mn}^2$-distribution with $2mn$ degrees of freedom, and $\mathbb{E}Z = 2mn$. Moreover, by the Laurent–Massart inequality (Laurent & Massart, 2000, Lemma 1), there exist absolute constants $c > 0$ such that

$$\mathbb{P}(\mathcal{E}_1) \geqslant 1 - \exp(-cmn). \tag{21}$$

Let us assume that $\mathcal{E}_1$ holds. For any $\boldsymbol{P}, \boldsymbol{Q} \in \mathbb{G}(k, n)$, let $f(\boldsymbol{P}, \boldsymbol{Q}) := \frac{1}{m} \sum_{j=1}^m \exp(i\omega \boldsymbol{a}_j^\top (\boldsymbol{P} - \boldsymbol{Q}) \boldsymbol{b}_j)$, and $\boldsymbol{P}^*, \boldsymbol{Q}^* \in \mathbb{G}(k, n)$ be associated net points with $\|\boldsymbol{P} - \boldsymbol{P}^*\|_F \leqslant \epsilon$ and $\|\boldsymbol{Q} - \boldsymbol{Q}^*\|_F \leqslant \epsilon$. Then

$$mT_1 = m|f(\boldsymbol{P}, \boldsymbol{Q}) - f(\boldsymbol{P}^*, \boldsymbol{Q}^*)| \leqslant \sum_{j=1}^m |\exp(i\omega \boldsymbol{a}_j^\top (\boldsymbol{P} - \boldsymbol{Q}) \boldsymbol{b}_j) - \exp(i\omega \boldsymbol{a}_j^\top (\boldsymbol{P}^* - \boldsymbol{Q}^*) \boldsymbol{b}_j)|.$$

Using $|\exp(ix) - \exp(iy)| \leqslant |x - y|$ for any $x, y \in \mathbb{R}$, we obtain

$$m|f(\boldsymbol{P}, \boldsymbol{Q}) - f(\boldsymbol{P}^*, \boldsymbol{Q}^*)| \leqslant \omega \sum_{j=1}^m |\boldsymbol{a}_j^\top ((\boldsymbol{P} - \boldsymbol{P}^*) - (\boldsymbol{Q} - \boldsymbol{Q}^*)) \boldsymbol{b}_j|.$$

Hence, by the triangle inequality,

$$|f(\boldsymbol{P}, \boldsymbol{Q}) - f(\boldsymbol{P}^*, \boldsymbol{Q}^*)| \leqslant \frac{\omega}{m} \sum_{j=1}^m \left(|\boldsymbol{a}_j^\top (\boldsymbol{P} - \boldsymbol{P}^*) \boldsymbol{b}_j| + |\boldsymbol{a}_j^\top (\boldsymbol{Q} - \boldsymbol{Q}^*) \boldsymbol{b}_j|\right).$$

Using Cauchy–Schwarz gives $|\boldsymbol{a}_j^\top (\boldsymbol{P} - \boldsymbol{P}^*) \boldsymbol{b}_j| = |\langle \boldsymbol{P} - \boldsymbol{P}^*, \boldsymbol{a}_j \boldsymbol{b}_j^\top \rangle| \leqslant \|\boldsymbol{P} - \boldsymbol{P}^*\|_F \|\boldsymbol{a}_j\|_2 \|\boldsymbol{b}_j\|_2$, and similarly for $\boldsymbol{Q}$. Using $\|\boldsymbol{P} - \boldsymbol{P}^*\|_F \leqslant \epsilon$, $\|\boldsymbol{Q} - \boldsymbol{Q}^*\|_F \leqslant \epsilon$ and $\|x\|\|y\| \leqslant (\|x\|^2 + \|y\|^2)/2$ yields

$$T_1 = |f(\boldsymbol{P}, \boldsymbol{Q}) - f(\boldsymbol{P}^*, \boldsymbol{Q}^*)| \leqslant \frac{2\omega\epsilon}{m} \sum_{j=1}^m \|\boldsymbol{a}_j\|_2 \|\boldsymbol{b}_j\|_2 \leqslant \frac{\omega\epsilon}{m} \sum_{j=1}^m (\|\boldsymbol{a}_j\|_2^2 + \|\boldsymbol{b}_j\|_2^2).$$

Therefore, under the event $\mathcal{E}_1$, $T_1 \leqslant 3\omega n\epsilon$. In particular, choosing $\epsilon = \frac{\delta}{9\omega n}$ ensures $T_1 \leqslant \frac{\delta}{3}$.

*(b) Bound on $T_2$:* We now control the second term $T_2$. For any fixed $\boldsymbol{U}^*, \boldsymbol{V}^* \in \mathcal{V}_\epsilon$, with associated projectors $\boldsymbol{P}^* = \phi^{\mathrm{P}}(\boldsymbol{U}^*)$ and $\boldsymbol{Q}^* = \phi^{\mathrm{P}}(\boldsymbol{V}^*)$, Prop. 12 ensures the existence of absolute constants $C, c > 0$ such that

$$\mathbb{P}\left(|\widehat{\kappa}^\circ(\boldsymbol{U}^*, \boldsymbol{V}^*) - \kappa^\circ(\boldsymbol{U}^*, \boldsymbol{V}^*)| \geqslant \frac{\delta}{3}\right) \leqslant C \exp(-cm\delta^2).$$

Let us define the second event $\mathcal{E}_2 = \left\{\forall \boldsymbol{U}^*, \boldsymbol{V}^* \in \mathcal{V}_\epsilon, |\widehat{\kappa}^\circ(\boldsymbol{U}^*, \boldsymbol{V}^*) - \kappa^\circ(\boldsymbol{U}^*, \boldsymbol{V}^*)| < \frac{\delta}{3}\right\}$. Since $|\mathcal{V}_\epsilon| = |\mathcal{N}_\epsilon|$, a union bound over all pairs $\boldsymbol{U}^*, \boldsymbol{V}^* \in \mathcal{V}_\epsilon$ yields

$$\mathbb{P}(\mathcal{E}_2^c) \leqslant C|\mathcal{N}_\epsilon|^2 \exp(-cm\delta^2).$$

Using the covering bound from Prop. 21 with the value of $\epsilon = \delta/9\omega n$ set above, we have $|\mathcal{N}_\epsilon| \leqslant (18\sqrt{k}/\epsilon)^{(2n+1)k} \leqslant (162\omega\sqrt{k}n/\delta)^{(2n+1)k}$, and hence, for some absolute constants $C, C', c > 0$,

$$\mathbb{P}(\mathcal{E}_2^c) \leqslant C \exp\left(C'nk \log(\omega\sqrt{k}n/\delta) - cm\delta^2\right).$$

In particular, updating the constants, if

$$m \geqslant C\delta^{-2}nk \log(\omega\sqrt{k}n/\delta),$$

then $\mathbb{P}(\mathcal{E}_2^c) \leqslant C' \exp(-cm\delta^2)$. Under this condition, $T_2 \leqslant \frac{\delta}{3}$ simultaneously for all pairs of $\boldsymbol{U}^*, \boldsymbol{V}^* \in \mathcal{V}_\epsilon$ with probability at least $1 - C' \exp(-cm\delta^2)$.

*(c) Bound on $T_3$:* We now control the deterministic term

$$T_3 = |\kappa^\circ(\boldsymbol{U}, \boldsymbol{V}) - \kappa^\circ(\boldsymbol{U}^*, \boldsymbol{V}^*)|.$$

Let us define for $\boldsymbol{P}, \boldsymbol{Q} \in \mathbb{G}(k, n)$, $h(\boldsymbol{P}, \boldsymbol{Q}) = \exp\left(i\omega \boldsymbol{a}^\top(\boldsymbol{P} - \boldsymbol{Q})\boldsymbol{b}\right)$, so that $\kappa^\circ(\boldsymbol{U}, \boldsymbol{V}) = \mathbb{E}[h(\boldsymbol{P}, \boldsymbol{Q})]$. By linearity of expectation and the triangle inequality,

$$T_3 \leqslant \mathbb{E}[|h(\boldsymbol{P}, \boldsymbol{Q}) - h(\boldsymbol{P}^*, \boldsymbol{Q}^*)|].$$

Using again $|\exp(ix) - \exp(iy)| \leqslant |x - y|$, we obtain by Cauchy–Schwarz

$$|h(\boldsymbol{P}, \boldsymbol{Q}) - h(\boldsymbol{P}^*, \boldsymbol{Q}^*)| \leqslant \omega(|\boldsymbol{a}^\top(\boldsymbol{P} - \boldsymbol{P}^*)\boldsymbol{b}| + |\boldsymbol{a}^\top(\boldsymbol{Q} - \boldsymbol{Q}^*)\boldsymbol{b}|) \leqslant \omega(\|\boldsymbol{P} - \boldsymbol{P}^*\|_F + \|\boldsymbol{Q} - \boldsymbol{Q}^*\|_F)\|\boldsymbol{a}\|_2\|\boldsymbol{b}\|_2,$$

so that, taking expectations, using $\mathbb{E}[\|\boldsymbol{a}\|_2\|\boldsymbol{b}\|_2] \leqslant n$ and the value of $\epsilon$ set above,

$$T_3 \leqslant \omega(\|\boldsymbol{P} - \boldsymbol{P}^*\|_F + \|\boldsymbol{Q} - \boldsymbol{Q}^*\|_F)\mathbb{E}[\|\boldsymbol{a}\|_2\|\boldsymbol{b}\|_2] \leqslant 2\omega n \epsilon \leqslant \tfrac{2}{9}\delta < \tfrac{1}{3}\delta,$$

which holds uniformly over all $\boldsymbol{U}, \boldsymbol{V} \in \mathbb{V}(k, n)$.

*Final bound:* We now combine the bounds $T_1 \leqslant \delta/3$, $T_2 \leqslant \delta/3$, and $T_3 \leqslant \delta/3$ obtained above, as well as the probabilities (by union bound) and conditions under which they hold. We proved that $\mathbb{P}[T_1 \leqslant \delta/3 | \mathcal{E}_1] = 1$ and $\mathbb{P}[T_2 \leqslant \delta/3 | \mathcal{E}_2] = 1$, with also $\mathbb{P}[T_3 \leqslant \delta/3] = 1$. Therefore, $\mathbb{P}[T_1 + T_2 + T_3 > \delta] \leqslant \mathbb{P}[T_1 > \delta/3] + \mathbb{P}[T_2 > \delta/3]$. However,

$$\mathbb{P}[T_1 > \delta/3] = 1 - \mathbb{P}[T_1 \leqslant \delta/3 | \mathcal{E}_1]\mathbb{P}[\mathcal{E}_1] - \mathbb{P}[T_1 \leqslant \delta/3 | \mathcal{E}_1^c]\mathbb{P}[\mathcal{E}_1^c] = 1 - \mathbb{P}[\mathcal{E}_1] - \mathbb{P}[T_1 \leqslant \delta/3 | \mathcal{E}_1^c]\mathbb{P}[\mathcal{E}_1^c] \leqslant \mathbb{P}[\mathcal{E}_1^c],$$

and, similarly, $\mathbb{P}[T_2 > \delta/3] \leqslant \mathbb{P}[\mathcal{E}_2^c]$. This shows that, for some constants $C, C', c > 0$, provided that

$$m \geqslant C\delta^{-2}nk\log(\omega\sqrt{k}n/\delta),$$

with probability exceeding $1 - \mathbb{P}(\mathcal{E}_1^c) - \mathbb{P}(\mathcal{E}_2^c) \geqslant 1 - C'\exp(-c\delta^2 m)$, we have

$$\sup_{\boldsymbol{U}, \boldsymbol{V} \in \mathbb{V}(k,n)} |\widehat{\kappa}^\circ(\boldsymbol{U}, \boldsymbol{V}) - \kappa^\circ(\boldsymbol{U}, \boldsymbol{V})| = T_1 + T_2 + T_3 \leqslant \delta,$$

which concludes the proof. $\qquad\square$

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
