# OpenReview forum: "Random features for Grassmannian kernel approximation with bounded rank-one projections"
_TMLR — Decision pending for TMLR_

### Review · Reviewer_XJL8 · 2026-04-13

**Summary Of Contributions:**

The main contribution of this paper is an efficient evaluation of kernel maps which consider similarity over a set of objects in a Grassmannian manifold.  The main idea is based on associating each subspace in the manifold with a projector UU^T (Gram matrix). The paper approaches the problem by considering bilinear maps of a projector (defined via rank-one projections). This then leads naturally to an approximated kernel, whereby one considers the inner product of maps applied to each projector. A key departure from existing works is the application of bounded operator: one is a sign function, and another is complex exponential (the latter inspired by the seminal random features paper). The main strength of the paper is that it provides a uniform bound on the kernel approximations. Beyond numerical experiments, section 6 which deals with faster random features is also a strength of the work.

**Additional Comments:**

None

**Audience:**

Yes

**Audience Explanation:**

Efficient kernel approximation would be relevant to the broad TMLR audience, as it intersects with manifold learning, representation learning, and subspace learning.

**Claims And Evidence:**

Yes

**Claims Explanation:**

The theory is supported by proofs, and numerical experiments showing advantages of the proposed method are shown.

**Requested Changes:**

This is a very well-written paper, and I appreciate the exposition (in particular the way the authors frame their contributions in the context of existing works and literature), the theory and experiments. I would appreciate if the authors could address my questions below:

*  The manuscript states that Nyström-based methods require computing all N^2 pairwise similarities. Could the authors clarify this point? My understanding is that standard Nyström approximations typically require only O(Nm) kernel evaluations (with m<<N), without forming the full Gram matrix.
* .What is the connection of this work to recovering Gram matrices from limited distance measurements? This is a central problem in Euclidean distance geometry (see references (1) and (2) below). In the context of the paper, UU^T is a Gram matrix and is low-rank (when n>>k), and features are obtained by (e_i-e_j)^T*UU^T*(e_i-e_j). Is the idea that one still obtains a poor estimate without applying bounded operators as done in Section 3.4? It would be interesting to understand whether recovery guarantees from that literature provide any insight, even heuristically, into how many measurements or features are needed in the present setting.

* The bound in Proposition 13 is uniform. Is this worst-case dependence necessary, or could one obtain improved sample complexity by restricting to a subset of subspaces with smaller metric complexity (e.g., bounded principal angles or clustered structure)?

Minor comments:
1. I think the definition of principal angle in Section 2.2 could be simplified for readability.
2. Page 6 (top): psd should be consistent with the rest of the paper

References
1. Dokmanic, Ivan, et al. "Euclidean distance matrices: essential theory, algorithms, and applications." IEEE Signal Processing Magazine 32.6 (2015): 12-30.
2. Tasissa, Abiy, and Rongjie Lai. "Exact reconstruction of euclidean distance geometry problem using low-rank matrix completion." IEEE Transactions on Information Theory 65.5 (2018): 3124-3144.

---

> ### Author Response · Authors · 2026-04-15
> **Responses to Reviewer 1**
>
> We would like to thank this reviewer for taking the time to write those comments which we address hereafter.
>
> ## 1. Nyström complexity
>
> The reviewer correctly points out that, conversely to what we wrote, the standard Nyström methods do not require computing all $N^2$ pairwise similarities. In practice, these methods approximate the Gram matrix by a low-rank approximation obtained from the selection of a subset of $m \ll N$ data points from the dataset. As a result, the overall memory storage and computational complexity are on the order of $O(mN)$ and $O(m^2N)$ respectively. We will revise the manuscript to correct this statement in the Introduction.
>
> That said, our original motivation still holds. Nyström methods remain data-dependent and their computational cost scales with the dataset size $N$, since kernel evaluations must be computed between all samples and the selected subset. In contrast, random feature methods construct an explicit embedding whose evaluation cost does not depend on $N$ once the mapping is fixed.
>
>
> ## 2. Connection to Gram matrices and distance geometry
>
> The reviewer highlights a possible connection between our work and Gram matrix recovery from limited distance measurements, a central problem in Euclidean distance geometry. The reviewer also wonders if we still obtain poor estimates without applying bounded operators as in our work.
>
> In our setting, we often consider low-rank, positive semi-definite projection matrices shaped as $P = U U^\top$. Such matrices can indeed be interpreted as Gram-type matrices, which explains the conceptual link with works that aim to recover them from a limited number of measurements, e.g., using the sensing model $(e_i-e_j)^TUU^T(e_i-e_j)$ (with the elements $\{e_i\}_{i=1}^n$ of the canonical basis) explained in the references provided by this reviewer.
>
> As the reviewer has certainly noticed, our approach does not aim to reconstruct the matrix $P$; our goal is to construct random feature maps $\Phi$ such that, when applied to two subspaces $P$ and $P'$, the inner product $\langle \Phi(P), \Phi(P')\rangle$ of their features approximate a given Grassmannian kernel $\kappa(P,P')$. More specifically, our approach consists in building such mappings from random rank-one projections combined with bounded non-linear functions, hence bounding the kernel approximation error thanks to known measure concentration results. In summary, we aim at encapsulating the similarity between subspaces (as measured by a kernel) through the random feature map $\Phi$ and not to use it to estimate the subspaces.
>
> The works mentioned by this reviewer in the context of the Euclidean distance geometry problem can be used to establish sample complexity bounds for Gram matrix recovery for specific recovery algorithms, even in the case where the ROP-like sensing operator is unbounded. While the question is interesting, it is not clear to us, however, how these results could be used to build kernel approximation between subspaces for an unbounded ROP model. For instance, as explained by (Chen et al., 2015; Cai & Zhang, 2015) (see Sec 3.2), as no restricted isometry property (RIP) is available for unbounded random ROP, a projection kernel approximation is hardly reachable for reasonable values of $m$ (i.e., scaling like $O(kn)$).
>
>
> ## 3. Uniform (worst-case) bounds and possible improvements
>
> As pointed by this reviewer, Prop. 13 determines from which number of measurements (i.e. the feature map dimension) one can uniformly bound the kernel approximation error. This is indeed a worst-case scenario established over all subspaces of fixed dimension $k$ in $\mathbb{R}^n$. Such a uniform bound is key to characterize kernel machine performances in machine learning, where we typically do not assume any particular structure on the data.
>
> As a result, the dependence on $nk$ is aligned with the complexity of each of these subspaces (i.e. their number of free parameters).
>
> Reduced sample complexity bounds could be obtained by restricting attention to smaller subsets of subspaces with additional structure (such as sparsity in the entries of $U$) provided these subsets have covering sets with smaller cardinality (as established through the proof of Prop. 13). While such an improvement is beyond the scope of the present work, we will insert this reviewer comment as a remark right after Prop. 13.
>
> ## 4. Minor comments
>
> **Principal angles definition (Section 2.2)**
>
> We thank the reviewer for this suggestion. We used the standard definition of principal angles found in a few other works, but agree that the current presentation may be heavier than needed. We will revise the wording to improve readability.
>
> **Consistency of "psd" notation**
>
> We thank the reviewer for pointing this out. We will ensure that the notation for positive semidefinite matrices ("psd") is consistent throughout the manuscript.
>
> We thank the reviewer for the comments, the manuscript will be updated once all reviews are received.

---

> > ### Comment · Reviewer_XJL8 · 2026-04-18
> > **comments addressed**
> >
> > I would like to thank the authors for their thorough response which has addressed all my questions and feedback on the paper. I do find the last remark under (2) quite interesting, and I also do agree that it is not immediately clear how these results from EDG might be applied. This is a well-written paper, and the authors have put the work well in context of existing works, and I appreciate the contribution. I recommend acceptance.

---

### Review · Reviewer_FSmh · 2026-05-14

**Summary Of Contributions:**

This paper introduces two new random feature approximation schemes for Grassmannian kernels, i.e. kernels that compare two subspaces of dimension $k$ of $\mathbb{R}^n$. To avoid the high computational cost of dense random projections, the paper uses random rank-one projections, and applies two types of bounded non-linearities to control the fluctuations of the random approximation. The authors additionally propose a structured alternative to generate the random features which reduces the computational and memory complexity of using random features.

*Strengths*: The paper is well-written and gives sufficient background about the Grassmannian manifold, kernel methods and random feature approximation. I did not check the math closely, but it seems like bounding the random feature magnitudes via a non-linearity can be an effective way to obtain suitable theoretical guarantees for feature approximation in this setting.

*Weaknesses*: The experiment on the ETH-80 dataset does not justify the use of binary or Fourier kernels, which are the main contributions of this work. From Tables 2 and 3, it seems like the sub-Weibull concentration argument is a bit pessimistic, and in reality using the unbounded rank-one projections perform better than either of the proposed kernels, both in the structured and unstructured setting.

Additionally, I think the authors should better motivate the case of using Grassmannian kernels in practice with large $N$. To my understanding, in the ETH-80 experiment, one a priori needs to know what object an input is representing to be able to derive the suitable subspace for that input. This makes the experiment unrealistic.

**Audience:**

Yes

**Audience Explanation:**

While this is not my line of work, I think that theoretical properties of random feature kernels for data on the Grassmannian manifold should be of interest to some individuals in the machine learning community.

**Broader Impact Concerns:**

Not applicable.

**Claims And Evidence:**

Yes

**Claims Explanation:**

While I did not check the proofs, the authors provide clear statements to support all their theoretical results. The experimental contribution however is a bit weaker. The authors do not compare rank-one projections with dense projections, and among rank-one projections it seems that unbounded features are performing the best. I think these limitations should be better discussed in the paper.

**Requested Changes:**

I have stated my main concerns in the `Summary of Contributions` section. The following are more minor adjustments:

* The authors frequently use the expression $O(kn^2 + k^2n)$. Since $k \leq n$, this can be simplified to $O(kn^2)$.
* For small frequency, the RBF kernel itself is an $O(\omega^2)$ perturbation of a constant ``kernel’’ 1. I think it could be highlighted that the
* What does HDHD mean in Table 3?

---

> ### Author Response · Authors · 2026-05-20
> **Response to reviewer FSmh, part I**
>
> We would like to thank this reviewer for taking the time to read our work and for pointing out several problems, mainly in the experimental section that would benefit from some clarification.
>
>
> ## Preliminary clarification: different embeddings induce different kernels
>
> Before addressing the comments, we would like to clarify that the random feature maps studied in the manuscript do not all approximate the same kernel. Dense Gaussian projections and unbounded ROPs approximate the same projection kernel, whereas the binary and periodic ROP features induce different Grassmannian kernels through their bounded non-linear functions.
>
> This distinction is important when interpreting the experiments. A difference in classification accuracy may reflect not only the quality of the random approximation, but also how well the induced kernel is adapted to the dataset and the considered task. We will make this clearer in the revised manuscript, as it is relevant to several of the reviewer’s comments.
>
> We now address the main comments of the reviewer in the order of their comments.
>
>
> ## 1. Performance of unbounded ROP compared with bounded features
>
> The reviewer correctly observes that, in Tables 2 and 3, the unbounded rank-one projection features perform very well and in several cases outperform the proposed binary and periodic bounded features. We agree that this point needs a better discussion in the manuscript.
>
> As clarified above, these methods should not all be read as approximations of the same kernel. The unbounded ROP features approximate the projection kernel, that may perform very well on ETH-80. The binary and periodic variants instead approximate different Grassmannian kernels through bounded non-linear functions. Therefore, lower classification accuracy for one of these bounded constructions does not necessarily mean that the random approximation itself is worse, it can also mean that the corresponding kernel is not as performant for this particular dataset.
>
> Our goal is not to claim that bounded features always improve classification accuracy compared with unbounded ROP features on every finite dataset. The motivation for introducing bounded non-linear functions is rather to obtain random feature maps for which uniform approximation guarantees over the continuous Grassmannian manifold can be derived with standard concentration tools. By applying a bounded function, such as the sign or the complex exponential, the random variables involved in the empirical kernel become bounded, which allows us to control the approximation error uniformly over a continuum of subspaces.
>
> This does not mean that unbounded ROP features are theoretically unjustified. For a fixed pair of subspaces, the empirical ROP kernel does concentrate around its expectation, the projection kernel, but with a sub-Weibull concentration rate. This leads to pessimistic bounds if one wants a uniform guarantee over the whole continuous Grassmannian. We can show that it will then require a much larger number of measurements than the bounded variants as it would scale as $m=O((kn)^2)$. However, on a fixed finite dataset (and over a finite testing set), we can still combine this concentration result with a union bound over all pairs to obtain a Johnson-Lindenstrauss-type statement (JL). As a result, we can show that we need $m = O((log S)^4)$ measurements for a dataset of size $S$, which may still be low enough for small data set. This explains why unbounded ROP can work well on a training/test set of reasonable size, while still not providing the same data-independent control over the full continuum of subspaces.
>
> We will revise the manuscript to make this distinction explicit. we will especially soften the discussion of unbounded ROP by explaining that it does show good behaviour for finite-sized datasets with a small JL lemma and explain that it is mainly problematic for uniform guarantees over continuous sets. We will also clarify that the bounded variants are introduced to create new kernels whose related embeddings reach stronger uniform approximation guarantees, e.g., over a continuous set of subspaces.

---

> > ### Author Response · Authors · 2026-05-20
> > **Response to reviewer FSmh, part II**
> >
> > ## 2. Motivation for large $N$ and ETH-80 experimental protocol
> >
> > The reviewer asks us to better motivate the use of Grassmannian kernels when the number of instances $N$ is large, and raises a related concern about the ETH-80 experiment by noting that constructing a subspace for a test input requires knowing which images should be grouped together to form this subspace.
> >
> > First, the classification task surrounding the use of ETH-80 should have been stated more clearly. Following earlier works on the design of subspace kernels, notably Wei et al. (2020) and Ji et al. (2015), ETH-80 helps us to illustrate and compare the effectiveness of the different subspace embeddings in an image-set classification problem. This is indeed considered as an instance of the subspace classification field, which covers a large literature, as images obtained from pose (as in ETH-80) or illumination changes of the same object are well approximated by a subspace of small dimension, i.e., it can be estimated from a few images (Hamm & Lee, 2008).
> >
> > If this was not sufficiently clear in our work, the selected task is thus not meant to classify a single new image by deciding, on its own, which subspace it should belong to.
> >
> > More specifically, in the considered image-set classification, we assume that each instance is already a group of images (obtained from several views of the same object) and this group is represented by the subspace spanned by these observations. The task is then to classify this group of images by comparing the corresponding subspace with training subspaces using specific Grassmannian kernels. This protocol is similar to the one followed by Wei et al. (2020) and Ji et al. (2015).
> >
> > Second, the dimension $N$ denotes the number of subspace-valued instances. Once a dataset contains many such instances, exact kernel methods require computing and storing an $N \times N$ Gram matrix, leading to $O(N^2)$ memory and pairwise kernel evaluation costs. Random features address this issue by replacing pairwise kernel evaluations with explicit feature vectors, after which linear methods can be applied, following the general strategy of Rahimi and Recht (2007).
> >
> > We will revise the manuscript accordingly by clarifying that the ETH-80 experiment assumes grouped multi-view observations, and that we do not address the single-image to subspace classification problem. We will also state more explicitly that the large-$N$ motivation refers to the number of subspace-valued instances to be compared or classified.
> >
> >
> > ## 3. Comparison with dense random projections
> >
> > The reviewer points out that the experimental section does not compare rank-one projections with dense random projections. We agree that this comparison is important, since dense projections are presented as the natural baseline for approximating the projection kernel.
> >
> > We have obtained additional results comparing dense Gaussian projections with unbounded ROPs. In fact, both empirical kernels approximate the same projection kernel (see equation (2)), with an approximation error that decreases as the number of features increases. Dense projections are slightly more accurate on average (lower variance), but they are much more expensive both in terms of memory requirements and computation time. For example in our experiment on $\mathrm{G}(5,100)$, at $m=2000$, dense projections are roughly $128$ times slower and require $50$ times more  memory (to store the probing matrices) than ROPs. We will add this experiment to the revised manuscript.
> >
> > This supports the motivation for using ROPs. Dense projections are an interesting basic construction, but each probe is a full matrix, making the method costly in memory and computation. ROPs use rank-one probes and give a much cheaper approximation of the same projection kernel. Making dense projections competitive at larger dimensions would require a more specialised implementation (with GPUs for instance), while ROPs already give a good cost-accuracy trade-off with a very simple implementation.
> >
> > Finally, we will clarify that the comparison with binary and periodic ROPs should be interpreted with care, since these bounded constructions induce different kernels. As we said at the beginning, differences in classification accuracy therefore reflect both approximation quality and the suitability of the induced kernel for the dataset.

---

> > > ### Author Response · Authors · 2026-05-20
> > > **Response to reviewer FSmh, part III**
> > >
> > > ## 4. Minor adjustments
> > >
> > > **Small-frequency behaviour of the periodic kernel**
> > >
> > > The reviewer’s sentence on this point seems to be truncated, so we are not entirely certain of the intended suggestion. We understand it as pointing out that, in the small-frequency regime, the RBF-type kernel discussed in the manuscript is itself close to the constant kernel $1$.
> > >
> > > We will clarify this point. The manuscript already discusses the link between the periodic kernel and an RBF kernel in the small-frequency regime, and this behaviour is illustrated in figure 2. When $\omega$ is very small, the resulting kernel indeed becomes close to constant and is therefore less discriminative. However, this regime may still be meaningful in applications where a very smooth similarity is desired. In that case, proposition 15 guarantees that the periodic kernel is close to the corresponding RBF-type kernel, with an error controlled by $\omega^4$.
> > >
> > > We will add this clarification in the manuscript. We will also mention that the kernel remains well-defined for all values of $\omega$, with larger frequencies leading to more discriminative similarities.
> > >
> > > **Complexity notation**
> > >
> > > We will find the repeated expression mentioned by the reviewer and simplify it throughout the manuscript where appropriate.
> > >
> > > **Meaning of HDHD in table 3**
> > >
> > > The acronym “HDHD” refers to the structured construction of equation (17), based on alternating Hadamard matrices and random diagonal sign matrices. In table 3, it denotes the structured rank-one embedding without the additional binary or periodic function applied.
> > >
> > > We will replace “HDHD” with simply “Structured ROP” as it is in table 2, and add a short clarification in the table caption.
> > >
> > > We would like to thank the reviewer again for the constructive comments. We hope that the clarifications and planned revisions address the concerns described in the review.

---

### Comment · Editors_In_Chief · 2026-06-03
**Third Review**

Due to some technical challenges, we are posting the third review here via comment.

---

Summary:
The paper proposes and theoretically analyzes sketching-style methods for approximating two types of kernels (the projection kernel, and the Binet-Cauchy kernel) defined on the Grassmanian manifold G(k,n).

The main technical ingredients in these methods consist of a few different tools. First, the authors propose asymmetric rank-one projections to get unbiased kernel estimates; however, such such projections are known to be heavy-tailed, and therefore do not provide favorable scaling for sketch complexity. Second, the authors propose capping the projected features with bounded nonlinearities (binary/sign, and sinusoid). This leads to better concentration behavior, and also approximates the kernels better. Third, this also motivates computationally efficient / hadamard-structured projections that resemble randomized FFT sketches. The authors wrap up with some experiment on viewpoint angle estimation from image data.



Evidence for claims:
The paper is outstandingly clear in its writing (kudos to the authors for making a difficult topic so clean and digestible). The main technical contribution is the right assembly of the proof ingredients, and the proofs appear to be correct (modulo some minor questions below). The building blocks are all mostly standard by now. Approximating kernels via sinusoidal random projections dates back to Rahimi-Recht (‘07-’08), rank-one projections to Foucart et al (and many others), fast fourier/Hadamard sketches to Alon ‘03, etc.

But the application to the Grassmanian is to my understanding new and therefore worthy of publication in TMLR. I liked the derivation of the closed form expressions for various kernel approximations; this may help practical adoption and experimentation even beyond the scope of this problem. The experimental results are fine (not super convincing, not sure if practically relevant) but this is squarely a theory paper and should be judged as such.

Relevance to TMLR audience:
Yes, relevant to machine learning theorists.

Requested changes:

1. Does Prop 9/Lm 17/Corr 18 hold for k=1? Would be nice to clarify this. I didn’t fully digest the proof to understand whether the edge case should be handled separately or not; my guess is it does need a separate analysis but this should be short.
2. Potential typo mismatch in omega values in Fig 2 versus text description right below it.

Broader impacts:
No concerns noted.

---

> ### Author Response · Authors · 2026-06-03
> **Response to the third reviewer**
>
> We would like to thank this reviewer for the careful reading and for the positive assessment of the clarity and contribution of the paper.
>
> ## 1. Case $k=1$ in Prop. 9, Lemma 17, and Corollary 18
>
> We thank the reviewer for pointing out that the case $k=1$ needs to be treated with more care. However, we also detected this issue and found a simple patch that allows the argument to cover this case as well.
>
> More precisely, we replace the choice of $\rho^2 = \epsilon^{\frac{2}{k}}\frac{k}{3e(n+k)}$ just before equation 20 by $\rho^2 = \epsilon^{\frac{2}{k+1}}\frac{k}{3e(n+k)}$. With this modification, the problematic expression becomes $\eta(\delta) := \Big(\frac{\sqrt k}{4(n+k)}\delta\Big)^{\frac{k+1}{k}}$ instead of $\eta(\delta) := \Big(\frac{\sqrt k}{4(n+k)}\delta\Big)^{\frac{k}{k-1}}.$
>
> The new expression is now defined for $k=1$ as well, so the result also covers this case. We will update the proof in the manuscript very soon.
>
> ## 2. Mismatch in the values of $\omega$ in Fig. 2
>
> The reviewer is correct. The values shown in the figure are the correct ones. We will update the text accordingly.
>
> We thank the reviewer again for the positive and constructive comments.

---

### Decision · Action_Editor_2f9K · 2026-07-11

**Recommendation:** Accept as is

**Audience:**

Yes

**Audience Explanation:**

The paper is likely to be of interest for theoretical machine learning researchers, many of whom publish regularly in TMLR.

**Claims And Evidence:**

Yes

**Claims Explanation:**

The paper proposes and theoretically analyzes sketching-style methods for approximating two types of kernels (the projection kernel, and the Binet-Cauchy kernel) defined on the Grassmanian manifold G(k,n). The main technical contribution is the right assembly of proof ingredients that have arisen in previous works on kernel approximations and extending them to the case of the Grassmanian. All reviewers agreed that the paper is worthy of publication, and especially appreciated the clarity of the presentation.